**Parametric soil water retention models: a critical evaluation of expressions for the full moisture range**

Raneem Madi[1], Gerrit Huibert de Rooij[1], Henrike Mielenz[2], Juliane Mai[3]

[1]Dept. Soil System Science, Helmholtz Centre for Environmental Research – UFZ, Halle, Germany

[2]Institute for Crop and Soil Science, Julius Kühn-Institut – JKI, Braunschweig, Germany

[3]Dept. Computational Hydrosystems, Helmholtz Centre for Environmental Research – UFZ, Leipzig, Germany
Currently at Dept. Civil & Environmental Engineering, Univ. of Waterloo, Canada

*Correspondence to*: Gerrit H. de Rooij (gerrit.derooij@ufz.de)

**Keywords:** soil physics**,** soil water retention functions, parameter estimation, soil hydraulic conductivity functions, unsaturated zone.

**Abstract.** Few parametric expressions for the soil water retention curve are suitable for dry conditions. Furthermore, expressions for the soil hydraulic conductivity curves associated with parametric retention functions can behave unrealistically near saturation. We developed a general criterion for water retention parameterizations that ensures physically plausible conductivity curves. Only three of the 18 tested parameterizations met this criterion without restrictions on the parameters of a popular conductivity curve parameterization. A fourth required one parameter to be fixed.

We estimated parameters by Shuffled Complex Evolution with the objective function tailored to various observation methods used to obtain retention curve data. We fitted the four parameterizations with physically plausible conductivities as well as the most widely used parameterization. The performance of the resulting 12 combinations of retention and conductivity curves was assessed in a numerical study with 751 days of semi-arid atmospheric forcing applied to unvegetated, uniform, 1-m freely draining columns for four textures.

Choosing different parameterizations had a minor effect on evaporation, but cumulative bottom fluxes varied by up to an order of magnitude between them. This highlights the need for a careful selection of the soil hydraulic parameterization that ideally does not only rely on goodness-of-fit to static soil water retention data but also on hydraulic conductivity measurements.

Parameter fits for 21 soils showed that extrapolations into the dry range of the retention curve often became physically more realistic when the parameterization had a logarithmic dry branch, particularly in fine-textured soils where high residual water contents would otherwise be fitted.

**1. Introduction**

The pore architecture of the soil influences its hydraulic behavior, typically described by two curves: the relationship between the amount of water present in the soil pores and the matric potential (termed soil water characteristic or soil water retention curve), and the relationship between the hydraulic conductivity and either matric potential or water content (the soil hydraulic conductivity curve). Numerical solvers of Richards' equation for water flow in unsaturated soils require these curves as descriptors of the soil in which the movement of water should be calculated. Many parametric expressions for the retention curve and fewer for the hydraulic conductivity have been developed for that purpose (see the supplemental material, Leij et al. (1997), Cornelis et al., (2005), Durner and Flühler (2005), Khlosi el al. (2008), and Assouline and Or (2013)).

A brief overview of retention curve parameterizations is given in the following while the references to the parameterizations in question are given in the supplemental material and section 2, where their equations are presented. The earliest developed parameterizations focused primarily on the wet end of the curve since this is the most relevant section for agricultural production. Numerical models were struggling with the discontinuity of the first derivative at the air-entry value. Observations with methods relying on hydrostatic equilibrium (Klute, 1986, p. 644-647) typically gave a more smooth shape around the matric potential where the soil started to desaturate as an artefact of the sample height, as was later demonstrated by Liu and Dane (1995). This led to the introduction of parameterizations that yielded a continuously differentiable curve.

The interest in the dry end of the retention curve was triggered by an increased interest in water scarcity issues (e.g. Scanlon et al., 2006; UN-Water, FAO, 2007; UNDP, 2006). For groundwater recharge under deep vadose zones, the dry end of the soil water retention curve affects both slow liquid water movement in film and corner flow (Tuller and Or, 2001; Lebeau and Konrad, 2010) and vapor phase transport (Barnes and Turner, 1998; de Vries and Simmers, 2002). The earlier parameterizations had an asymptote at a small (or zero) water content. This often gave poor fits in the dry end, and several parameterizations emerged in which the dry branch was represented by a logarithmic function that reached zero water content at some point.

A non-parametric approach was advocated by Iden and Durner (2008). They estimated nodal values of volumetric water content from evaporation experiments and derived a smooth retention curve by cubic Hermite interpolation. They extrapolated the retention function to the dry range and compute a coupled conductivity function based on the Mualem model.

Liu and Dane (1995) were the first to point out that the smoothness of observed curves around the air-entry value could be an artefact related to experimental conditions. Furthermore, it became apparent that a particular parameterization that gave a differentiable curve led to unrealistically large increases of the soil hydraulic conductivity near saturation (Durner, 1994; Vogel et al., 2001). This was eventually linked to the non-zero slope at saturation (Ippisch et al., 2006), implying the existence of unphysically large pores with air-entry values up to zero. This led to the re-introduction of a discrete air-entry value.

Most of the parameterizations are empirical, curve-fitting equations (Kosugi et al., 2002). One exception is the very dry range, where measurement techniques are often not so reliable (e.g., Campbell and Shiozawa, 1992) and were not always employed. The proportionality of the water content in this range to the logarithm of the absolute value of the matric potential that has frequently been invoked conforms to the adsorption theory of Bradley

(1936), which considers adsorbed molecules to build up in a film consisting of layers, with the net force of electrical attraction diminishing with every layer (Rossi and Nimmo, 1994).

The empirical power-law relationship between water content and matric potential introduced by Brooks and Corey (1964) was later given a theoretical foundation by Tyler and Wheatcraft (1990), who showed that the exponent was related to the fractal dimension of the Sierpenski carpet used to model the hierarchy of pore sizes

occurring in the soil. The sigmoid shape of the Kosugi's (1996, 1999) retention curve was derived rigorously from an assumed lognormal distribution of effective pore sizes, making this the only parameterization discussed in this paper developed from a theoretical analysis.

Some soils have different types of pore spaces: one type appears between individual grains. Its architecture is determined by soil texture, and by the geometry of the packing of the individual grains. The second

type appears at a larger scale: the soil may consist of aggregates (e.g., Coppola, 2000, and references therein), and the pore space between these aggregates is very different from those between the grains. Biopores formed by roots that have since decayed, soil fauna, etc. also can create a separate type of pore space. In shrinking soils, a network of cracks may form. The volume and architecture of these pore spaces are essentially independent of the soil texture (Durner, 1994), even though a certain texture may be required for these pores to form. In soils with such distinct

pore spaces, the derivative of the soil water retention curve may have more than a single peak, and for this reason multimodal retention curves have been proposed, e.g., by Durner (1994) and Coppola (2000). Most of the parametric expressions for the soil water retention curve are unimodal though. Durner (1994) circumvented this by constructing a multimodal retention curve by summing up several sigmoidal curves of van Genuchten (1980) but with different parameter values. He presented excellent fits of bimodal retention functions at the price of adding three or four

parameters depending on the chosen parameterization. Priesack and Durner (2006) derived the corresponding expression of the hydraulic conductivity function. Romano et al. (2011) developed a bimodal model based on Kosugi's (1994) curve and derived the associated hydraulic conductivity function. Coppola (2000) used a single-parameter expression for the intra-aggregate pore system superimposed on a 5-parameter expression for the inter-aggregate pores, thereby reducing the number of fitting parameters and the degree of correlation among these. The

primary focus of this paper is on unimodal functions, but we briefly discuss three multimodal models as well.

The wealth of parameterizations for the soil water retention curve calls for a robust fitting method applicable to various parameterizations and capable of handling data with different data errors. These errors arise from the various measurement techniques used to acquire data over the full water content range. Parameter fitting codes are available (e.g., Schindler et al., 2015), but they do not fit the parameterizations focusing on the dry end.

The first objective of this paper is to introduce a parameter fitting procedure that involves an objective function that accounts for varying errors, embedded in a shell that allows a wide spectrum of retention function parameterizations to be fitted.

The analysis by Ippisch et al. (2006) of the effect of the shape of the soil water retention curve on the hydraulic conductivity near saturation considered van Genuchten's (1980) parameterization in combination with

Mualem's (1976) conductivity model only. Iden et al. (2015) approached the same problem but only examined the conductivity curve. They too focused on the van Genuchten-Mualem configuration only. The analysis of Ippisch et

al. (2006) could well have ramifications for other parameterizations. A second objective of this paper therefore is the development of a more general analysis based on Ippisch et al. (2006) and its application to other parameterizations of the retention and conductivity curves.

Several hydraulic conductivity parameterizations that relied only on observations of soil water retention data have been developed (see the reviews by Mualem (1992) and Assouline and Or (2013)). Many of these consider the soil layer or sample for which the conductivity is sought as a slab of which the pore architecture is represented by a bundle of cylindrical tubes with a given probability density function (pdf) of their radii. This slab connects to another slab with a different pore radius pdf. By making different assumptions regarding the nature of the tubes and

their connectivity, different expressions for the unsaturated hydraulic conductivity can be found (Mualem and Dagan, 1978). Raats (1992) distinguished five steps in this process: 1) Specify the effective areas occupied by connected pairs of pores of different radii that reflect the nature of the correlation between the connected pore sizes; 2) Account for tortuosity in one of various ways; 3) Define the effective pore radius as a function of both radii of the connected pairs of pores; 4) Convert the pore radius to a matric potential at which the pore fills or empties; 5) Use

the soil water retention curve to convert from a dependence upon the matric potential to a dependence upon the water content. Only step 5 constitutes a direct effect of the choice of the retention curve parameterization on the conductivity curve. Choices made in steps 1-3 result in different conductivity curves associated with any particular retention curve parameterization.

        These conductivity parameterizations give the hydraulic conductivity as a function of matric potential or

water content relative to the value at saturation. They therefore require a value for the saturated hydraulic conductivity, either independently measured or estimated from soil properties. Assouline and Or (2013) review numerous expressions for the saturated hydraulic conductivity. Interestingly, approaches have emerged to estimate the saturated hydraulic conductivity from the retention curve parameters (Nasta et al., 2013; Pollacco et al., 2013, 2017).

The functions based on the pore bundle approach discussed by Mualem and Dagan (1978), Mualem (1992), and Raats (1992) that have found widespread application in numerical models can be captured by Kosugi's (1999) generalized model. In this paper, we limit ourselves to three parameterizations as special cases of Kosugi's general model, and discuss them in more detail in section 2. In doing so, we add to the existing body of comparative studies of parametric retention curves by explicitly including the associated hydraulic conductivity curves according

to these conductivity models. Papers introducing new parameterizations of the soil water retention curve as well as reviews of such parameterizations typically show the quality of the fit to soil water retention data (e.g., van Genuchten, 1980; Rossi and Nimmo, 1994; Cornelis et al., 2005; Khlosi et al., 2008). The role of these parameterizations is to be used in solutions of Richards' equation, usually in the form of a numerical model. Their performance can therefore be assessed through the water content and water fluxes in the soil calculated by a

numerical Richards solver. This is not often done, one exception being the field-scale study by Coppola et al. (2009) comparing unimodal and bimodal retention curves and the associated conductivity curves in a stochastic framework on the field scale, for a 10-day, wet period. A third objective therefore is to carry out a numerical modeling exercise to examine the differences in soil water fluxes calculated on the basis of various parameterizations by the same

model for the same scenario. By doing so, the inclusion of the conductivity curves in the comparison is taken to its logical conclusion by carrying out simulations for all possible combinations of retention and conductivity models.

Should the differences in the fluxes be small, the choice of the parameterizations can be based on convenience. If they are significant, even if the fits to the data are fairly similar, this points to a need of a more thorough selection process to determine the most suitable parameterization.

## 2. Theory

### 2.1 Hydraulic conductivity models and their behavior near saturation

Numerous functions have been proposed to describe the soil water retention curve, several of them reviewed below. Fewer functions exist to describe the soil hydraulic conductivity curve. When these rely on the retention parameters, one can use the retention curve to predict the conductivity curve. However, when both retention and conductivity data exist, a single set of parameters does not always fit both curves well, even if both sets of data are used in the fitting process. It may therefore be prudent to attempt to find a retention-conductivity pair of curves that share a number of parameters that could be fitted on retention data only and has additional parameters that only occur in the expression for the hydraulic conductivity.

Various theoretical models exist to determine the unsaturated hydraulic conductivity $K$ [$LT^{-1}$] as a function of matric potential $h$ [L] or volumetric water content $\theta$ from the soil water retention curve (see the Appendix for a list of the variables used in this paper). Hoffmann-Riem et al. (1999) and Kosugi (1999) identified a generalized model that captured the two most widely used hydraulic conductivity models and several others. The formulation according to Kosugi (1999) is:

$$K(S_e) = K_s S_e^\tau \left( \frac{\int_0^{S_e} |h|^{-\kappa}(x)dx}{\int_0^1 |h|^{-\kappa}(x)dx} \right)^\gamma \tag{1}$$

where the subscript $s$ denotes the value at saturation, $x$ is an integration variable, and $\gamma$, $\kappa$, and $\tau$ are dimensionless shape parameters. The degree of saturation $S_e$ is defined as:

$$S_e(h) = \frac{\theta(h) - \theta_r}{\theta_s - \theta_r} \tag{2}$$

where the subscript $r$ denotes the irreducable value ($\geq 0$). After a change of variables this gives (Ippisch et al. 2006)

$$K(h) = K(h(S_e)) = \begin{cases} K_s S_e^{\tau} \left( \dfrac{\displaystyle\int_{-\infty}^{h(S_e)} |h|^{-\kappa} \dfrac{dS}{dh} \, dh}{\displaystyle\int_{-\infty}^{h_{ae}} |h|^{-\kappa} \dfrac{dS}{dh} \, dh} \right)^{\gamma}, & h \leq h_{ae} \\[30pt] K_s, & h \geq h_{ae} \end{cases} \tag{3}$$

where $h_{ae}$ [L] is the air-entry value of the soil and $S$ denotes the degree of saturation moving between 0 and the actual value $S_e$. Note that the value of $S(h)$ and $dS/dh$ are directly related to the soil water retention curve $\theta(h)$ through Eq. (2). Specific models can be found by fixing the parameters: Burdine's (1953) model is obtained with $\gamma = 1$, $\kappa = 2$, and $\tau = 2$, the popular model of Mualem (1976) results when $\gamma = 2$, $\kappa = 1$ and $\tau = 0.5$, and the model of Alexander and Skaggs (1986) requires $\gamma = \kappa = \tau = 1$. Assouline and Or (2013) give parameter values for additional conductivity models. When any of these models are used, the soil water retention parameters can be used to predict the conductivity curve if no conductivity data are available and the saturated hydraulic conductivity can be estimated independently (see Jarvis et al., 2002, and references therein). Note that positive values of $\kappa$ ensure that large pores (emptying at smaller values of $|h|$) contribute more to the overall hydraulic conductivity than small pores, which is physically sound. Parameter $\gamma$ should be positive as well. Negative values would lead to a switch of the numerator and denominator (which scales the numerator by its maximum value) in Eq. (1), which is illogical. Peters (2014) required that the conductivity curve monotonically decreases as the soil dries out and derived a minimal value of -2 for $\tau$ from that requirement. Indeed, negative values of this parameter have been reported (e.g. Schaap and Leij, 2000), even though the three predictive models mentioned above all have positive values of $\tau$.

Driven by the occasionally unrealistic shape of Mualem's (1976) hydraulic conductivity curve near saturation, Ippisch et al. (2006) rigorously analyzed the version of Eq. (3) specific to Mualem's (1976) model. They concluded that the integrand must approach zero near saturation in order to prevent unrealistically large virtual pores dominating the hydraulic conductivity of very wet soils, a point raised earlier by Durner (1994). We generalize their criterion for prohibiting excessively larger pores from dominating the conductivity near saturation for arbitrary parameter values (after converting $dS/dh$ to $d\theta/dh$) by

$$\lim_{h \to 0} \left( |h|^{-\kappa} \frac{d\theta}{dh} \right) = 0 \tag{4}$$

This condition is automatically met by retention curves with non-zero air-entry values, but restricts the permissible value of $\kappa$ if the retention curve has non-zero derivatives at saturation, and couples it to this derivative.

Iden et al. (2015) argued that limiting the maximum pore size of the pore-bundle models that gave rise to models of the type of Eq. (1) eliminated the large pores that caused the excessively rapid rise of the hydraulic conductivity near saturation. By only modifying the conductivity function without changing the water retention

function, a discrepancy emerges between the retention curve (which reflects the presence of unphysically large pores) and the conductivity curve (which does not). Retention curves with a distinct air-entry value maintain the desired consistency, at the price of having non-continuous derivatives. Computational tests by Ippsisch et al. (2006) suggest that state-of-the-art numerical solvers of Richards' equation are capable of handling this.

**2.2.1 Critical evaluation of unimodal parametric functions of the soil water retention curve**

The supplement reviews 18 parameterizations of the soil water retention curve. Their derivatives are presented and used to verify the physical plausibility of the hydraulic conductivity near saturation according to Eq. (4). In this section only those equations that satisfy the criterion in Eq. (4) are presented, together with the associated
hydraulic conductivity functions. For comparison, the most widely used parameterization is also included here. To facilitate cross-referencing between the supplement and the main text, the equations lifted from the supplement into the main text have the same number in the main text as in the supplement.

The water retention function of Brooks and Corey (1964) is

$$\theta(h) = \begin{cases} \theta_r + \left(\theta_s - \theta_r\right)\left(\dfrac{h}{h_{ae}}\right)^{-\lambda}, & h \le h_{ae} \\ \theta_s, & h > h_{ae} \end{cases} \tag{S1a}$$

where $\lambda$ is a dimensionless fitting parameter. This equation is referred to as BCO below. The analytical expression for the generalized $K(h)$ function (Eq. (3)) for the water retention function of Brooks and Corey (1964) is

$$K(h) = \begin{cases} K_s\left(\dfrac{h(S_e)}{h_{ae}}\right)^{-\lambda\tau}\left\{\dfrac{\left[\dfrac{\lambda\left(\theta_s - \theta_r\right)\left|h_{ae}\right|^{\lambda}}{\kappa + \lambda + 2}\left|h\right|^{-\kappa-\lambda}\right]_{-\infty}^{h}}{\left[\dfrac{\lambda\left(\theta_s - \theta_r\right)\left|h_{ae}\right|^{\lambda}}{\kappa + \lambda + 2}\left|h\right|^{-\kappa-\lambda}\right]_{-\infty}^{h_{ae}}}\right\}^{\gamma} = K_s\left(\dfrac{h_{ae}}{h}\right)^{\lambda(\gamma+\tau)+\gamma\kappa}, & h \le h_{ae} \\ K_s, & h > h_{ae} \end{cases} \tag{S1c}$$

Van Genuchten's (1980) formulation is continuously differentiable:

$$\theta(h) = \theta_r + \left(\theta_s - \theta_r\right)\left(1 + \left|\alpha h\right|^n\right)^{-m}, \quad h \le 0 \tag{S4a}$$

where $\alpha$ [L$^{-1}$], $n$, and $m$ are shape parameters. Often $m$ is set equal to $1 - 1/n$. This equation is denoted by VGN below. The hydraulic conductivity only exhibits acceptable behavior near saturation if $\kappa < n$-1. For many fine and/or

poorly sorted soil textures, $n$ ranges between 1 and 2. Therefore, this restriction even excludes Mualem's (1976) conductivity model when $n < 2$. For this reason we refrain from formulating analytical conductivity equations, even though van Genuchten (1980) presented such expressions for Burdine's (1953) and Mualem's (1976) models. Because of its popularity we will include it in the further evaluation anyway.

Ippisch et al. (2006) proposed to introduce an air-entry value and scale the unsaturated portion of VGN by its value at the water-entry value:

$$\theta(h) = \begin{cases} \theta_r + (\theta_s - \theta_r)\left(\dfrac{1 + |\alpha h|^n}{1 + |\alpha h_{ae}|^n}\right)^{-m}, & h < h_{ae} \\ \theta_s, & h \geq h_{ae} \end{cases}$$  (S7a)

This equation is labeled VGA below. With the common restriction of $m = 1 - 1/n$, an expression can be found for $\kappa = 1$ that is slightly more general than Eq. (11) in Ippisch et al. (2006):

$$K(h) = \begin{cases} K_s\left(\dfrac{\theta - \theta_r}{\theta_s - \theta_r}\right)^{\tau}\left[\dfrac{1 - \left(1 - \dfrac{1}{B(h)}\right)^{\frac{n}{n-1}}}{1 - \left(1 - \dfrac{1}{C}\right)^{\frac{n}{n-1}}}\right]^{\gamma} \\ = K_s\left(\dfrac{B(h)}{C}\right)^{\tau\left(\frac{1}{n}-1\right)}\left[\dfrac{1 - \left(1 - \dfrac{1}{B(h)}\right)^{\frac{n}{n-1}}}{1 - \left(1 - \dfrac{1}{C}\right)^{\frac{n}{n-1}}}\right]^{\gamma}, & h < h_{ae} \\ K_s, & h \geq h_{ae} \end{cases}$$  (S7c)

where

$$B(h) = 1 + |\alpha h|^n$$  (S7d)

$$C = 1 + |\alpha h_{ae}|^n$$  (S7e)

This equation can be used to define conductivity models according to Mualem (1976) and Alexander and Skaggs (1986), which both require that $\kappa = 1$.

Rossi and Nimmo (1994) preferred a logarithmic function over the Brooks-Corey power law at the dry end to better represent the adsorption processes that dominates water retention in dry soils, as opposed to capillary processes in wetter soils. They also implemented a parabolic shape at the wet end as proposed by Hutson and Cass (1987). Rossi and Nimmo (1994) presented two retention models, but only one (the junction model) permitted an analytical expression of the unsaturated hydraulic conductivity. Here, we modified the junction model by removing the parabolic expression for the wet end of the retention curve in favor of the discontinuous derivative at the air-entry value:

$$\theta(h) = \begin{cases} 0, & h \le h_d \\ \theta_s \beta \ln\left(\dfrac{h_d}{h}\right), & h_d < h \le h_j \\ \theta_s \left(\dfrac{h_{ae}}{h}\right)^\lambda, & h_j < h \le h_{ae} \\ \theta_s, & h > h_{ae} \end{cases} \tag{S9a}$$

which is denoted RNA below.

Rossi and Nimmo (1994) required the power law and logarithmic branches as well as their first derivatives to be equal at the junction point ($\theta_j$, $h_j$). With $h_d$ fixed (Rossi and Nimmo found a value of $-10^5$ m for six out of seven soils and $-5\cdot10^5$ m for the seventh), these constraints allow two of the five remaining free parameters to be expressed in terms of the other three. Some manipulation leads to the expressions:

$$\lambda = \frac{1}{\ln|h_d| - \ln|h_j|} \tag{S9c}$$

$$\beta = \lambda \left(\frac{h_{ae}}{h_j}\right)^\lambda \tag{S9d}$$

This gives the fitting parameters $h_{ae}$, $h_j$, and $\theta_s$. The associated conductivity model is

$$K(h) = \begin{cases} 0, & h \le h_d \\[2em] K_s S_e^\tau \left\{ \dfrac{\left[ -\dfrac{\theta_s \beta}{\kappa} |h|^{-\kappa} \right]_{h_d}^{h}}{\left[ -\dfrac{\theta_s \beta}{\kappa} |h|^{-\kappa} \right]_{h_d}^{h_j} - \left[ \dfrac{\theta_s \lambda}{\lambda + \kappa} |h_{ae}|^\lambda |h|^{-(\lambda+\kappa)} \right]_{h_j}^{h_{ae}}} \right\}^\gamma \\[2em] = K_s \left[ \beta \ln\left( \dfrac{h_d}{h} \right) \right]^\tau \left[ \dfrac{E(h)}{E(h_j) + F\left( |h_j|^{-\lambda-\kappa} - |h_{ae}|^{-\lambda-\kappa} \right)} \right]^\gamma, & h_d < h \le h_j \\[2em] K_s S_e^\tau \left\{ \dfrac{\left[ -\dfrac{\theta_s \beta}{\kappa} |h|^{-\kappa} \right]_{h_d}^{h_j} - \left[ \dfrac{\theta_s \lambda}{\lambda + \kappa} |h_{ae}|^\lambda |h|^{-(\lambda+\kappa)} \right]_{h_j}^{h}}{\left[ -\dfrac{\theta_s \beta}{\kappa} |h|^{-\kappa} \right]_{h_d}^{h_j} - \left[ \dfrac{\theta_s \lambda}{\lambda + \kappa} |h_{ae}|^\lambda |h|^{-(\lambda+\kappa)} \right]_{h_j}^{h_{ae}}} \right\}^\gamma \\[2em] = K_s \left( \dfrac{h_{ae}}{h} \right)^{\lambda\tau} \left[ \dfrac{E(h_j) + F\left( |h_j|^{-\lambda-\kappa} - |h|^{-\lambda-\kappa} \right)}{E(h_j) + F\left( |h_j|^{-\lambda-\kappa} - |h_{ae}|^{-\lambda-\kappa} \right)} \right]^\gamma, & h_j < h \le h_{ae} \\[2em] K_s, & h > h_{ae} \end{cases}$$

(S9e)

where

$$E(h) = \frac{\beta}{\kappa} \left( |h_d|^{-\kappa} - |h|^{-\kappa} \right)$$

(S9f)

$$F = \frac{\lambda}{\lambda + \kappa} |h_{ae}|^\lambda$$

(S9g)

Fayer and Simmons (1995) used the approach of Campbell and Shiozawa (1992) to have separate terms for adsorbed and capillary bound water. If the capillary binding is represented by a Brooks-Corey type function, the retention model becomes

$$\theta(h) = \begin{cases} 0, & h \le h_d \\[1em] \theta_a \left( 1 - \dfrac{\ln|h|}{\ln|h_d|} \right) + \left[ \theta_s - \theta_a \left( 1 - \dfrac{\ln|h|}{\ln|h_d|} \right) \right] \left( \dfrac{h_{ae}}{h} \right)^\lambda, & h_d < h < h_{ae} \\[1em] \theta_s, & h \ge h_{ae} \end{cases}$$

(S12a)

This expression is denoted FSB below. Note that this model is valid if $h_{ae}$ does not exceed -1 cm. This condition will usually be met, unless the soil texture is very coarse. The corresponding conductivity model is

$$K(h) = \begin{cases} 0, & h \le h_d \\ K_s S_e^\tau \left\{ \dfrac{\left[ \dfrac{|h_{ae}|^\lambda}{\ln|h_d|(\lambda+\kappa)} \left[ \theta_a \left( \dfrac{\lambda+\kappa-1}{\lambda+\kappa} - \ln|h| \right) - \lambda(\theta_s - \theta_a)\ln|h_d| \right] |h|^{-\lambda-\kappa} \right]_{h_d}^{h}}{\left[ \dfrac{|h_{ae}|^\lambda}{\ln|h_d|(\lambda+\kappa)} \left[ \theta_a \left( \dfrac{\lambda+\kappa-1}{\lambda+\kappa} - \ln|h| \right) - \lambda(\theta_s - \theta_a)\ln|h_d| \right] |h|^{-\lambda-\kappa} \right]_{h_d}^{h_{ae}}} \right\}^\gamma \\ \quad = K_s \left\{ \dfrac{\theta_a}{\theta_s}\left(1 - \dfrac{\ln|h|}{\ln|h_d|}\right) + \left[1 - \dfrac{\theta_a}{\theta_s}\left(1 - \dfrac{\ln|h|}{\ln|h_d|}\right)\right]\left(\dfrac{h_{ae}}{h}\right)^\lambda \right\}^\tau \\ \quad \left\{ \dfrac{[\theta_a(G - \ln|h|) - I]|h|^{-\lambda-\kappa} - J}{[\theta_a(G - \ln|h_{ae}|) - I]|h_{ae}|^{-\lambda-\kappa} - J} \right\}^\gamma, & h_d < h \le h_{ae} \\ K_s, & h \ge h_{ae} \end{cases}$$

(S12c)

where

$$G = \frac{\lambda + \kappa - 1}{\lambda + \kappa}$$

(S12d)

$$I = \lambda(\theta_s - \theta_a)\ln|h_d|$$

(S12e)


$$J = \left[\theta_a(G - \ln|h_d|) - I\right]|h_d|^{-\lambda-\kappa}$$

(S12f)

    In the original equations as presented by Fayer and Simmons (1995), the adsorbed water content reached zero at $h_d$, while there is still some capillary bound water at and below that matric potential, which is inconsistent.
Furthermore, the terms with ratios of logarithms become negative for matric potentials below $h_d$. We therefore modified the original equations by setting the water content to zero below $h_d$.

    In the supplement we argue that most of the retention curves examined result in conductivity curves with physically unacceptable behavior near saturation, even though several of these expressions were derived with the explicit purpose of providing closed-form expressions for the hydraulic conductivity. Only the Brooks-Corey
function (1964) (BCO, Eq. (S1a)), the junction model of Rossi and Nimmo (1994) without the parabolic correction (RNA, Eq. (S9a)), and the model of Fayer and Simmons (1995) based on the Brooks-Corey (1964) retention

function (FSB, Eq. (S12a)) lead to an acceptable conductivity model with full flexibility (three free parameters: $\kappa$, $\gamma$, $\tau$). The modified van Genuchten (1980) retention curve with a distinct air-entry value by Ippisch et al. (2006) (VGA, Eq. (S7a)) leads to a conductivity model with two fitting parameters if $m = 1- 1/n$ because $\kappa = 1$.


### 2.2.2 Multimodal parametric functions of the soil water retention curve

The multimodal model of Durner (1994) is a weighted sum of van Genuchten's (1980) retention functions (Eq. (S4a)) with zero residual water content. The bimodal retention model of Coppola (2000) adds a rapidly

decaying asymptotic function representing the aggregate pore space to Eq. (S4a), also with zero residual water content. Because they are derived from Eq. (S4a), neither multimodal retention model meets the criterion of Eq. (4). The asymptotic nature of the dry end of either multimodal retention model limits their usefulness under very dry conditions.

The bimodal model of Romano et al. (2011) consists of two of Kosugi's (1994) retention functions.

Romano et al.'s expression for the derivative shows that at least for $\kappa = 1$ the criterion of Eq. (4) is met. The asymptotic dry end that was removed in the unimodal version by Khlosi et al. (2008) (Eq. (S14a) remains though, limiting its applicability in dry soils. Khlosi et al.'s modification led to additional complications detailed in the supplement, which is why we did not pursue this for the bimodal version. The remainder of the paper therefore only considers the unimodal models discussed above.


## 3. Materials and methods

### 3.1. Soil water retention and hydraulic conductivity data

### 3.1.1 Soil hydraulic data for the model simulations

Data were obtained from Schelle et al. (2013) who measured soil water retention curves for a range of soil textures (clay, silt, silt loam, and sand). They took undisturbed and disturbed samples of a silt loam, a silt, and a sand near Braunschweig (northern Germany), and of a clay near Munich (southern Germany). The retention data were measured on soil samples using different laboratory methods and cover the moisture range from saturation to near oven dryness at pF approximately 7. For silt, silt loam, and sand they used data obtained by suction plates, pressure

plates and the dew point method. For clay they used data from the evaporation method HYPROP[®] (UMS, 2015) (until pF 3), pressure plate and dew point methods. Here, we trimmed the disproportionally large data set in the HYPROP[®] range by stratifying the data into intervals of 0.5 on the pF scale and then randomly picking one data point for each interval. This ensured an adequate sensitivity of the fit in the dry range for all textures. For some of the soil samples, hydraulic conductivity data were available, including the values at saturation (unpublished).

Hydraulic conductivity data were obtained by the evaporation method according to Peters and Durner (2008).

Undisturbed samples of 4.0 cm height and 100 cm$^3$ volume were used for the suction plate method, with 4 to 6 replicates for each soil. The HYPROP[®] setup worked with an undisturbed sample of 5.0 cm height and 250 cm$^3$ volume (one replicate). The pressure plate method required disturbed samples of 1.0 cm height and 5.2 cm$^3$ volume

(5 or 6 replicates for each soil). The dew point method worked with disturbed samples of approximately 10 g dry

mass (7 to 24 replicates with pF values between 3.5 and 6.2). Additional details are given by Schelle et al. (2013).

The fitting routine uses the variance of the data error to determine the weighting factor each data point. We estimated these on the basis of estimated measurement errors of water level readings, pressure gauges, sample masses, etc. Typically, the estimated standard deviation in the matric potential was 0.05 cm for $h = 0$, in the range of the sandbox apparatus (> -200 cm) it was 1.0 cm, and beyond that it was 10 cm. For the water content, the

estimated standard deviation was 0.01 at saturation and 0.02 anywhere else. If we had specific information about the accuracy of the instruments and their gauges and scales, these values were adapted accordingly.

When the three conductivity parameters are set to the values dictated by Burdine (1953), Mualem (1976), or Alexander and Skaggs (1986), hydraulic conductivity curves can be derived from soil water retention data only, supplemented by an estimate for the saturated hydraulic conductivity. For the soils with available conductivity data

we compared the hydraulic conductivity curves to the direct measurements.

**3.1.2 Soil water retention data used to evaluate various retention curve parameterizations**

We selected 21 soils from the UNSODA database (Nemes et al., 2001; National Agricultural Library

website). The database has relatively many records for sandy soils, and hardly any in heavy clays. The selected soils do not have organic matter contents that would lead to considering them as organic soils, have texture data records that allows their texture class to be determined, are fairly uniformly distributed over the textures covered by the database, have data points on the main drying curve, and have measurements over a sufficiently wide range of matric potentials to allow retention curves to be fitted to them.

We classified the texture of the selected soils according to the USDA classification as well as the hydrologically-oriented classification developed by Twarakavi et al. (2010). The latter distinguishes 12 texture classes, grouped in three sets (A, B, C) of four each (1 through 4). Soils with (nearly) 100% sand, silt, or clay are classified as A1, B1, and C1, respectively. Numbers larger than 1 identify texture classes that must have at least two of the components sand, silt, and clay. B3 and C4 are the only categories that must have all three components. The

differences with the USDA classification are considerable for clayey and silty soils, and we refer to Twarakavi et al. (2010) for full details. Figure 1 shows the distribution of the selected soils over the soil texture triangle.

**3.2. Parameter fitting**

**3.2.1. Selected parameterizations**


We fitted the original Brooks-Corey (BCO, Eq. (S1a)) and van Genuchten (VGN, Eq. (S4a)) parameterizations, and the derivates thereof that do not lead to unrealistic hydraulic conductivities near saturation: FSB (Eq. (S12a)) and RNA (Eq. (S9a)), both of which emerged from BCO, and VGA (Eq. (S7a)), which emerged from VGN. Thus, BCO, FSB, and RNA all have a power law shape in the mid-range of the matric potential (and for

BCO over the full range below the air-entry value). The slope therefore monotonically increases with decreasing

water content. VGN and VGA have a sigmoid shape and therefore are able to fit curves that have an inflection point. As Groenevelt and Grant (2004) pointed out, $\theta_r$ serves as the third required shape parameter for curves with an inflection point, frequently resulting in improbable values for this parameter. Table 1 shows the fitting parameters and their physically permitted range.

All three conductivity models are compatible with BCO, FSB and RNA. Burdine's (1953) and Mualem's (1976) conductivity models can be used with VGA. VGN does not meet the criterion of Eq. (4) but is very often used in conjunction with Mualem's conductivity model (1976). It was therefore included for comparison.

### 3.2.2. The objective function and its weighting factors


         A set of parameters describing the soil water retention curve must be optimized to provide the best fit to an arbitrary number of data points. To do so, an objective function was minimized, construed by the sum of weighted squares of the differences between observed and fitted values. The fitted values depend on the parameter values in the parameter vector $\mathbf{x}$. Assume $q_\theta$ observation pairs of water content vs. matric head $(h_i, \theta_i)$. Here, $\theta_i$ denotes the $i$th

observation of the volumetric water content, $h_i$ [L] is the matric head at which that water content was observed (expressed as an equivalent water column), and $i \in \{1,2,\ldots,q_\theta\}$ is a counter. In the code, the assumed units are cm water column for $h$ and $cm^3$ $cm^{-3}$ for $\theta$.

         The definition of the objective function $F_R(\mathbf{x}_{p,R})$ at the $R^{th}$ iteration during the fitting operation is:

$$F_R(\mathbf{x}_{p,R}) = \mathbf{w}_{\theta,R}{}^T \mathbf{d}_\theta(\mathbf{x}_{p,R}, \mathbf{x}_f) \quad R \in \{1,2,\ldots,R_{max}\}$$
(5)

Here, $\mathbf{d}_\theta$ denotes a vector of length $q_\theta$ of squared differences between observations and fits that are functions of the fitted parameter values $\mathbf{x}_p$ and the fixed (non-fitted) parameters in vector $\mathbf{x}_f$. Together, $\mathbf{x}_p$ and $\mathbf{x}_f$ constitute $\mathbf{x}$. Each squared difference is weighted. The weight factor vector is denoted by $\mathbf{w}_{\theta,R}$. Its dependence on the water content and

iteration step is explained below. The superscript T indicates that the vector is transposed. To terminate infinite loops, the number of iterations is capped by $R_{max}$.

         For relatively wet soils ($0 > h > -100$ to $-200$ cm), measurement methods are available that create a hydrostatic equilibrium in a relatively large sample. In such cases $h_i$ reflects the matric potential at the center of the sample but $\theta_i$ is that determined for the entire sample. The vertical variation of $h$ results in a non-uniform water

content, and the average water content of the sample ($\theta_i$) may not be well represented by the water content corresponding to $h_i$. For these cases, the height of the sample can be specified on input. The code then divides the sample into 20 layers, calculates $h$ in the center of each layer, computes the corresponding water contents from $\mathbf{x}_{p,R}$, and averages these to arrive at an estimate of $\theta_i$.

         If and only if the standard deviation of the measurement error of the individual observations is known, a

maximum-likelihood estimate of the soil hydraulic parameters can be obtained (Hollenbeck and Jensen, 1998). To ensure this, the weighting factors in vector $\mathbf{w}_{\theta,R}$ must be equal to the reciprocal of the variance of the measurement error. Note that this choice eliminates any effect of measurement units because the squared differences have the

same units as the variances by which they are divided (Hollenbeck and Jensen, 1998). Only then can model adequacy be examined. A model is considered adequate if the residuals after parameter fitting are solely caused by measurement noise (Hollenbeck et al., 2000). Furthermore, only if these conditions are met can confidence intervals of fitted parameters be determined (Hollenbeck and Jensen, 1998). Even in that case, the contouring of the parameter space for permissible increases of the objective function required to determine the confidence region is not practically feasible for four or more parameters, and very laborious even for fewer parameters. A popular approximation based on the Cramer-Rao theorem was shown to be rather poor by Hollenbeck and Jensen (1998), so we refrained from implementing it. Instead we record the evolution of the parameter values through the iterative process. Low information content (indicated by large random fluctuations of a parameter value), correlated parameters, and parameters trending towards a minimum or maximum permitted value can usually be diagnosed from such records.

Data points for a retention curve over the whole moisture range cannot be obtained by a single method. Furthermore, measurement errors occur in both $h_i$ and $\theta_i$. To accommodate this, the error standard deviations $\sigma_{h,i}$ and $\sigma_{\theta,i}$ for $h$ and $\theta$, respectively can be provided individually for any data point $i$. To improve the performance of the fitting routine, the values of $\sigma_{\theta,i}$ are scaled to ensure their average equals 0.20, i.e., the same order of magnitude as $\theta$. The values of $\sigma_{h,i}$ are then scaled by the same scaling factor. The weighting factor $w_{R,I}$ for observation $\theta_i$ during iteration $R$ is:

$$w_{R,i} = \sigma^*_{i,R}{}^{-2} = \left( \sigma^*_{h,i} \frac{\mathrm{d}\theta}{\mathrm{d}h} \Big|_{R,i} + \sigma^*_{\theta,i} \right)^{-2} \tag{6}$$

where the asterisk denotes a scaled value. The subscripts $i$ and $R$ label data points and iteration steps as above. The gradient is determined from the $R^{\text{th}}$ fitted $\theta(h)$ relationship defined by $\mathbf{x}_{p,R}$. Thus, the weighting factors are updated for every iteration.

In the code, the gradient is approximated by $\Delta\theta/\Delta h$ computed from the water contents at $h_i \pm \max(1 \text{ cm } H_2O, 0.01 \cdot h_i)$. For data points acquired at hydrostatic equilibrium, this would require 40 additional calls to the function that computes the $\theta$ corresponding to a given value of $h$, which would be rather inefficient. Instead, the water content is calculated for one virtual layer below and one above the sample. By subtracting the water content of the top (bottom) layer in the sample and adding the water content of the virtual layer below (above) the sample, the water content corresponding to $h_i + H/20$ ($h_i - H/20$) can be found, with $H$ the sample height in cm. In this way, $\Delta\theta/\Delta h$ can be computed with only two additional calls to the function that defines the parameterized $\theta(h)$ relationship.

### 3.2.3. Parameter optimization by Shuffled Complex Evolution

The calibration algorithm employed here is the Shuffled Complex Evolution (SCE) algorithm introduced by Duan et al. (1992) with parameter adjustments of Behrangi et al. (2008). The strategy of this algorithm is to form out

of $j + 1$ parameter sets, where $j$ is the number of model parameters, so-called complexes (e.g. triangles in 2D). Each vertex of the complex not only represents one of the $j + 1$ parameter sets but also the model's skill $F_R(\mathbf{x}_{p,R})$ to match the observed data when it is forced with the according parameter set $\mathbf{x}_{p,R}$. This skill is usually referred to be the objective function value of an objective to be minimized. The vertex with the worst skill or largest objective function value is subsequently perturbed in order to find a better substitute parameter set. This strategy is repeated until the volume of the complex, i.e. the agreement of the parameter sets, is smaller than a threshold. To avoid that the search gets stuck in a local optimum, a number of $Y$ complexes are acting in parallel. After a certain number of iterations the $Y \cdot (j + 1)$ vertexes are shuffled and newly assigned to $Y$ complexes. The algorithm converges when the volume of all complexes is lower than a threshold which means that all $Y \cdot (j + 1)$ vertexes are in close proximity to each other. Infinite runs of the SCE are avoided by $R_{\max}$, but convergence should be the desired target for termination of the SCE.

The SCE algorithm used here is configured with two complexes each consisting of $(2j + 1)$ ensemble members. The different parameterizations we fitted had 3 to 5 fitting parameters. In each iteration, $j + 1$ parameters are randomly selected and the vertex with the worst skill is perturbed. The reflection and contraction step lengths in the Simplex method (e.g., Press et al., 1992, p. 402-404) were set to 0.8 and 0.45, respectively. SCE seems to have an order of about $O(j^2)$. In our case it required between roughly 250 and 3000 model evaluations to find the optimal parameter set. For each parameter estimation run, three sets of initial guesses of the fitting parameters must be provided. The results of the three trials were compared to reduce the chance of accepting a local minimum of the objective function. The selection of SCE was based on its widespread usage in hydrological studies and according to a preliminary experiment where the SCE outperformed other algorithms like the Simulated Annealing (Kirkpatrick et al., 1983) and the Dynamically Dimensioned Search algorithm (Tolson et al., 2007) in optimizing more than 80 analytical test functions with $j$ ranging from 2 to 30.

### 3.3. Scenario study by numerical simulations

As stated in the Introduction, previous tests of parametric expressions of soil water retention functions mostly focused on the quality of the fit to direct observations of points on the water retention curve. Here, we will also examine how the various parameterizations affect the solution of Richards' equation by simulating water fluxes and soil water profiles for a scenario involving infiltration and evaporation. We set up a hypothetical 999-day scenario representative of a desert climate with prolonged drying, infiltration into dry soil, and redistribution after rainfall, permitting a comprehensive test of the parameterizations. We used the HYDRUS 1-D model version 4.xx (Šimůnek et al., 2013, http://www.pc-progress.com/en/Default.aspx?hydrus-1d) to solve Richards' equation in a 1-dimensional soil profile. We permitted flow of liquid water as well as diffusive water vapor fluxes.

We considered an unvegetated uniform soil profile of 1 m depth, initially in hydrostatic equilibrium with -400 cm matric potential at the soil surface. The lower boundary condition was that of free drainage. In combination with the hydrostatic initial condition this briefly caused some rapid drainage immediately after the start of the simulation as the lowest part of the profile adapted to the unit gradient conditions in the two lowest nodes that

the free drainage condition imposed. The upper boundary conditions were atmospheric (during dry periods: prescribed matric potential set to -50000 cm; during rain: prescribed flux density equal to the daily rainfall rate derived from observed daily sums). The weather data (daily rainfall and temperature) were taken from the NOAA data base (http://www.ncdc.noaa.gov/cdo-web/) for a station in Riyadh city (Saudi Arabia) between June 4, 1993 and February 27, 1996. In this period spanning nearly three years, there were three clusters of rainfall events (Fig.

2). The second cluster was the heaviest with a maximum daily sum of approximately 5.4 cm at the day 656. A prolonged dry spell preceded the first rainfall cluster. We used the first 250 days of this period as a 'burn-in' period to minimize the effect of the initial condition on the calculated fluxes. This leaves a period of 751 days for analysis.

The simulation period involved large hydraulic gradients when water infiltrated a very dry soil, limited infiltration of small showers followed by complete removal of all water, deeper infiltration after clusters of rainfall

that delivered large amounts of water followed by prolonged periods in which flow of liquid water and water vapor occurred simultaneously. These processes combined permitted a comprehensive comparison of the various parameterizations. We were interested in the magnitude of the fluxes of liquid water and water vapor and the partitioning of infiltration into evaporation, storage change, and deep infiltration under various conditions, and the effect on these fluxes and storage effects of the choice of parameterization. We did not intend or desire to carry out a

water balance study. Under semi-arid conditions this would have required a much longer meteorological record, which was not available.

The various parameterizations are not implemented in HYDRUS. We therefore used the MATER.IN input file to supply the soil hydraulic property curves in tabular form to the model. The retention models BCO, FSB, and RNA permitted all three conductivity models (Burdine, Mualem and Alexander and Skaggs) to be used. VGA only

gives useful expressions for Burdine and Mualem. VGN only allows Mualem's conductivity model. Thus, there are 12 combinations of retention and conductivity curves that we tested on four different textures, leading to 48 different simulations (and MATER.IN files) in total.

**4. Results and discussion**

**4.1 Fitted parameters and quality of the fits for the soils used in the simulations**

Table 1 presents the fitted parameters for all combinations of texture and parameterization for the soils used in the simulations. The parameter with the best-defined physical meaning is $\theta_s$. All parameterizations give comparable values for it for each texture, which reflects the relatively narrow data clouds near saturation. The values

of $\theta_r$ are relatively high for the three parameterizations in which it occurs. The air-entry values ($h_{ae}$) should increase (move closer to zero) from clay to silt loam to silt to sand, which is the case for BCO, FSB, and RNA, but not for VGA. The data in Fig. 3 support relatively similar values for all textures other than clay, which is somewhat surprising. RNA gives rather high values in silt and sand, and VGA does very poorly in sand and silt loam. The high value for $h_{ae}$ for FSB in clay may be related somehow to the very high value of the maximum adsorbed water

content $\theta_a$, which we fixed close to $\theta_s$. The value of $\theta_a$ for clay should be larger than that for silt loam, so it cannot be more than about 0.2 off though. The spread of $h_j$ for RNA across the textures show that this parameter needs to be

allowed to fitted over its full range (between $h_d$ and at least the minimum value of $h_{ae}$). Even with initial guesses that differed by several orders of magnitude, the fits were still quite consistent, so evidently these values are supported by the data and not an artefact.

In three of the 48 parameter estimation runs, the fits pushed one of the parameters to one of its bounds (even after expanding these to their physical limits), irrespective of their initial guess: FSB for clay (we fixed $\theta_a$ to 0.5), VGN for sand and RNA for silt (we fixed $\theta_s$ on the basis of the data in both cases). For BCO and VGA in sandy soil, the code could not converge to a global minimum, indicated by the volume of the complexes, which exceeded the threshold. The fitted parameters should be viewed critically in these two cases.

The Root Mean Square Error (RMSE) of the fits (Table 2) illustrate why VGN has been very popular for over three decades. It gives the best fit in three cases (sand, silt and silt loam) and the second-best fit in the fourth (clay). BCO performs poorest in three cases (sand, silt and silt loam) and second-poorest in one (clay). The other three have varying positions, with no clearly strong or weak performers. FSB has the best performance in the finest soil (clay).The overall difference in the RMSE values between textures reflects the different scatter in the underlying

data clouds.

The soil water retention curves defined by the different parameterizations are plotted in (Fig. 3). The models that were not developed with dry conditions in mind (BCO, VGA, and VGN) have relatively high water contents in the dry end of clay and silt loam. The logarithmic dry end of FSB and RNA eliminates this asymptotic behavior. The cutoff to zero of the FSB parameterization is quite strong in fine-textured soils. The fixed value of $h_d$

(where the water content is zero) of RNA seems to be too small for clay while appearing adequate for the other textures.

In the intermediate range, all fits are close to one another. RNA underperforms in sand and silt compared to the others. In the wet range, the absence of an air-entry value in VGN results in a poor fit for sand. Here, the contrast between VGN and VGA is very clear. Overall, the inclusion of the water-entry value as a parameter seems

beneficial to the fits. FSB has the most satisfactory overall performance.

For sand, silt, and silt loam, independent observations of $K(h)$ were available. The fits of Burdine's (1953) and Mualem's (1976) parameterizations based on retention data only were remarkably good for all parameterizations. The function of Alexander and Skaggs (1986) severely overestimated the hydraulic conductivity in all three cases, but very accurately described the slope of the curve for silt loam. Fig. 4 demonstrates this for FSB,

the results for the other parameterizations were comparable.

**4.2 Simulation results**

For all simulations, the vapor flux within the profile was of little consequence compared to the liquid water

flow. For that reason it will not be discussed in detail here. Vapor flow may play a larger role under more natural conditions with day-night temperature cycles and in the presence of plant roots.

**4.2.1 Silt**

We start the analysis by examining the flux at the bottom of the soil profile. Panels a-e of (Fig. 5) show all combinations of parameterizations of the retention and conductivity curves.

        The early rainfall cluster event at around $t = 300$ d did not generate any bottom flux, and therefore only wetted up the soil profile. In doing so it increased the effect of the heavier rainfall around $t = 656$ d on the bottom flux.

For the individual parameterizations, Mualem (M) and Burdine (B) gave reasonably similar results in which the second and third rainfall cluster generated a little more downward flow for B than for M. In all cases, Alexander and Skaggs (AS) gave a more rapid response of a very different magnitude. Clearly visible is a sustained, constant flux leaving the column during prolonged dry periods for the AS conductivity curves. This is physically implausible.

        Fig. 5f shows the substantial effect of the parameterization of the water retention curve on bottom fluxes

when the M-type $K(h)$ function is deployed. The results for B-type $K(h)$ were comparable. Different retention curves gave very different responses to the initial conditions (not shown), highlighting the need to add a sufficiently long lead time ahead of the target time window to the simulated time period. RNA's response to the second and third rainfall clusters was about 2.4 times that of the others. At $h = -300$ cm (pF 2.48), $K$ according to M is at least 5 times higer for RNA than for the rest, while the water content at that matric potential and higher values is relatively small

(Fig. 3c). Thus, infiltrated water was transported downward with relative ease, giving rise to the relatively high bottom fluxes and low evaporation rates that were computed for RNA (Figs. 5f, 6f). The parameterizations other than RNA behaved rather similar, except for the fact that VGA responded much faster to a change in the forcings than the other parameterizations.

        Fig. 5g shows the similar comparison of all parameterizations for the AS-type $K(h)$ function. The response

to rainfall was very fast and short-lived, which seems improbable for a silt soil that is far from full saturation. The non-physical bottom flux during dry periods (especially for VGA), the slow calculation times (half as fast as the others) with the time step always at the smallest permitted value, and non-negligible mass balance errors all point to numerical problems associated with AS.

        The evaporative flux was nearly identical for B and M conductivity functions (Fig. 6a-c). Since their

bottom fluxes differed, this necessarily implies that the storage in the soil profile must also be different for B and M. The AS parameterization gave a much more spiky response of evaporative flux to rainfall than B or M, with zero evaporation most of the time (Fig. 6a-d). In terms of cumulative evaporation, AS responded more strongly to the second rainfall cluster around $t = 650$ d (Fig. 6a-c). Overall, the effect of the conductivity function on the relative differences in evaporation was less pronounced than on the relative differences in the bottom flux. The same was

true for the parameterization of the retention curve, as demonstrated by the relatively similar shapes of the curves in panels f and g of Fig. 6.

        Given the non-physical behavior of the bottom flux of AS for VGA in particular (Fig. 5d), we also examined the infiltration. We first compare infiltration for VGA with M and AS-type conductivity (Fig. 7a), and clearly see the zero infiltration for VGA during periods without rain contrasted to the impossible non-zero

infiltration rates for AS during dry spells. For the other water retention parameterizations in combination with AS,

the effect is less pronounced (Fig. 7b). Still, the AS conductivity should be used with care and the results and mass balance checked.

Table 3 summarizes the bottom and evaporative fluxes. For evaporation, the differences are inconsequential except for the markedly low values for RNA. For the bottom flux, the difference between B and M is small enough to be within the margin of error for typical applications. The effect of the parameterization of the retention curve is an order of magnitude between the smallest bottom flux (for VGA) and the largest (for RNA).

### 4.2.2. Sand

The relationship between the bottom (Fig. 8) and evaporative fluxes (Fig. 9) as generated by the various parameterizations for the sandy soil were comparable to those for silt, and the analysis applied to the silt carries over to sand. The bottom fluxes in sand responded faster and with less tailing than in silt, and the third rainfall cluster near the end of the simulation period produced a clear signal (Fig. 8).

The FSB (Fig. 8b) and RNA (Fig. 8c) parameterizations were both in their logarithmic dry range when bottom fluxes occurred, and gave comparable values. BCO is not well adapted for dry conditions, and this is reflected by a bottom flux that is four times lower than the others (Fig 8g).

The bottom fluxes for BCO and FSB with AS-type $K(h)$ are similar (Fig. 8h), in stark contrast to the bottom fluxes based on B (Fig. 8f) and M (Fig. 8g) for these parameterizations. The similarity in the fluxes for AS reflect the facts that the evaporative fluxes (occurring in the wet range, where BCO and FSB both have Brooks-Corey retention curves) are very similar and the spiky response typical for AS results in small difference in storage between BCO and FSB. Consequently, the bottom flux, as the only remaining term of the water balance, cannot differ strongly between BCO and FSB. The difference in the bottom fluxes generated for VGN and VGA with M-type $K(h)$ (Fig. 8g) is even more extreme than in case of the silty soil.

For both B and M conductivity functions, the evaporation (Figs. 9a and b) and the bottom flux (Figs. 8a, f, and g) for BCO differed from the other parameterizations. These differences seem to have been dominated by the complementary responses of evaporation and bottom fluxes to the rainfall events around $t = 656$ d. BCO converted roughly 5-7 cm more of this rainfall to evaporation than the other parameterizations, for both B and M. Therefore, less water was available for downward flow, resulting in a cumulative bottom flux for BCO that was roughly 6 to 8 cm smaller than for the other parameterizations.

The AS-type $K(h)$ function again gave a spiky response (Fig. 9a). Nevertheless, the differences in the evaporation and the bottom flux compared to those of B and M are not very large. The bottom fluxes resulting from rainfall events were considerably smaller for RNA than for the other parameterizations.

Coarse-textured soils have the sharpest drop in the hydraulic conductivity as the soil desaturates. We therefore used the result for the sandy column to study the relationship between the matric potential at the bottom of the column and the bottom flux in order to evaluate water fluxes in dry soils. The free drainage lower boundary condition ensures there is always a downward flux that is equal to the hydraulic conductivity at the bottom at any

time. Particularly for coarse soils this can still lead to negligible bottom fluxes for considerable periods of time. We first consider FSB and RNA, these being the parameterizations specifically developed to perform well in dry soils.

The difference in matric potentials between FSB and RNA is immediately clear from Figs. 10a, b and 11a, b. The effect of the conductivity function is manifest by including Figs. 10c and 11c in the comparison. The effect of the first rainfall cluster is visible in the matric potential in all cases (Figs. 10 and 11), but not enough to generate a significant flux. A flux through the lower boundary first occurs when the matric potential there exceeds (i.e. becomes less negative than) -70 cm for FSB (Fig. 10a and b) and -30 cm for RNA (Figs. 11a and b).

The second rainfall cluster at $600 < t < 700$ d did not rely on prewetting: it produced a bottom flux no matter how dry the soil was. The third rainfall cluster around day 930 probably would not have generated a bottom flux for B- and M-type $K(h)$ functions, had the previous rainfall cluster not prewetted the soil. Note that the previous rainfall affects matric potentials at 1 m depth for several hundreds of days for B and M-type conductivity functions, but only for a few months at most for AS.

The AS-type $K(h)$ function gave such rapid responses that only the second flux event at about 694 d was a result of recent pre-wetting at $t \approx 656$ d (Figs. 10c and 11c). Despite the very different matric potentials at the bottom, the cumulative bottom fluxes produced by a single rainfall cluster generated by FSB and RNA were quite similar for B and M and only somewhat larger for AS (Figs. 10 and 11).

The AS conductivity function led the soil to dry out so completely that the atmospheric matric potential during dry spells was reached at 1 m depth in a few months (Figs. 10c and 11c). This seems unrealistic, and seems to be related to the significant overestimation of the unsaturated hydraulic conductivity by AS evidenced in Fig. 4.

For comparison, the bottom matric potentials and fluxes are given for BCO as well (Fig. 12). They are very different, and given the poor suitability of BCO for dry soils and the poor fitting performance probably incorrect. The differences between the parameterizations illustrate the need to carefully consider the suitability of the parameterization for the intended purpose.

### 4.2.3. Silt loam and clay

The bottom fluxes from the clay and the silt loam soil for all combinations of parameterizations for the soil water retention and hydraulic conductivity curves were similar to those for the silt soil (Figs. 13 and 16), with two notable exceptions: for RNA, there was a much more damped response to the rainfall around $t = 656$ d for either the B or the M-type $K(h)$ function (Fig. 13c), in comparison to the rapidly increasing bottom flux in silt. In clay, there was virtually no response anymore (Fig. 16c). In general, the bottom fluxes for all parameterizations displayed comparable behavior with the exception of those with AS-type $K(h)$ functions (Figs. 13 and 16).

The behavior of the evaporative fluxes from the silt loam and the clay soil for all combinations of parameterizations for the soil water retention and hydraulic conductivity curves was essentially similar to that for the silty soil (Figs. 14 and 17). The main difference was the less gradual response of the evaporation for VGA, particularly for clay, which was, in fact, rather similar to the notoriously spiked response of the AS-type

conductivity function. The relative amounts of evaporation of the various parameterizations varied from one texture to another.

For AS in combination with the VGA retention curve, there was significant infiltration during periods of zero rainfall (Figs. 15 and 18). This numerical artefact led to erroneous simulations of the bottom flux. This is the most significant occurrence of mass balance errors that plague the simulations with AS-type $K(h)$ functions in silt loam and clay, as they did in silt. Evidently, the AS parameters for the $K(h)$ curve cause numerical problems in fine-textured soils.


### 4.3 Fits for a wide range of textures

        The fits for the clayey soils selected from the UNSODA database (Fig. S1 in the supplemental material, first panel) show that with data ranging to pF $\approx$ 4, data points in the drier region would have helped guiding the

fitting process. VGA and VGN produced good fits but struggled with high residual water contents, as did BCO. We modified FSB by requiring that the capillary bound water content goes to zero when the adsorbed water content does, a modification of the original equation by Fayer and Simmons (1995). The cut-off value of the matric potential was clearly too small for these fine-textured soils, and a unrealistic jump to zero water content occurred for the C2 and C4 soils with nrs. 1122, 1123, 1135, 1181, and 1182 in the UNSODA database. The matric potential at oven

dryness evidently needs to be extremely low for soils with high clay content.

        Rossi and Nimmo (1994) fixed the matric potential in their parameterization at which the water content became zero. The fits for the soils used in the simulations showed that fixing $h_d$ for RNA not always gave satisfactory fits in the dry range, and we therefore made $h_d$ a fitting parameter. Figure S1 (first panel) shows that this parameter may need a large lower boundary, similar to FSB: the maximum value (pF = 10) still gave poor fits for

some of the fine-textured soils (soils with nrs. 1122 and 1123, both C4 in Twarakavi et al.'s classification, which category is centered roughly around the point where sand, silt, and clay all contribute 1/3 to the total mineral soil).

        Soil 1180 (Fig. S1, first panel) had a large discrepancy between the porosity and the unsaturated water contents. The effect on the shape of FSB points to the effect of the weighting factors: the accuracy of the porosity was assumed to be higher than that of the water content measurements. Because the weighting factors of the data

points are inversely proportional to the measurement error as quantified by its estimated standard deviation, the outlier was given more weight in this case. If weighting factors are manipulated to improve the quality of the fit, the fitted parameter values can no longer be qualified as maximum likelihood estimates.

        For silty soils (Fig. S1, second panel), the fits were generally good, with some evidence that the fitted residual water contents were somewhat high for some soils (3260, 3261). The extrapolations to zero water content

by FSB and RNA appeared plausible even though they differed significantly in some cases (3251, 4450), highlighting the desirability of data points in the dry range.

        For sandy soils with some clay and or silt (A3 and A4, Fig. S1, third panel), residual water contents for BCO, VGN, and VGA were often large (1120, 1143, 2110, 1133). When the data range was limited (below pF $\approx$ 4), considerable extrapolation was required. In most cases, FSB and RNA did so better than VGN and VGA. If there is

a discrepancy between the porosity and near-saturated water contents (1121, 1143, and 2110), BCO and FSB tended to shift their saturated branches towards the porosity, because of the higher weight assigned to this data point.

For sandy soils (A1 and A2, Fig. S1, fourth panel), the fits were good if the data covered the full water content range. In all cases, VGA and VGN fitted the residual water content close to driest data point, which is very unrealistic if the dry range was not covered (1142).

The RMSE values in Tables S5-S8 reflect the observations based on the curves above. If the curves have a clear inflection point, which is the case for the sands and some of the silty soils, the van Genuchten-based curves (VGN and VGA) outperform the Brooks-Corey based curves (BCO, FSB, RNA) (Tables S6-S8). With two exceptions in clays and silty soils, VGA and VGN have very similar RMSE values. As discussed above, the upper limit of $h_d$ in the RNA parameterization was very high but still too small for clayey soils, leading to very poor

RMSE values for RNA in a few cases (Table S5).

For the fits of the four soils used for the simulation and the 21 soils, sets of three optimizations were independently run for all five parameterizations, with initial guesses that covered the full range over which the parameters were allowed to vary. In about a quarter of the cases we found no more than a single acceptable fit, and we ran these again with other sets of initial guesses (again widely different from one another) and/or expanded

parameter ranges. For only two of the 125 fitted parameter sets did this procedure not lead to convincing convergence.

In none of the cases did the three independent runs yield parameter estimates that differed by more than 10% while the sum of squares of the fits differed by less than 10%, even though in all cases the initial guesses were very different, thereby ensuring that the starting points of the different searches were located in completely different

regions of the parameter space. We take this as evidence of the absence of parameter correlations, since one would expect correlated parameters to vary over a considerable range, with the RMSE of different combinations of parameter values remaining nearly constant. We found that the fitted values obtained from the different runs were very similar, with an occasional outlier in a local minimum with a considerably larger RMSE.

In order to determine the correlation matrix of the fitted parameters correctly, a Markov Chain Monte Carlo

approach would be required for each of the 125 combinations of soils and parameterizations. Given the lack of evidence that significant correlations exist we considered this beyond the focus of and the computational resources available for this work.

Some of the data sets displayed multimodality. None of the parameterizations we tested can account for that, which is why we did not examine this further in this paper. If one wishes to reproduce this by summing several

curves of the same parameterization but with different parameter values (advocated by Durner, 1994), one needs a sigmoidal curve. If physically realistic conductivity curves near saturation are deemed desirable, VGA is the only viable parameterization for this purpose among those evaluated in this paper.

**4.4 General ramifications**


We found that 14 out of 18 parameterizations of the soil water retention curve were shown to cause non-physical hydraulic conductivities when combined with the most popular (and effective) class of soil hydraulic conductivity models. For one of these cases (VGN), Ippisch et al. (2006) demonstrated convincingly that their alternative (VGA) significantly improved the quality and numerical efficiency of soil water flow model simulations, and our simulations confirmed the profound effect of this modest modification on the model results. We hope that the general criterion we developed for verifying the physical plausibility of the near-saturated conductivity will be used in the selection of suitable soil hydraulic property parameterizations for practical applications of numerical modeling of water flow in soils, and likewise will be of help in improving existing parameterizations (as we have done in a few cases here) and developing new ones.

Replacing the residual water content in a retention curve parameterization by a logarithmic dry branch generally improved the fits in the dry range for many soils. If data in the dry range were lacking, the logarithmic extension provided a physically realistic extrapolation into the dry range, but the spread between the different fits showed the level of uncertainty in this extrapolation caused by the limited range of the data. The cut-off to zero water content of FSB could be excessive for fine-textured soils, but this is only a problem if the soil actually so far that it reaches $h_d$. For RNA, adequate fits in the dry range require that the matric potential at which the water content reaches zero is to be treated as a fitting parameter. With the added flexibility of this fourth fitting parameter, RNA emerged as a very versatile parameterization, producing mostly good fits for a wide range of textures. Nevertheless, its lack of an inflection point was occasionally a limitation.

The ability of both Burdine's (1953) and Mualem's (1976) models of the soil hydraulic conductivity function to predict independent observations of the soil hydraulic conductivity curve on the basis of soil water retention parameters fitted on water content data only is reasonably good, at least for the limited data available to test this. The conductivity model of Alexander and Skaggs (1986) overestimated the conductivity of the soils for which independent data were available. This resulted in a rapid and unrealistically strong response to changes in atmospheric forcings even at 1 m depth, as shown in our simulation study.

The simulations with different parameterizations showed that under the given boundary conditions the choice of the parameterization had a modest effect on evaporation, but strongly affected the partitioning between soil water storage and deep percolation. The uncritical use of a default soil hydraulic parameterization or selecting a parameterization solely based on the quality of the fit to soil water retention data points entails the risk of an incomplete appreciation of the potential errors of the water fluxes occurring in the modeled soil. This points to the importance of carefully considering the soil hydraulic parameterization to be used for long-term water balance studies. Such studies typically aim to determine or predict the variation of seasonal water availability to plants or long-term groundwater recharge to assess the sustainability of extractions from an underlying aquifer. If at all possible, observations during dynamic flow (water contents, matric potentials, fluxes) should be included in the parameterization selection process. In this context it would be interesting to see if parameter-estimation-processes based on inverse modeling of a non-steady unsaturated flow experiment would lead to a different choice of parameterization than fitting parameters to data points obtained at hydrostatic equilibrium. This requires the

inclusion of all the parametric expressions of interest in the numerical solvers of Richards' equation capable or running in parameter estimation mode.

## 795 Code availability

The parameter optimization code is available upon request from G.H. de Rooij. At a later time we intend to make the code available through a website.

**Appendix: List of variables**

| Variables | Dimensions | Properties, and equation to which the variable pertains (where applicable) |
|---|---|---|
| $A_1$ | $L^{-2}$ | Constant, Eq. (S3) |
| $A_2$ | - | Constant of, Eq. (S8a) |
| $B(h)$ | $L^n$ | Function simplifying notation, Eq. (S7c) |
| $b$ | - | Shape parameter, Eq. (S21) |
| $C$ | - | Constant simplifying notation, Eq. (S7c) |
| $c_1$ | - | Constant, Eq. (S10a) |
| $c_2$ | - | Constant, Eq. (S11a) |
| $\mathbf{d}_\theta$ | Varies | Vector of length $q_\theta$ of squared differences between observations and fits, Eq. (27) |
| $E(h)$ | $L^{-\kappa}$ | Function simplifying notation, Eq. (S9e) |
| $F$ | $L^\lambda$ | Constant simplifying notation, Eq. (S9e) |
| $F_R(\mathbf{x}_{p,R})$ | - | Objective function |
| $G$ | - | Constant simplifying notation, Eq. (S12c) |
| $g_0, g_1$ | - | Fitting parameter, Eq. (S15a) |
| $H$ | L | Sample height |
| $h$ | L | Matric potential |
| $h_a$ | L | Matric potential at which the soil reaches the maximum adsorbed water content |
| $h_{ae}$ | L | Air entry value of the soil |
| $h_c$ | L | Fitting parameter |
| $h_d$ | L | Pressure head at oven dryness |
| $h_i$ | L | Matric potential at the inflection point |
| $h_j$ | L | Pressure head at junction point |
| $h_m$ | - | Fitting parameter representing the matric potential at median pore size |
| $h_s$ | L | Minimum capillary height |
| $I$ | - | Constant simplifying notation, Eq. (S12c) |
| $J$ | $L^{-\lambda-\kappa}$ | Function simplifying notation, Eq (S12c) |
| $j$ | - | Counter |
| $K$ | $L\,T^{-1}$ | Unsaturated hydraulic conductivity |
| $K_s$ | $L\,T^{-1}$ | Saturated hydraulic conductivity |
| $L$ | - | Constant simplifying notation, Eq. (S14c) |

| | | |
|---|---|---|
| $M_1$ | - | Constant simplifying notation, Eq. (S14c) |
| $M_2$ | - | Constant simplifying notation, Eq. (S14g) |
| $m$ | - | Shape parameter of $\theta(h)$ |
| $n$ | - | Shape parameter of $\theta(h)$ |
| $P(h)$ | - | Function simplifying notation, Eq. (S14c) |
| $R$ | - | Iteration step |
| $R_{MAX}$ | - | Maximum number of iteration |
| $S$ | $L^3 L^{-3}$ | Variable running from 0 to $S_e$ |
| $S^{ad}$ | - | Adsorbed water, Eq. (S16) |
| $S^{cap}$ | - | Capillary water, Eq. (S16) |
| $S_e$ | - | Degree of saturation |
| T | - | Indicates that the vector is transposed |
| $w$ | - | Weighting factor ranging between 0 and 1, Eq. (S16) |
| $w_{\theta,R}$ | - | Weight factor vector |
| $w_{R,i}$ | - | Individual weighting factor in $w_{\theta,R}$ |
| $x$ | Varies | Integration variable |
| $\mathbf{x}$ | Varies | Parameter vector |
| $\mathbf{x}_f$ | Varies | Vector of non- fitted parameters |
| $\mathbf{x}_{p,R}$ | Varies | Vector of fitted parameters |
| $Y$ | - | Number of complexes |
| $\alpha$ | $L^{-1}$ | Shape parameter of $\theta(h)$ |
| $\beta$ | - | Constant |
| $\gamma$ | - | Shape parameter of $K(h)$ |
| $\zeta_1$ | - | Constant, Eq. (S10a) |
| $\zeta_2$ | - | Constant, Eq. (S11a) |
| $\eta$ | - | Fitting parameter |
| $\theta$ | $L^3 L^{-3}$ | Volumetric water content |
| $\theta_a$ | $L^3 L^{-3}$ | Curve fitting parameter representing the volumetric water content when $h = -1$cm |
| $\theta_i$ | $L^3 L^{-3}$ | $i$th observation of the volumetric water content |
| $\theta_j$ | $L^3 L^{-3}$ | Volumetric water content at junction point |
| $\theta_m$ | $L^3 L^{-3}$ | Water content at $h_m$ |
| $\theta_r$ | $L^3 L^{-3}$ | Residual water content |
| $\theta_s$ | $L^3 L^{-3}$ | Saturated water content |
| $\kappa$ | - | Shape parameter of $K(h)$ |

| | | |
|---|---|---|
| $\lambda$ | - | Fitting parameter of $\theta(h)$ |
| $\sigma$ | | Fitting parameter that characterizes the width of the pore size distribution |
| $\sigma_{h,i}$, $\sigma_{\theta,i}$ | - | Error standard deviations respectively for the $i$th matric potential and the $i$th water content |
| $\sigma^*_{h,i}$, $\sigma^*_{\theta,i}$ | - | Scaled values of $\sigma_{h,i}$, $\sigma_{\theta,i}$ |
| $\sigma^*_{i,R}$ | - | Scaled standard deviation of $(h_i, \theta_i)$ during iteration $R$ |
| $\tau$ | - | Shape parameter of $K(h)$ |

**Author contribution**


RM gathered the soil hydraulic functions from the literature. RM and GHdR carried out the parameter optimization runs with the SCE-based code. RM and GHdR designed the test problem (column size, initial and boundary conditions) for the test simulations with HYDRUS-1D. RM set up, ran, and analyzed these model simulations. GHdR wrote the shell of the optimization code, selected the UNSODA soils, and carried out the mathematical

analysis of the soil hydraulic functions. HM carried out the experiments that generated the data for the soils used in the simulations. JM wrote the SCE parameter optimization code. RM and GHdR wrote the paper. All authors were involved in checking and improving the paper.

**Competing interests**

The authors declare that they do not have a conflict of interest.

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

 **Figures:**

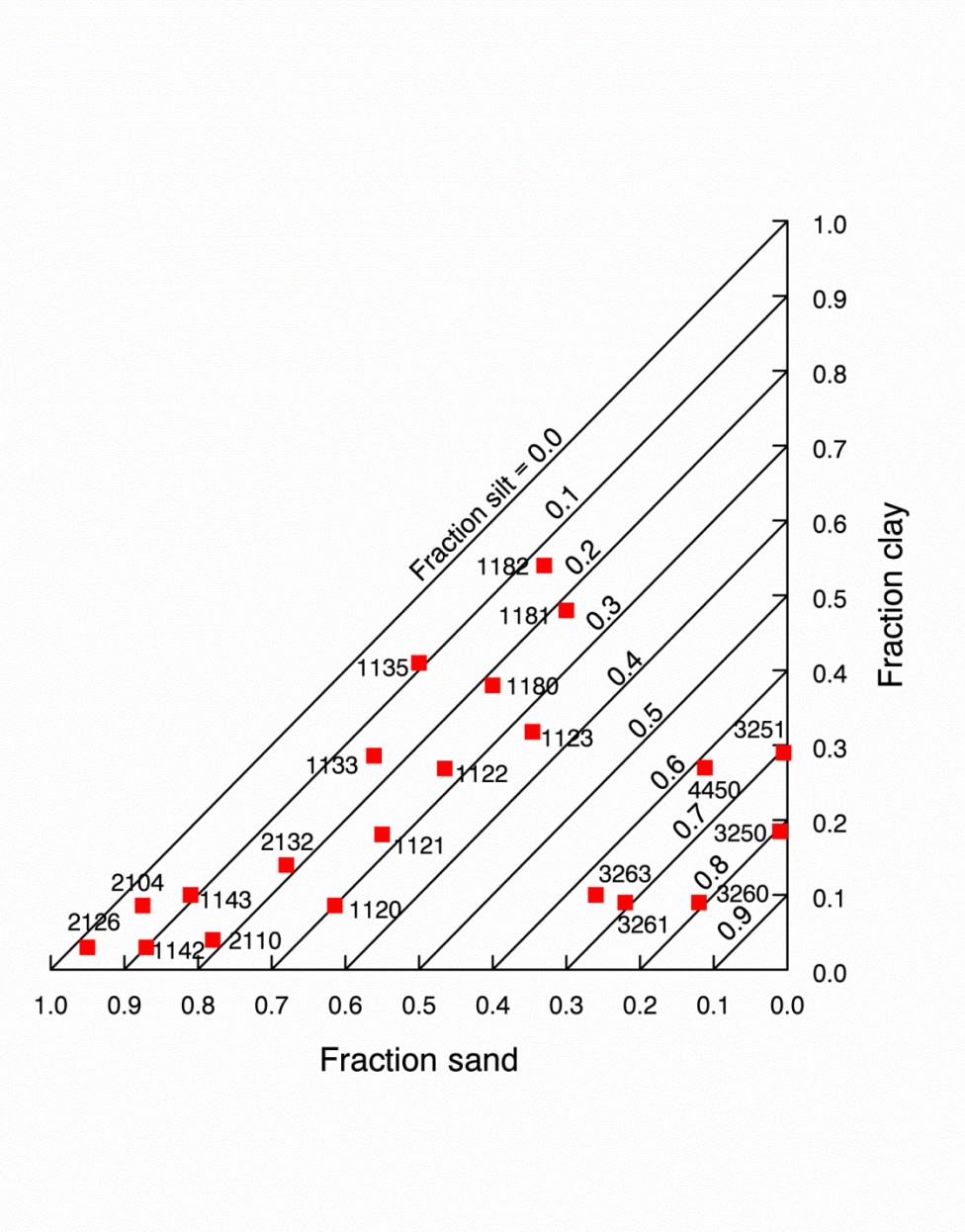

Figure 1: The textures of the soils used to test the fitting capability of selected soil water retention curve

parameterizations. The numbers next to the data points are the identifiers used in the UNSODA database to distinguish individual soils.

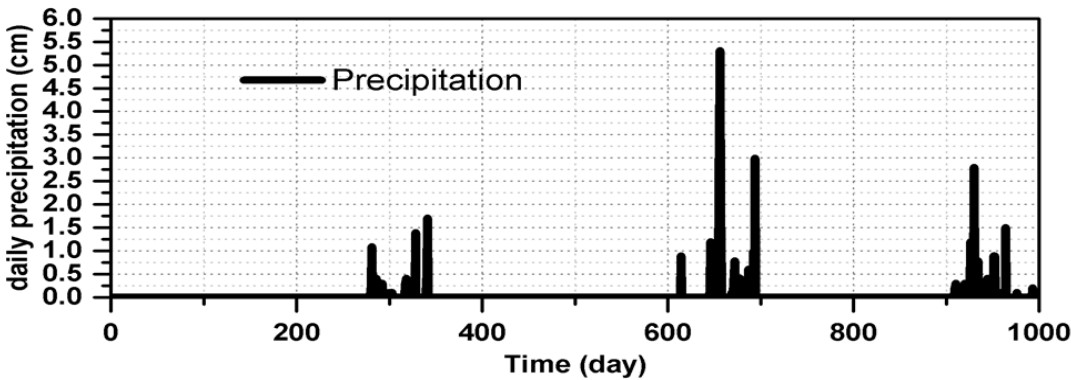

Figure 2: The record of daily rainfall sums from Riyadh city that was used in the numerical scenario study. Three rainfall clusters are visible. The largest daily rainfall amount (5.4 cm) fell on day 656. The observation period starts at June 4, 1993, and ends at February 27, 1996.



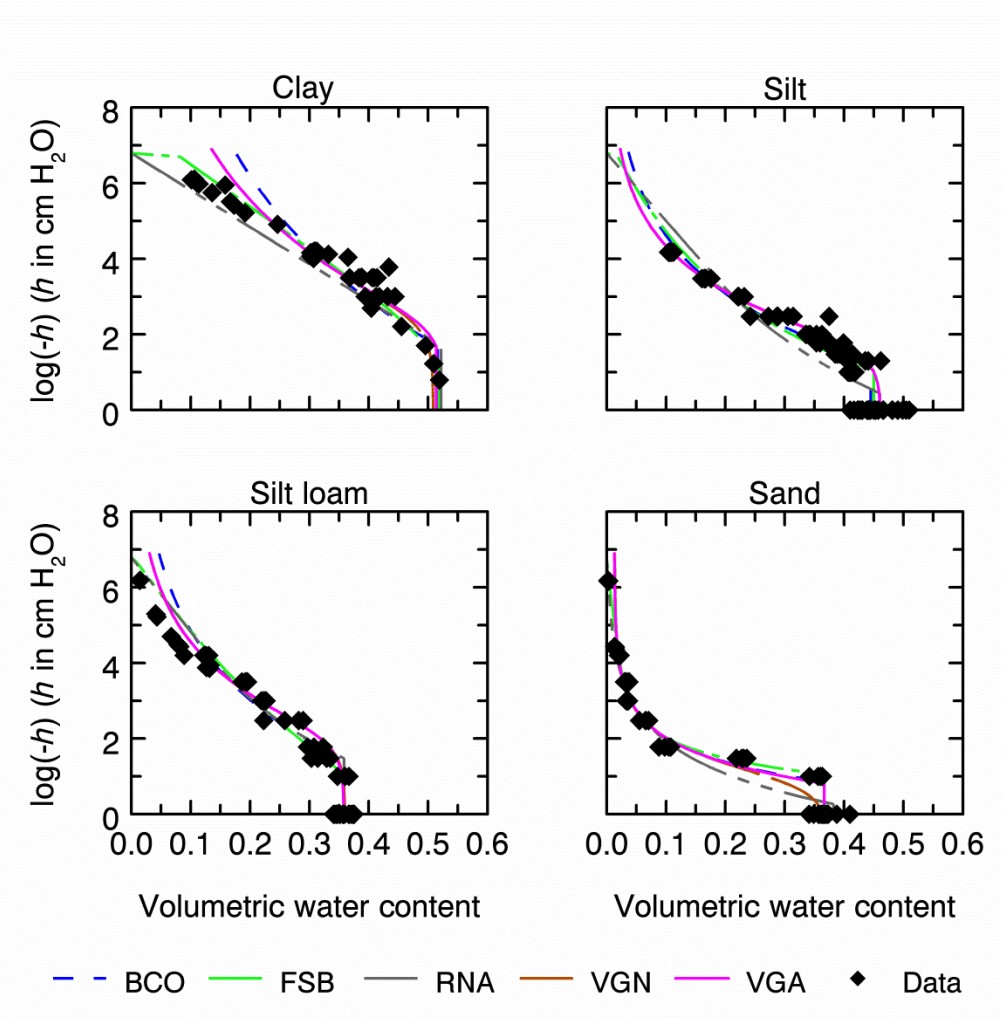


Figure 3: Observed and fitted retention curves for the different soil textures.

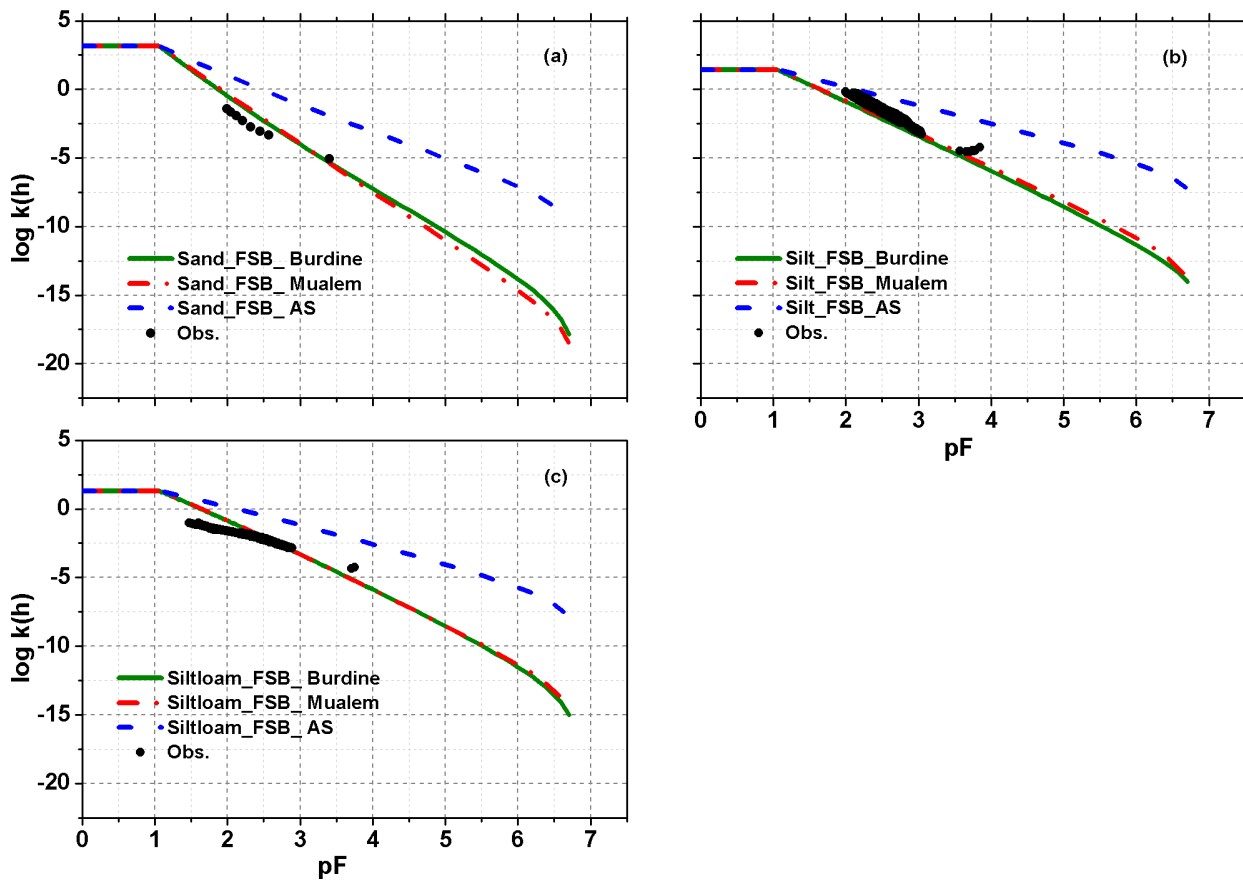


Figure 4: The observed and fitted hydraulic conductivity curve according to Burdine (1953), Mualem (1976) and Alexander and Skaggs (1986) using the fitted parameters of the Fayer and Simmons soil water retention curve (1995) for (a) sand, (b) silt, and (c) silt loam. The units of $K$ are cm d$^{-1}$.

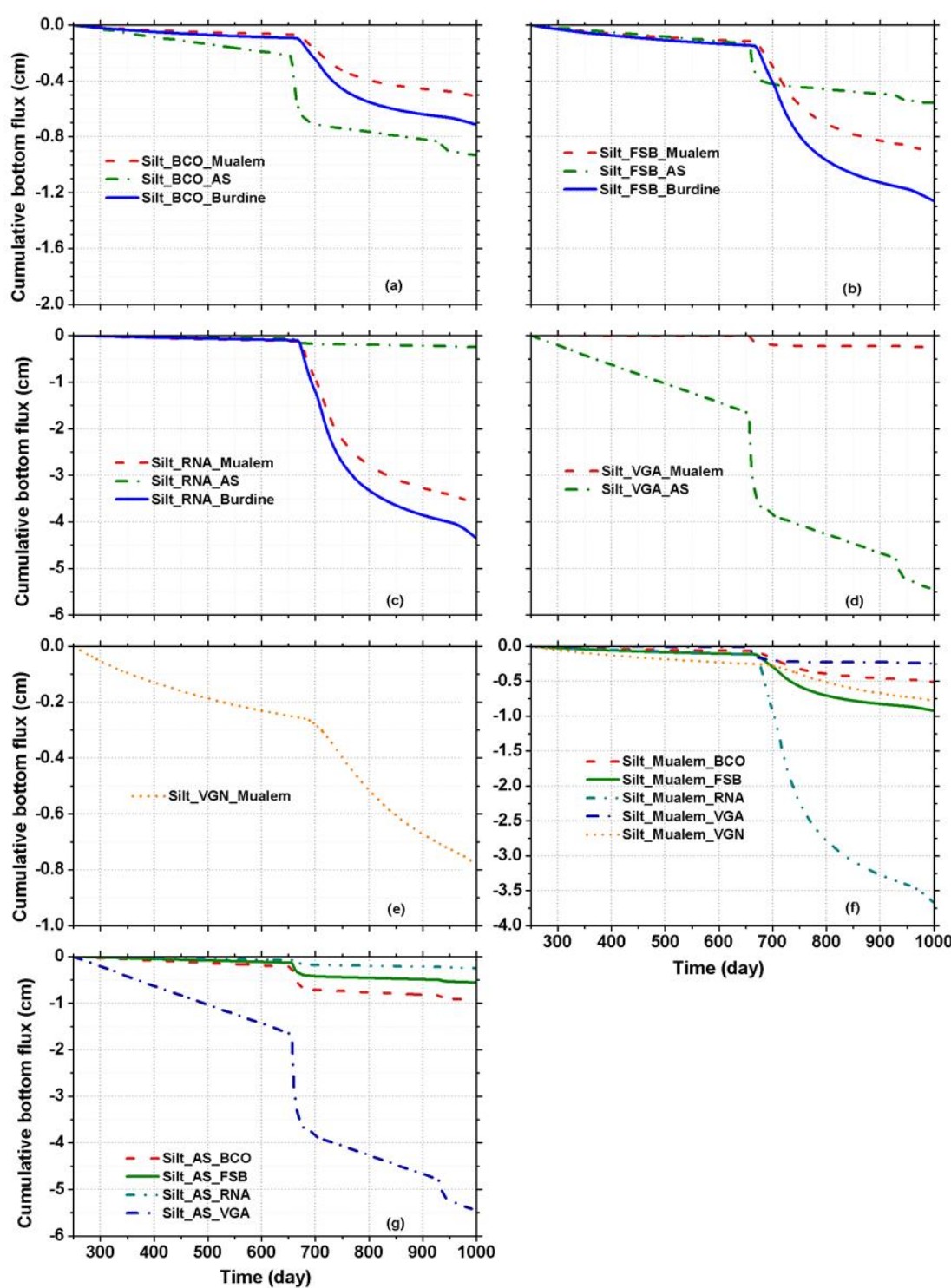

 Figure 5: The cumulative bottom fluxes leaving a silt soil column for the different combinations of soil water retention curve and hydraulic conductivity parameterizations. Panels a through e present the results for the indicated retention parameterizations (see Table 1). Panels f and g organize the results according to the conductivity function: either Mualem (1976) (f) or Alexander and Skaggs (1986) (g).

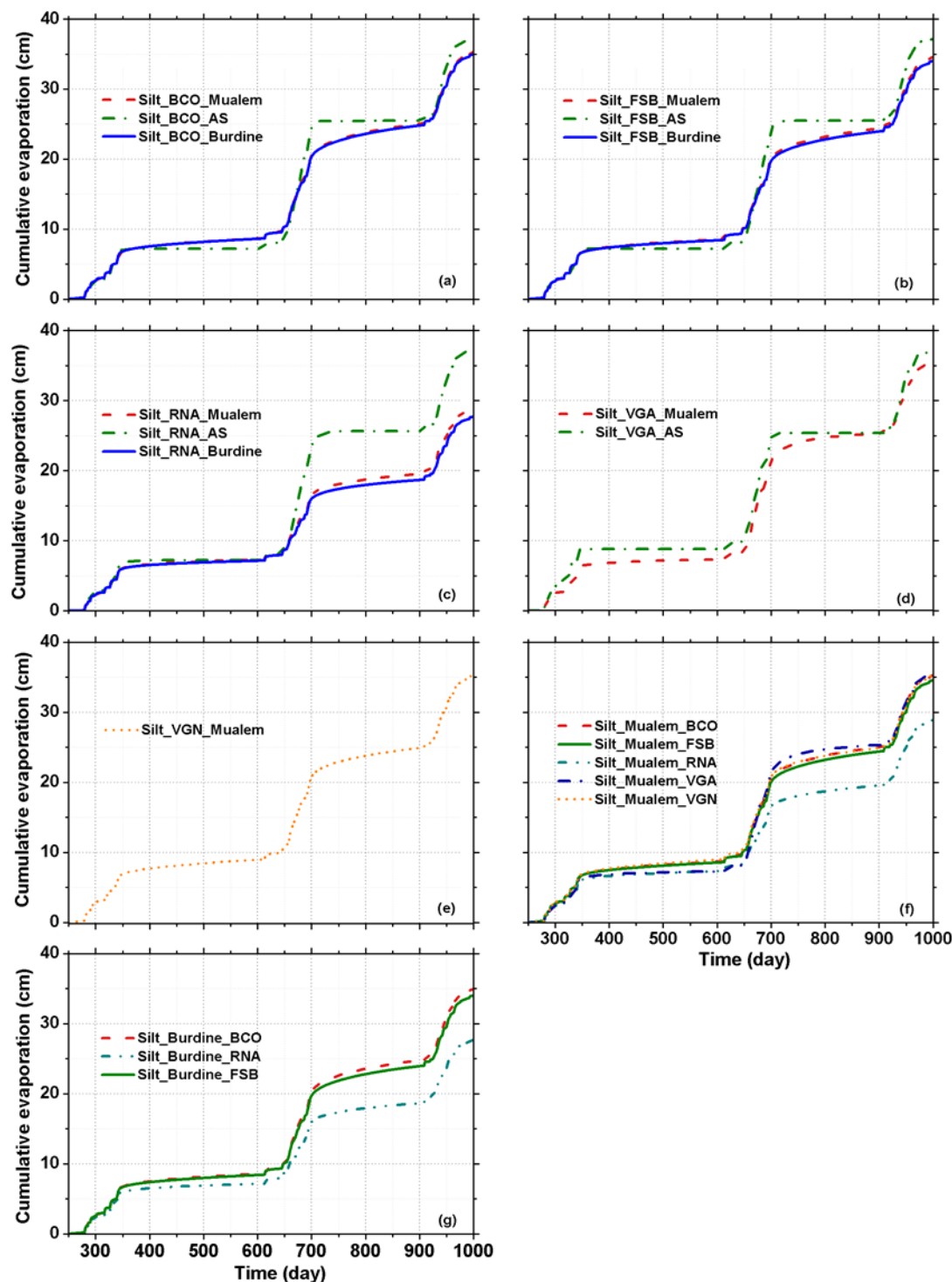

Figure 6: Cumulative evaporation from a silt soil column for the different combinations of soil water retention and hydraulic conductivity parameterizations. Panels a through e present the results for the indicated retention parameterizations (see Table 1). Panels f and g organize the results according to the conductivity function: either Mualem (1976) (f) or Burdine (1953) (g).

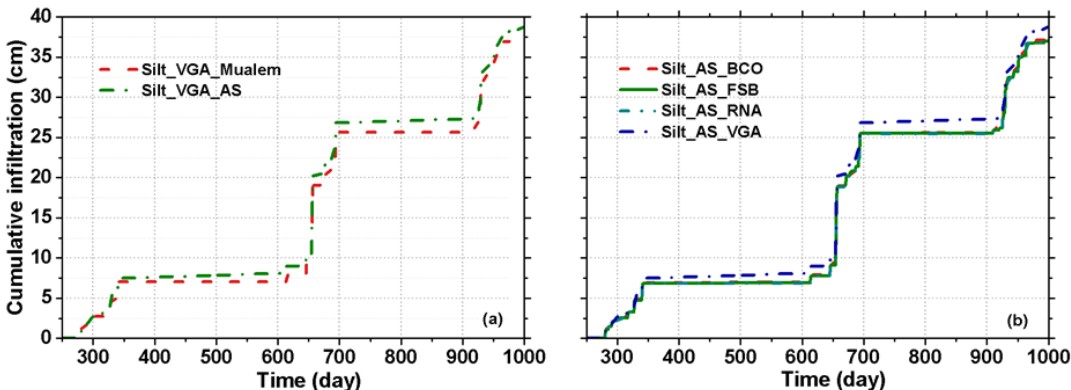


Figure 7: Cumulative infiltration in a silt profile for the VGA parameterization (see Table 1) with conductivity functions according to Mualem (1976) and Alexander and Skaggs (1986) (a) and four different parameterizations for the retention curve (see Table 1) with the Alexander and Skaggs conductivity function (b).

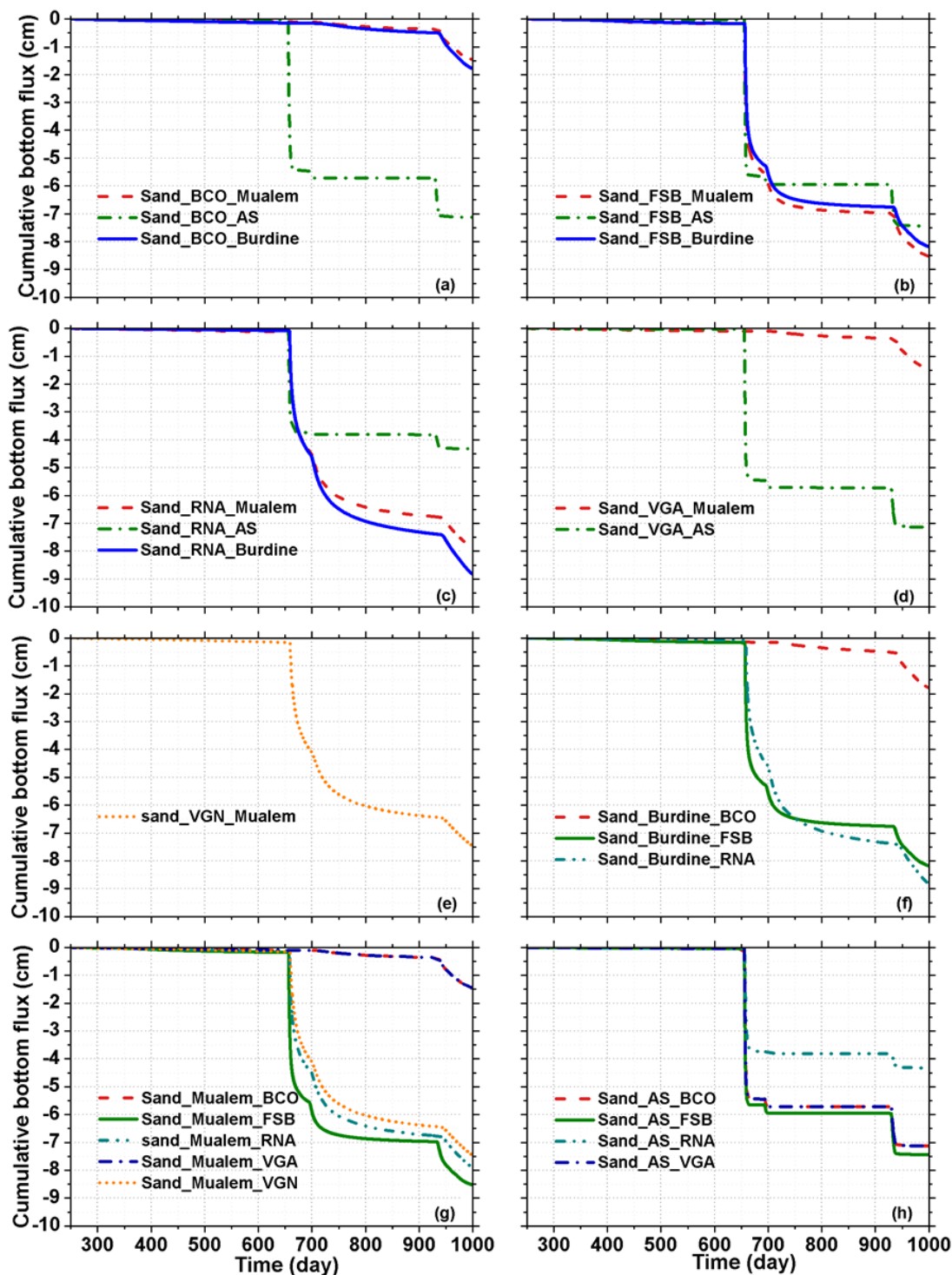

Figure 8: As Fig. 5, but for a sandy soil column. Unlike Fig. 4, the results of Burdine's (1953) conductivity curve are shown (panel f).

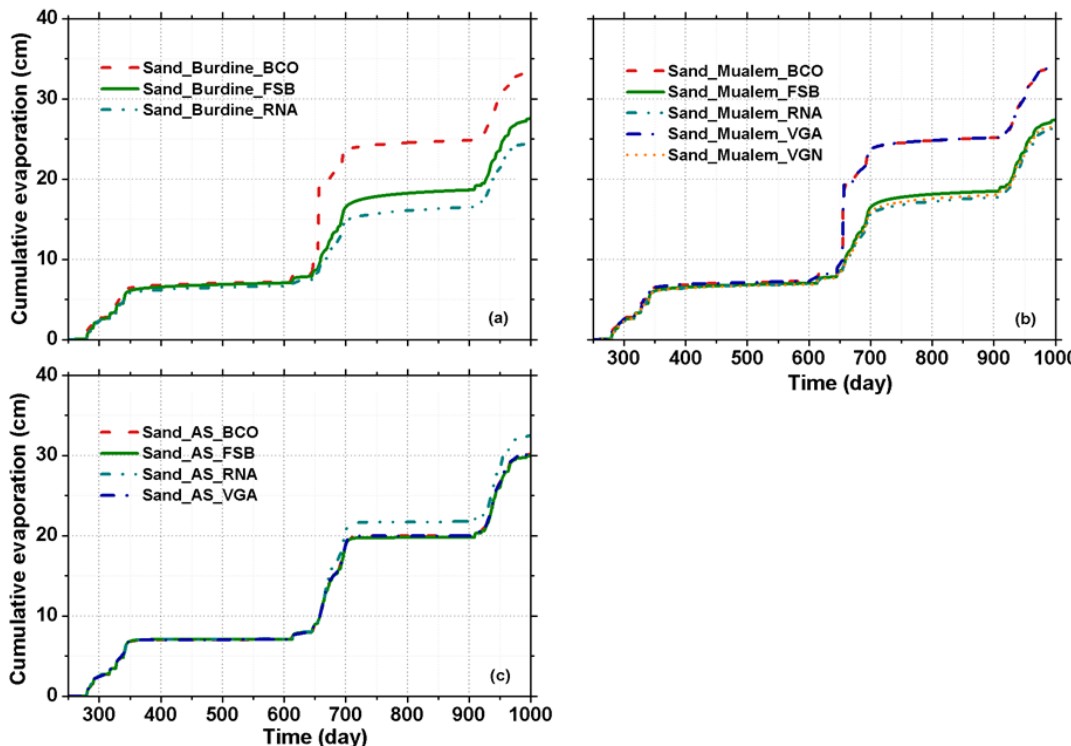

Figure 9: Cumulative evaporation from a sandy profile for the different combinations of retention curve
parameterizations (see Table 1) and hydraulic conductivity functions: Burdine (1953) (a), Mualem (1976) (b) or
Alexander and Skaggs (1986) (c).

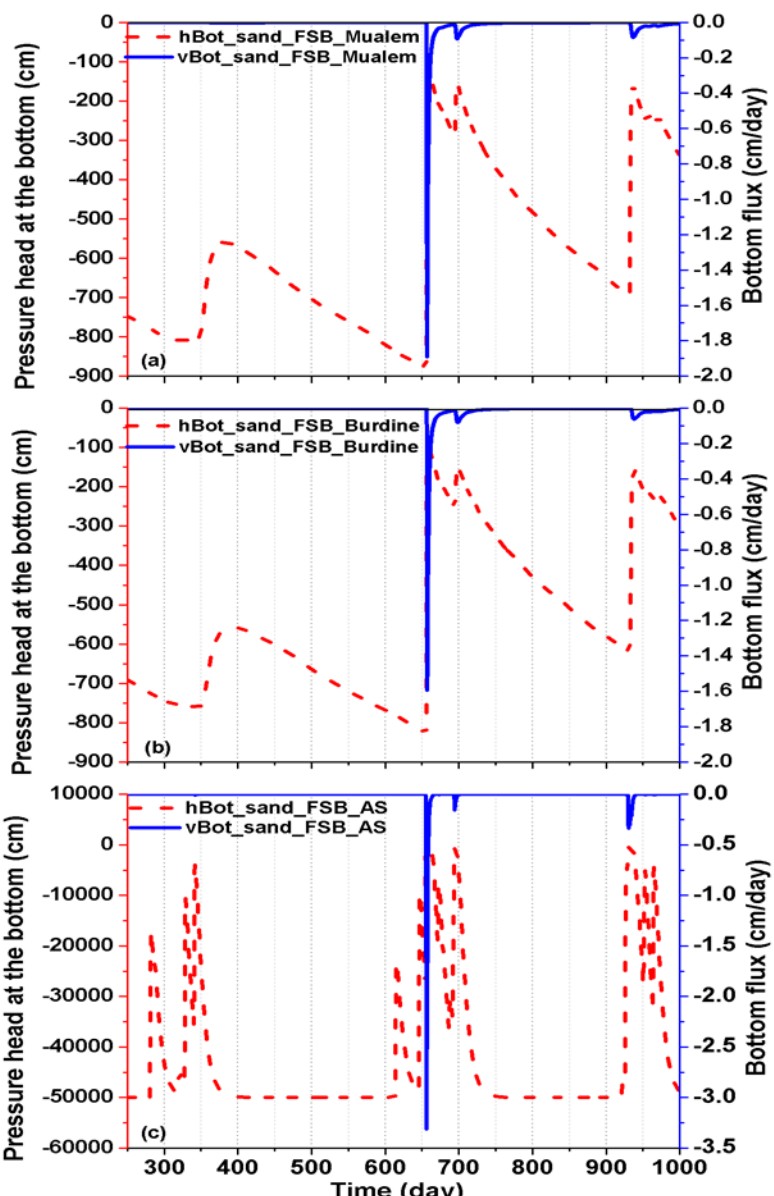

Figure 10: Pressure head hBot and flux density vBot at the bottom of the sand column for the FSB parameterization (see Table 1) and the conductivity functions of Mualem (1976) (a), Burdine (1953) (b) and Alexander and Skaggs (1986) (c).

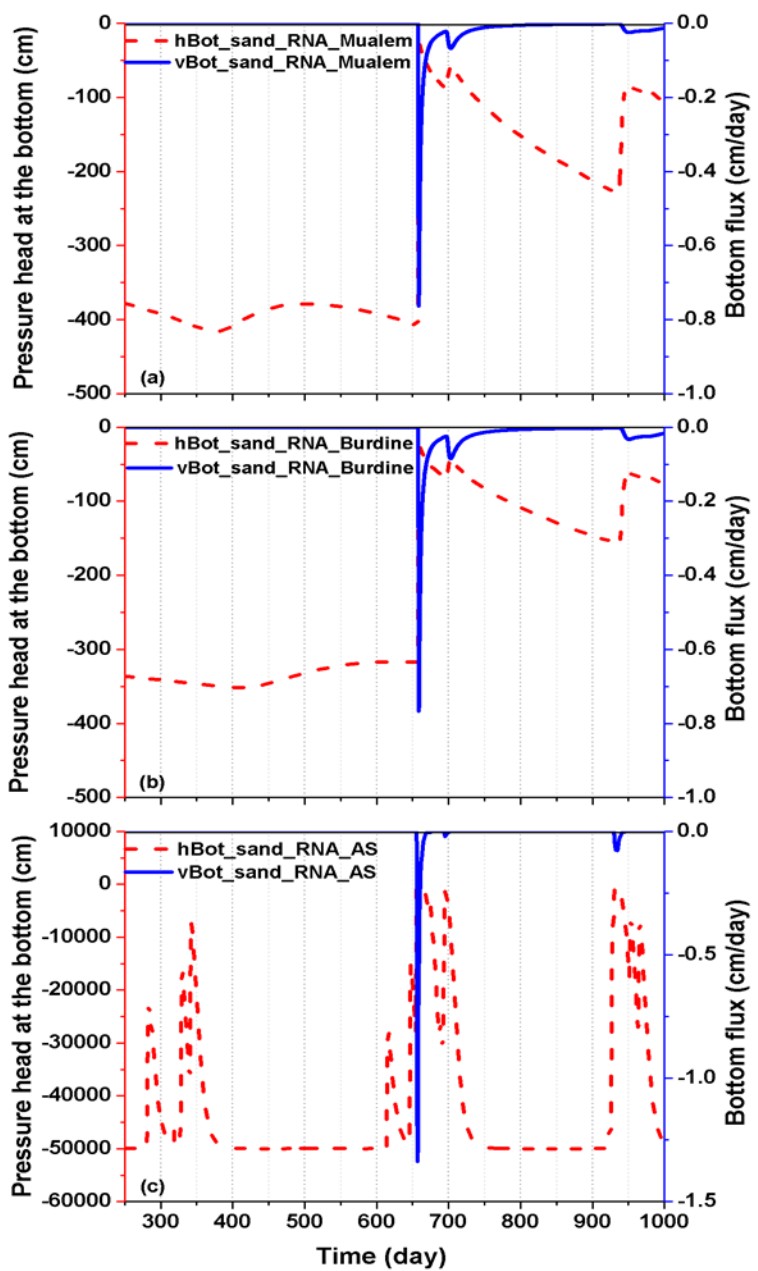

Figure 11: As Figure 10, but for the RNA parameterization (see Table 1).


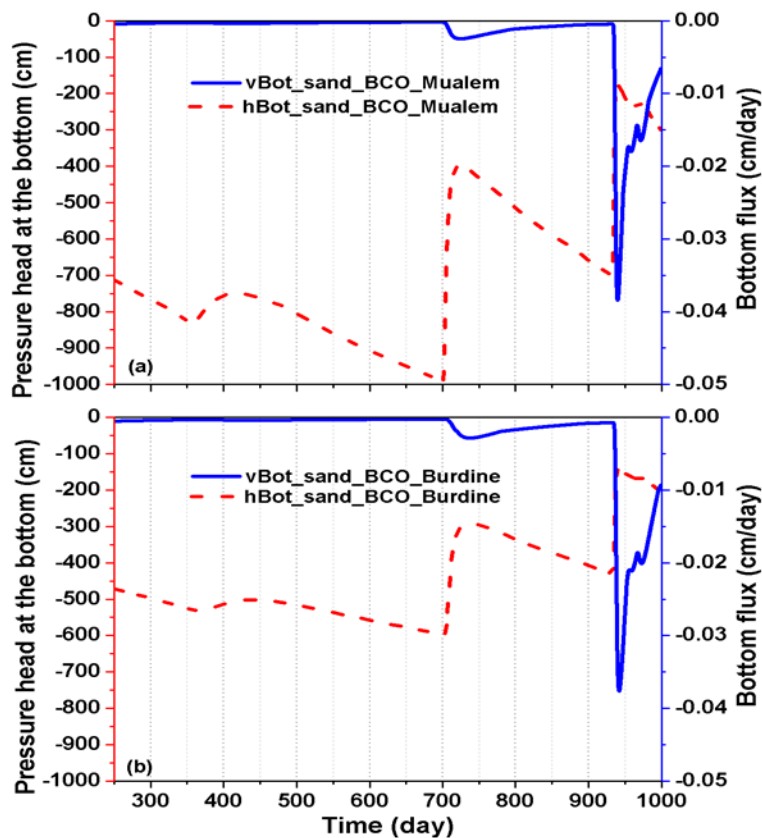

Figure 12: As Figure 10, but for the BCO parameterization (see Table 1).

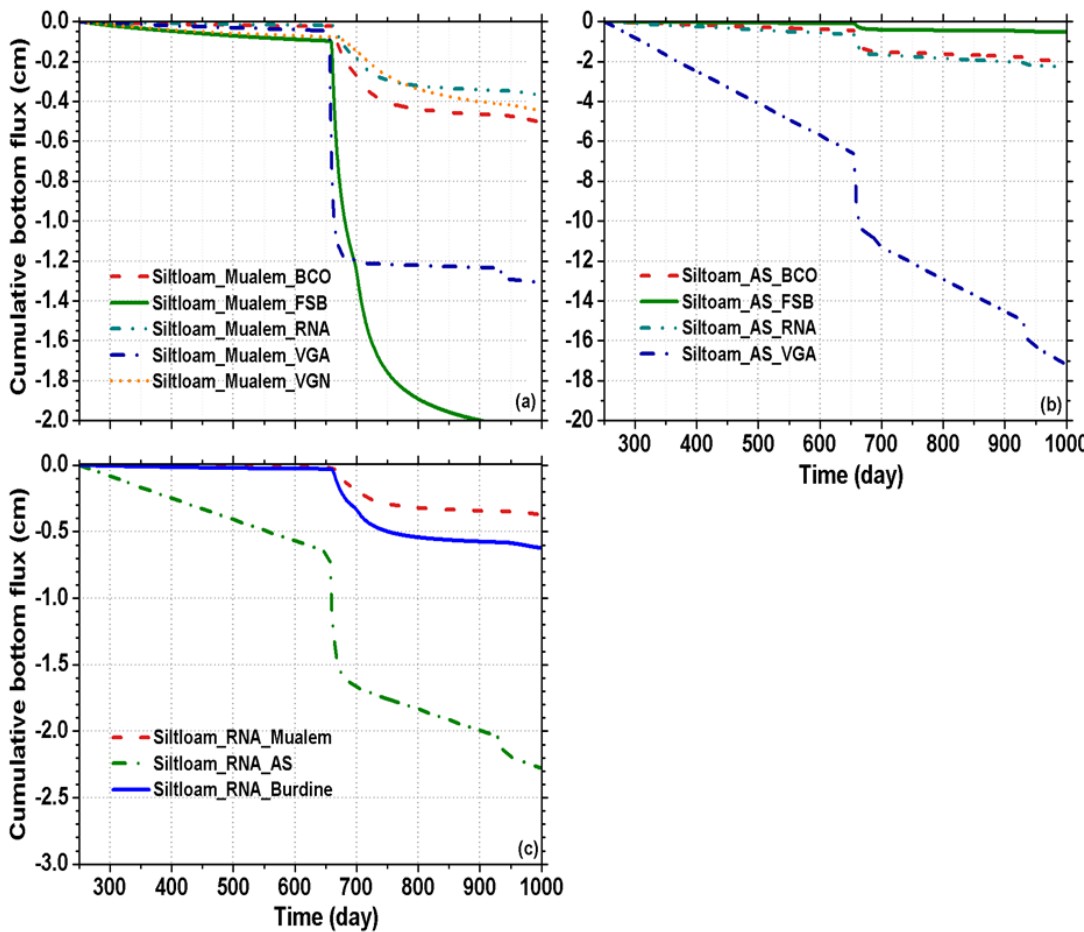

Figure 13: Cumulative bottom fluxes from a silt loam profile for all combinations of parameterizations (see Table 1) and Mualem's (1976) (a) and Alexander and Skaggs' (1986) conductivity functions (b), and for the RNA parameterization with all three conductivity functions (c).

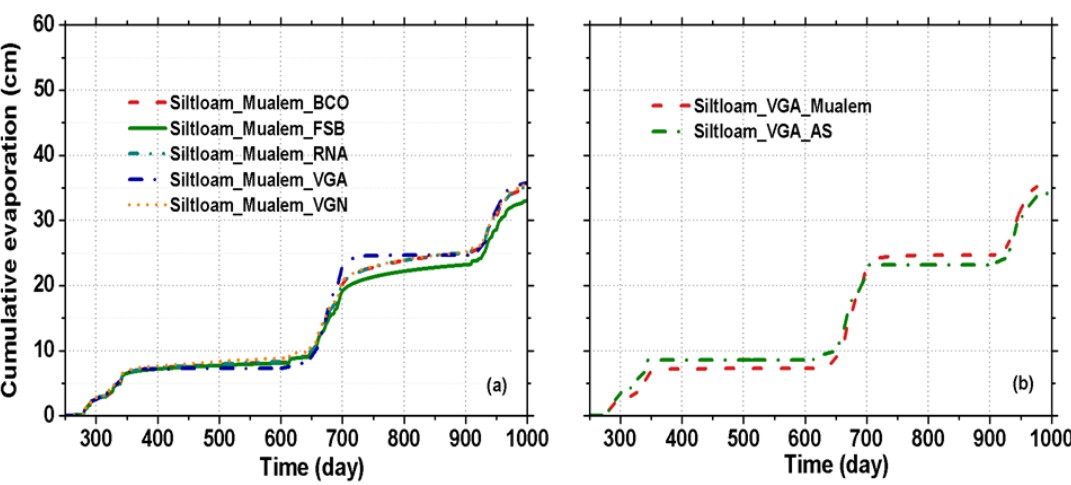


Figure 14: Cumulative evaporation from a silt loam profile for all parameterizations (see Table 1) with Mualem's (1976) conductivity function (a) and the VGA parameterization with conductivity functions according to Mualem (1976) and Alexander and Skaggs (1986) (b).




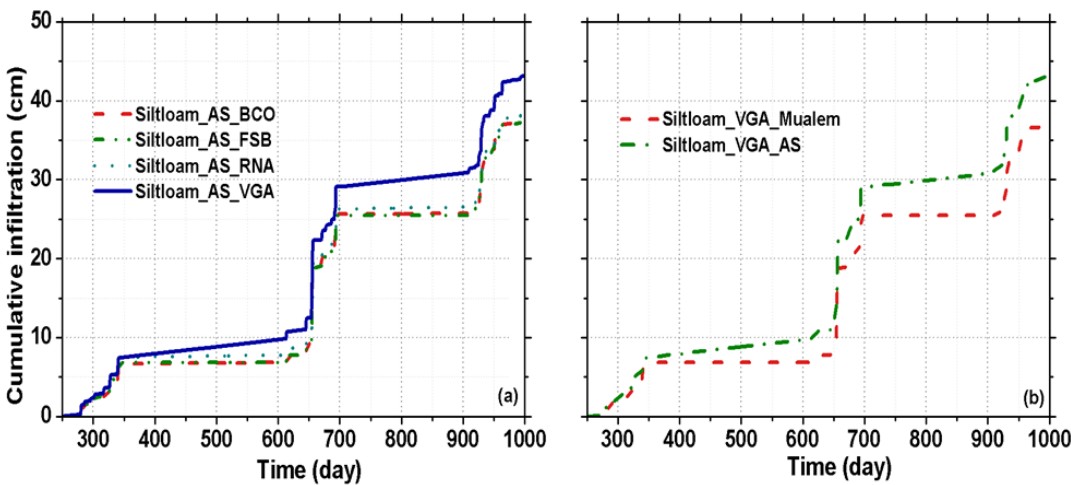

Figure 15: Cumulative infiltration from a silt loam profile for four parameterizations (see Table 1) with the Alexander and Skaggs (1986) conductivity function (a) and for the VGA parameterizations with conductivity functions according to Mualem (1976) and Alexander and Skaggs (1986) (b).


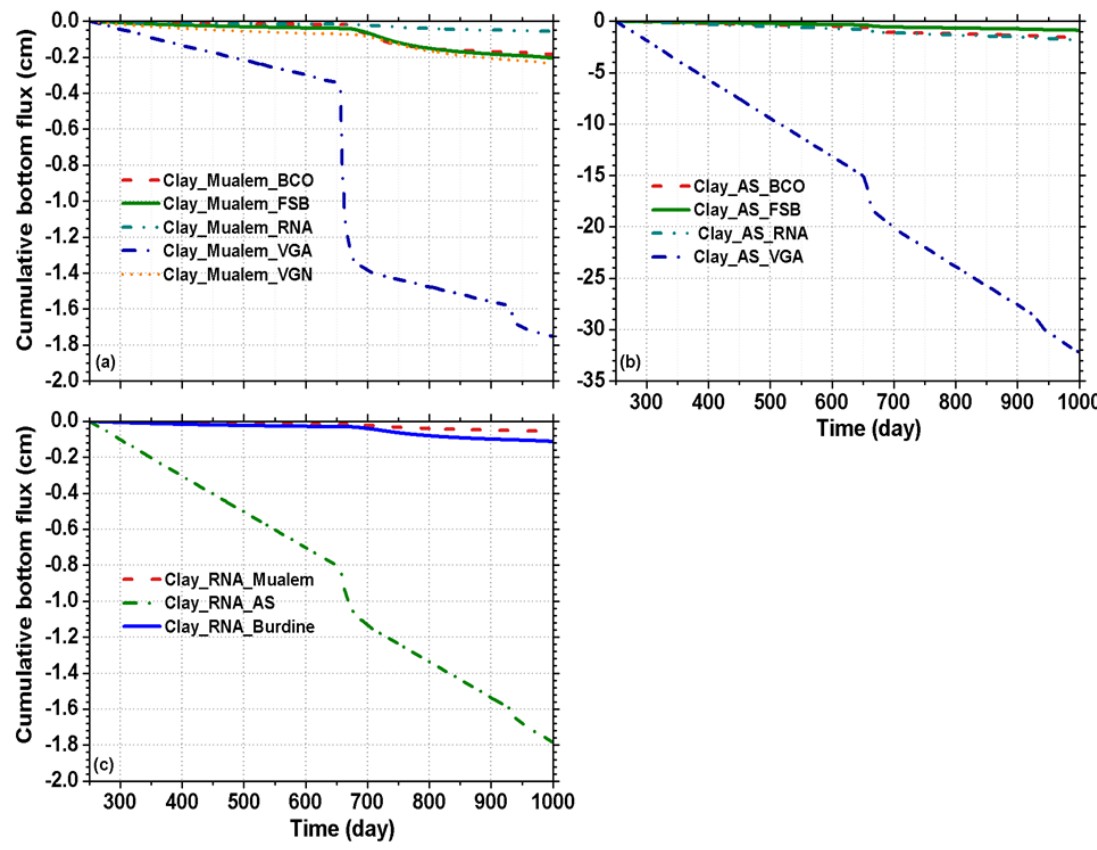

Figure 16: As Fig. 13, for clay.



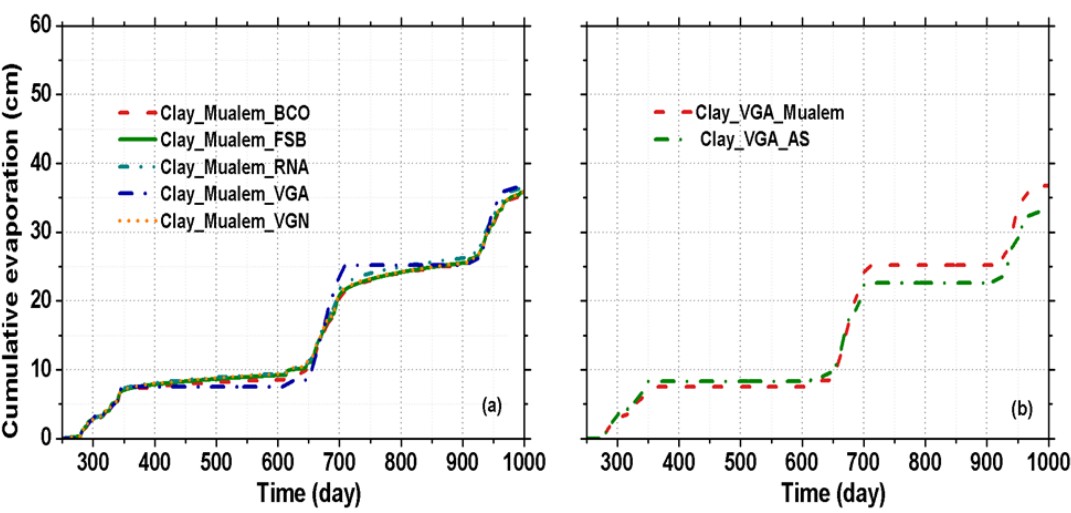

Figure 17: As Fig. 14, for clay.





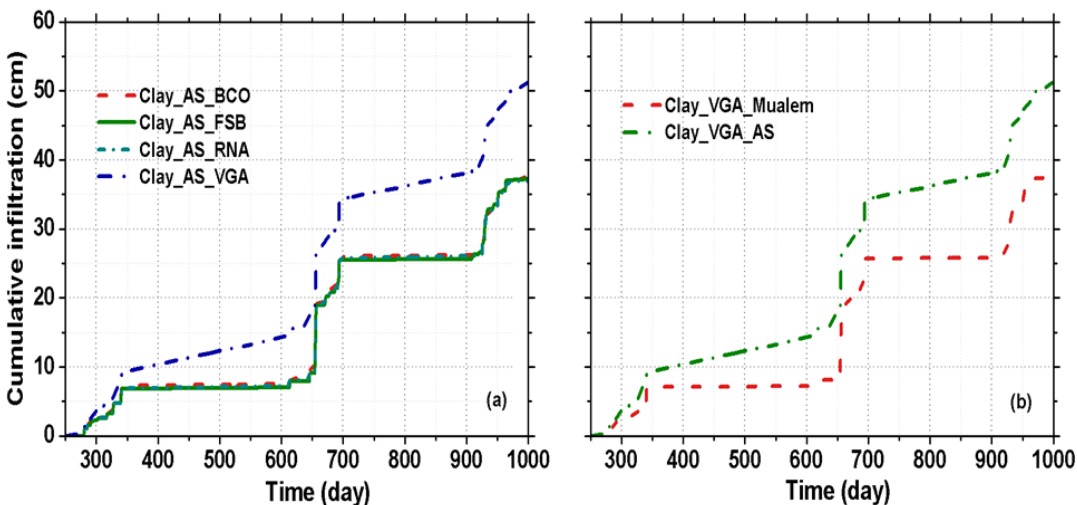

Figure 18: As Fig. 15, for clay.


**Tables:**

Table 1: The fitting parameters for five parameterizations, their physically permitted ranges, and their fitted values
for four textures. The three-character parameterization label is explained in the main text.

| Parameter-ization | Fitted parameter | Unit | Range | Texture | | | |
| --- | --- | --- | --- | --- | --- | --- | --- |
| | | | | Silt | Sand | Clay | Silt loam |
| | $\theta_r$ | - | $0 - \theta_s$ | 0.000127 | 0.013300 | 0.000004 | 0.000015 |
| BCO | $\theta_s$ | - | $\theta_r - 1$ | 0.445 | 0.366 | 0.516 | 0.358 |
| | $h_{ae}$ | cm | $-\infty - 0$ | -21.426 | -7.161 | -50.577 | -30.440 |
| | $\lambda$ | - | $0 - \infty$ | 0.197 | 0.520 | 0.091 | 0.163 |
| | $\theta_s$ | - | $\theta_a - 1$ | 0.449 | 0.366 | 0.519 | 0.358 |
| FSB | $\theta_a$ | - | $0 - \theta_s$ | 0.177 | 0.048 | 0.500 | 0.312 |
| | $h_{ae}$ | cm | $h_d - 0$ | -11.537 | -11.508 | -16.783 | -11.668 |
| | $\lambda$ | - | $0 - \infty$ | 0.254 | 0.719 | 0.152 | 0.364 |
| | $\theta_s$ | - | $0 - 1$ | 0.460 | 0.382 | 0.522 | 0.358 |
| RNA | $h_{ae}$ | cm | $h_j - 0$ | -2.826 | -1.884 | -50.856 | -30.250 |
| | $h_j$ | cm | $h_d - h_{ae}$ | -2876 | -359000 | -49.882 | -11641 |
| | $\theta_r$ | - | $0 - \theta_s$ | 0.000133 | 0.012880 | 0.000019 | 0.000041 |
| VGA | $\theta_s$ | - | $\theta_r - 1$ | 0.461 | 0.366 | 0.514 | 0.358 |
| | $\alpha$ | cm$^{-1}$ | $0 - \infty$ | 0.0197 | 0.8391 | 0.0055 | 0.0093 |
| | $n$ | - | $1 - \infty$ | 1.252 | 1.511 | 1.127 | 1.219 |
| | $h_{ae}$ | cm | $-\infty - 0$ | -0.0015 | -6.4626 | -47.2530 | -0.0081 |
| | $\theta_r$ | - | $0 - \theta_s$ | 0.000025 | 0.013560 | 0.001160 | 0.000003 |
| VGN | $\theta_s$ | - | $\theta_r - 1$ | 0.461 | 0.370 | 0.509 | 0.360 |
| | $\alpha$ | cm$^{-1}$ | $0 - \infty$ | 0.0200 | 0.1353 | 0.0042 | 0.0095 |
| | $n$ | - | $1 - \infty$ | 1.251 | 1.528 | 1.127 | 1.219 |

Table 2: Root mean square of errors (RMSE) for the different parameterizations.

| Parameterization | Texture | | | |
| | Silt | Sand | Clay | Silt loam |
|---|---|---|---|---|
| BCO | 0.1422 | 0.1164 | 0.1858 | 0.1122 |
| FSB | 0.1248 | 0.1163 | 0.1205 | 0.1068 |
| RNA | 0.0341 | 0.0130 | 0.2192 | 0.1101 |
| VGA | 0.0118 | 0.1164 | 0.1604 | 0.0412 |
| VGN | 0.0118 | 0.0111 | 0.1547 | 0.0411 |





Table 3: Cumulative bottom and evaporative fluxes (positive upwards) for silt from day 281 (the start of the first
rainfall) onwards for Burdine and Mualem conductivity functions with the different parameterizations. The
hydraulic conductivity at $h = -300$ cm (the initial condition at the bottom) is also given.

| | Cumulative bottom flux (cm) | | Cumulative evaporation (cm) | | $K(-300)$ (cm d$^{-1}$) |
|---|---|---|---|---|---|
| Parameterization | Burdine | Mualem | Burdine | Mualem | Mualem |
| BCO | -0.70 | -0.500 | 34.147 | 34.445 | 0.00080 |
| FSB | -1.240 | -0.910 | 33.219 | 33.736 | 0.00147 |
| RNA | -4.337 | -3.650 | 27.046 | 28.184 | 0.00702 |
| VGA | - | -0.248 | - | 34.956 | 0.00014 |
| VGN | - | -0.744 | - | 34.359 | 0.00119 |

**Supplemental material**

**S1. Soil water retention and hydraulic conductivity functions**

This section reviews the most popular parameterizations of the soil water retention curve and several lesser-known others that were developed to improve the fit in the dry range or at least eliminate the need for the physically poorly defined residual water content. At this time, we consider unimodal functions only. The physical plausibility in terms of the rate of change near saturation of the corresponding conductivity models is verified, thereby maintaining the consistency between the retention and the conductivity curves that would have been lost in Iden et al.'s (2015) approach. In all cases but one, this physical plausibility is checked for the first time. The plausibility check requires that the derivative of each retention curve is determined and the criterion in Eq. (4) of the is used to define the permissible range for $\kappa$. If this range does not include any of the values $\{1, 2\}$ used by the conductivity models described above, or if the permitted values are non-physical ($< 0$), the retention model does not have a conductivity model associated with it, which limits its practical value. As above, $h$ denotes the matric potential, which is negative in unsaturated soils. Many of the cited papers adopt this notation for its opposite, the suction.

The water retention function of Brooks and Corey (1964) is

$$\theta(h) = \begin{cases} \theta_r + \left(\theta_s - \theta_r\right)\left(\dfrac{h}{h_{ae}}\right)^{-\lambda}, & h \leq h_{ae} \\ \theta_s, & h > h_{ae} \end{cases}$$

(S1a)

This equation is referred to as BCO below. The derivative is

$$\frac{\mathrm{d}\theta}{\mathrm{d}h} = \begin{cases} \dfrac{-\lambda\left(\theta_s - \theta_r\right)}{h_{ae}}\left(\dfrac{h}{h_{ae}}\right)^{-\lambda-1}, & h \leq h_{ae} \\ 0, & h > h_{ae} \end{cases}$$

(S1b)

where $\lambda$ is a dimensionless fitting parameter. If $\theta_r$ is set to zero, Campbell's (1974) equation is obtained.

The analytical expression for the generalized $K(h)$ function (Eq. (3)) for the water retention function of Brooks and Corey (1964) is

$$K(h) = \begin{cases} K_s \left( \dfrac{h(S_e)}{h_{ae}} \right)^{-\lambda\tau} \left\{ \dfrac{\left[ \dfrac{\lambda(\theta_s - \theta_r)|h_{ae}|^\lambda}{\kappa + \lambda + 2} |h|^{-\kappa-\lambda} \right]_{-\infty}^{h}}{\left[ \dfrac{\lambda(\theta_s - \theta_r)|h_{ae}|^\lambda}{\kappa + \lambda + 2} |h|^{-\kappa-\lambda} \right]_{-\infty}^{h_{ae}}} \right\}^\gamma = K_s \left( \dfrac{h_{ae}}{h} \right)^{\lambda(\gamma+\tau)+\gamma\kappa}, & h \le h_{ae} \\[2em] K_s, & h > h_{ae} \end{cases}$$

(S1c)

Note that the Brooks-Corey retention curve allows all three parameters of the associated conductivity model to be fitted.

The derivative of the Brooks-Corey function is discontinuous at $h_{ae}$. Hutson and Cass (1987) added a parabolic approximation at the wet end to make the first derivative continuous. For $\theta_r = 0$, they proposed

$$\theta(h) = \begin{cases} \theta_s \left( \dfrac{h}{h_{ae}} \right)^{-\lambda}, & h \le h_i \\[2em] \theta_s \left[ 1 - \left( \dfrac{h}{h_{ae}} \right)^2 \dfrac{\left( 1 - \dfrac{2}{\lambda+2} \right)}{\left( \dfrac{2}{\lambda+2} \right)^{-\frac{2}{\lambda}}} \right], & 0 \ge h > h_i \end{cases}$$

(S2a)

where $h_i$ [L] is the matric potential at the inflection point, given by:

$$h_i = h_{ae} \left( \dfrac{2}{2-\lambda} \right)^{\frac{1}{\lambda}}.$$

(S2b)

The derivative is

$$\dfrac{d\theta}{dh} = \begin{cases} -\dfrac{\lambda\theta_s}{h_{ae}} \left( \dfrac{h}{h_{ae}} \right)^{-\lambda-1}, & h \le h_i \\[2em] \dfrac{2\theta_s}{h_{ae}} \dfrac{\left( \dfrac{2}{\lambda+2} - 1 \right)}{\left( \dfrac{2}{\lambda+2} \right)^{-\frac{2}{\lambda}}} \left( \dfrac{h}{h_{ae}} \right), & 0 \ge h > h_i \end{cases}.$$

(S2c)

The parameter $h_{ae}$ no longer is an air-entry value and should be considered a pure fitting parameter. It should be noted that the smooth transition to saturation that this function and several others mimic may at least in part be caused by the non-zero height of the soil cores used in experiments to determine soil water retention curves. At hydrostatic equilibrium, the matric potential along the vertical varies in the soil core, resulting in a differentiable shape of the apparent soil water retention curve, even if the soil in the core has a uniform air-entry value that leads to a locally non-differentiable curve (Liu and Dane, 1995).

The parabolic approximation of Hutson and Cass (1987) leads to the following expression for the term in Eq. (4)

$$\lim_{h \to 0} A_1 |h|^{1-\kappa} = 0$$

(S3)

where $A_1$ is a constant. This leads to the requirement that $\kappa < 1$, ruling out the usual models. Although the parabolic approximation in itself does not preclude the existence of a closed-form expression for $K$, the restriction on $\kappa$ is quite severe, so we do not pursue this further.

Van Genuchten's (1980) formulation is also continuously differentiable:

$$\theta(h) = \theta_r + (\theta_s - \theta_r)(1 + |\alpha h|^n)^{-m}, \quad h \leq 0$$

(S4a)

where $\alpha$ [L$^{-1}$], $n$, and $m$ are shape parameters. This equation is denoted by VGN below. It has the derivative

$$\frac{d\theta}{dh} = \alpha m n (\theta_s - \theta_r)|\alpha h|^{n-1}(1 + |\alpha h|^n)^{-m-1}, \quad h \leq 0$$

(S4b)

where often $m$ is set equal to $1 - 1/n$.

The limit of the derivative of van Genuchten's (1980) retention curve near saturation is

$$\frac{d\theta}{dh}\bigg|_{h=0} = \alpha^n m n (\theta_s - \theta_r)|h|^{n-1}$$

(S5)

leading to the requirement that $\kappa < n-1$. For many fine and/or poorly sorted soil textures, $n$ ranges between 1 and 2. Therefore, this restriction can be even more severe than the one required for a parabolic wet end, even excluding Mualem's (1976) conductivity model when $n < 2$. For this reason we refrain from formulating analytical conductivity equations, even though van Genuchten (1980) presented such expressions for Burdine's (1953) and Mualem's (1976) models.

Vogel et al. (2001) presented a modification to improve the description of the hydraulic conductivity near saturation without being aware of the physical explanation of the poor behavior presented later by Ippisch et al. (2006). Their retention function reads

$$\theta(h) = \begin{cases} \theta_r + (\theta_m - \theta_r)\left(1 + |\alpha h|^n\right)^{-m}, & h < h_s \\ \theta_s, & h \geq h_s \end{cases}$$

(S6a)

where $h_s$ [L] is a fitting parameter close to zero with which $\theta_m$ can be defined as

$$\theta_m = \theta_r + (\theta_s - \theta_r)\left(1 + |\alpha h_s|^n\right)^{-m}$$

(S6b)

The derivative is

$$\frac{d\theta}{dh} = \begin{cases} \alpha mn(\theta_m - \theta_r)|\alpha h|^{n-1}\left(1 + |\alpha h|^n\right)^{-m-1}, & h < h_s \\ 0, & h \geq h_s \end{cases}$$

(S6c)

Schaap and van Genuchten (2006) reported a value of $h_s$ of –4 cm to work best for a wide range of soils to improve the description of the near-saturated hydraulic conductivity. The parameter $h_s$ should therefore not be viewed as an air-entry value.

Although an expression can be derived for $K(h)$ if $\kappa$ is set to 1 and $m = 1 - 1/n$, we prefer to adopt the formulation by Ippisch et al. (2006), given its solid physical footing. They proposed to introduce an air-entry value and scale the unsaturated portion of the retention curve by its value at the water-entry value:

$$\theta(h) = \begin{cases} \theta_r + (\theta_s - \theta_r)\left(\dfrac{1 + |\alpha h|^n}{1 + |\alpha h_{ae}|^n}\right)^{-m}, & h < h_{ae} \\ \theta_s, & h \geq h_{ae} \end{cases}$$

(S7a)

with derivative

$$\frac{d\theta}{dh} = \begin{cases} \alpha mn(\theta_s - \theta_r)|\alpha h|^{n-1}\left(1 + |\alpha h_{ae}|^n\right)^{m}\left(1 + |\alpha h|^n\right)^{-m-1}, & h < h_{ae} \\ 0, & h \geq h_{ae} \end{cases}$$

(S7b)

With the common restriction of $m = 1 - 1/n$, an expression can be found for $\kappa = 1$ that is slightly more general than Eq. (11) in Ippisch et al. (2006):

$$K(h) = \begin{cases} K_s \left( \dfrac{\theta - \theta_r}{\theta_s - \theta_r} \right)^{\tau} \left[ \dfrac{1 - \left( 1 - \dfrac{1}{B(h)} \right)^{\frac{n}{n-1}}}{1 - \left( 1 - \dfrac{1}{C} \right)^{\frac{n}{n-1}}} \right]^{\gamma} \\ \\ = K_s \left( \dfrac{B(h)}{C} \right)^{\tau\left(\frac{1}{n}-1\right)} \left[ \dfrac{1 - \left( 1 - \dfrac{1}{B(h)} \right)^{\frac{n}{n-1}}}{1 - \left( 1 - \dfrac{1}{C} \right)^{\frac{n}{n-1}}} \right]^{\gamma}, \quad h < h_{ae} \\ \\ K_s, \hspace{6cm} h \geq h_{ae} \end{cases}$$

(S7c)

where

$$B(h) = 1 + |\alpha h|^n$$

(S7d)

$$C = 1 + |\alpha h_{ae}|^n$$

(S7e)

This equation can be used to define conductivity models according to Mualem (1976) and Alexander and Skaggs (1986), which both require that $\kappa = 1$.

None of the retention models discussed so far performs very well in the dry range. Campbell and Shiozawa (1992) introduced a logarithmic section in the dry end to improve the fit in the dry range:

$$\theta(h) = \theta_a \left( 1 - \frac{\ln|h|}{\ln|h_d|} \right) + A_2 \left( \frac{1}{1 + |\alpha h|^4} \right)^m$$

(S8a)

with derivative

$$\frac{d\theta}{dh} = \frac{\theta_a}{\ln|h_d|} \frac{1}{h} + 4\alpha m A_2 |\alpha h|^3 \left( 1 + |\alpha h|^4 \right)^{-m-1}$$

(S8b)

where $\theta_a$ represents the maximum amount of adsorbed water, $A_2$ is a constant and $h_d$ is the matric potential at oven-dryness, below which the water content is assumed to be zero. The first term in the derivative leads to the requirement that $\kappa < -1$, and therefore no conductivity model can be derived from Eq. (S8a).

Rossi and Nimmo (1994) also preferred a logarithmic function over the Brooks-Corey power law at the dry end to better represent the adsorption processes that dominates water retention in dry soils, as opposed to capillary processes in wetter soils. They also implemented a parabolic shape at the wet end as proposed by Hutson and Cass (1987). Rossi and Nimmo (1994) presented two retention models, but only one (the junction model) permitted an analytical expression of the unsaturated hydraulic conductivity. Here, the junction model is presented with and without the parabolic expression for the wet end of the retention curve. With the discontinuous derivative at the air-entry value, the expression reads

$$\theta(h) = \begin{cases} 0, & h \le h_d \\ \theta_s \beta \ln\left(\dfrac{h_d}{h}\right), & h_d < h \le h_j \\ \theta_s \left(\dfrac{h_{ae}}{h}\right)^{\lambda}, & h_j < h \le h_{ae} \\ \theta_s, & h > h_{ae} \end{cases}, \qquad (S9a)$$

which is denoted RNA below. The derivative is

$$\frac{\mathrm{d}\theta}{\mathrm{d}h} = \begin{cases} 0, & h \le h_d \\ \dfrac{\theta_s \beta}{-h}, & h_d < h \le h_j \\ \lambda \theta_s |h_{ae}|^{\lambda} |h|^{-\lambda-1}, & h_j < h \le h_{ae} \\ 0, & h > h_{ae} \end{cases} \qquad (S9b)$$

Rossi and Nimmo (1994) required the power law and logarithmic branches as well as their first derivatives to be equal at the junction point $(\theta_j, h_j)$. With $h_d$ fixed (Rossi and Nimmo found a value of $-10^5$ m for six out of seven soils and $-5 \cdot 10^5$ m for the seventh), these constraints allow two of the five remaining free parameters to be expressed in terms of the other three. Some manipulation leads to the expressions:

$$\lambda = \frac{1}{\ln|h_d| - \ln|h_j|} \qquad (S9c)$$

$$\beta = \lambda \left(\frac{h_{ae}}{h_j}\right)^{\lambda} \qquad (S9d)$$

but other choices are possible. This choice leads to fitting parameters $h_{ae}$, $h_j$, and $\theta_s$. The associated conductivity model is

$$K(h) = \begin{cases} 0, & h \leq h_d \\[2em] K_s S_e^\tau \left\{ \dfrac{\left[ -\dfrac{\theta_s \beta}{\kappa} |h|^{-\kappa} \right]_{h_d}^{h}}{\left[ -\dfrac{\theta_s \beta}{\kappa} |h|^{-\kappa} \right]_{h_d}^{h_j} - \left[ \dfrac{\theta_s \lambda}{\lambda + \kappa} |h_{ae}|^{\lambda} |h|^{-(\lambda+\kappa)} \right]_{h_j}^{h_{ae}}} \right\}^{\gamma} \\[2em] = K_s \left[ \beta \ln\left( \dfrac{h_d}{h} \right) \right]^{\tau} \left[ \dfrac{E(h)}{E(h_j) + F\left( |h_j|^{-\lambda-\kappa} - |h_{ae}|^{-\lambda-\kappa} \right)} \right]^{\gamma}, & h_d < h \leq h_j \\[2em] K_s S_e^\tau \left\{ \dfrac{\left[ -\dfrac{\theta_s \beta}{\kappa} |h|^{-\kappa} \right]_{h_d}^{h_j} - \left[ \dfrac{\theta_s \lambda}{\lambda + \kappa} |h_{ae}|^{\lambda} |h|^{-(\lambda+\kappa)} \right]_{h_j}^{h}}{\left[ -\dfrac{\theta_s \beta}{\kappa} |h|^{-\kappa} \right]_{h_d}^{h_j} - \left[ \dfrac{\theta_s \lambda}{\lambda + \kappa} |h_{ae}|^{\lambda} |h|^{-(\lambda+\kappa)} \right]_{h_j}^{h_{ae}}} \right\}^{\gamma} \\[2em] = K_s \left( \dfrac{h_{ae}}{h} \right)^{\lambda\tau} \left[ \dfrac{E(h_j) + F\left( |h_j|^{-\lambda-\kappa} - |h|^{-\lambda-\kappa} \right)}{E(h_j) + F\left( |h_j|^{-\lambda-\kappa} - |h_{ae}|^{-\lambda-\kappa} \right)} \right]^{\gamma}, & h_j < h \leq h_{ae} \\[2em] K_s, & h > h_{ae} \end{cases}$$

(S9e)

where

$$E(h) = \frac{\beta}{\kappa} \left( |h_d|^{-\kappa} - |h|^{-\kappa} \right)$$

(S9f)

$$F = \frac{\lambda}{\lambda + \kappa} |h_{ae}|^{\lambda}$$

(S9g)

It is worth noting that recent studies that considered the conductivity of water films in relatively dry soils show a reduction in the rate at which the log($K$) dropped with increasing log(-$h$). This implies that requiring continuity of the first derivative at the junction where $h = h_j$ could be too strict (e.g. Tuller and Or (2001) and Assouline and Or (2013)).

The junction model of Rossi and Nimmo (1994) with a continuous first-order derivative achieved through the correction by Hutson and Cass (1987) reads

$$\theta(h) = \begin{cases} 0, & h \le h_d \\ \theta_s \zeta_1 \ln\left(\dfrac{h_d}{h}\right), & h_d < h \le h_j \\ \theta_s \left(\dfrac{h_{ae}}{h}\right)^\lambda, & h_j < h \le h_i \\ \theta_s \left[1 - c_1\left(\dfrac{h}{h_c}\right)^2\right], & h_i \le h \le 0 \end{cases}$$

(S10a)

with the derivative

$$\frac{d\theta}{dh} = \begin{cases} 0, & h \le h_d \\ \dfrac{\theta_s \zeta_1}{-h}, & h_d < h \le h_j \\ \lambda \theta_s |h_{ae}|^\lambda |h|^{-\lambda-1}, & h_j < h \le h_i \\ \dfrac{-2c_1\theta_s}{h_c^2} h, & h_i \le h \le 0 \end{cases}$$

(S10b)

where

$$h_i = h_c \left(\frac{\lambda}{2} + 1\right)^{\frac{1}{\lambda}}$$

(S10c)

$$h_j = h_d e^{-\frac{1}{\lambda}}$$

(S10d)

$$c_1 = \frac{\lambda}{2}\left(\frac{2}{\lambda+2}\right)^{\frac{\lambda+2}{\lambda}}$$

(S10e)

$$\zeta_1 = e\lambda \left(\frac{h_c}{h_d}\right)^\lambda$$

(S10f)

where $h_c$ [L] is a fitting parameter, together with $\lambda$ and $\theta_s$. The parabolic wet end restricts $\kappa$ to values between 0 and 1. For this reason, an expression for the conductivity curve is not derived.

Rossi and Nimmo (1994) also introduced an equation that summed up the power law and logarithmic contributions (the sum model):

$$\theta(h) = \begin{cases} 0, & h \leq h_d \\ \theta_s\left[\left(\dfrac{h_c}{h}\right)^\lambda - \left(\dfrac{h_c}{h_d}\right)^\lambda + \zeta_2 \ln\left(\dfrac{h_d}{h}\right)\right], & h_d \leq h \leq h_i \\ \theta_s\left[1 - c_2\left(\dfrac{h}{h_c}\right)^2\right], & h_i \leq h \leq 0 \end{cases}$$

(S11a)

with derivative

$$\frac{\mathrm{d}\theta}{\mathrm{d}h} = \begin{cases} 0, & h \leq h_d \\ -\dfrac{\theta_s}{h}\left[\lambda\left(\dfrac{h_c}{h}\right)^\lambda + \zeta_2\right], & h_d \leq h \leq h_i \\ \dfrac{-2c_2\theta_s}{h_c^2}h, & h_i \leq h \leq 0 \end{cases}$$

(S11b)

in which we have

$$\zeta_2 = \left[1 - \left(\frac{\lambda}{2}+1\right)\left(\frac{h_c}{h_i}\right)^\lambda + \left(\frac{h_c}{h_d}\right)^\lambda\right]\left[\frac{1}{2}+\ln\left(\frac{h_d}{h_i}\right)\right]^{-1}$$

(S11c)

and

$$c_2 = \left(\frac{h_c}{h_i}\right)^2\left[1 - \left(\frac{h_c}{h_i}\right)^\lambda + \left(\frac{h_c}{h_d}\right)^\lambda - \frac{1 - \left(\dfrac{\lambda}{2}+1\right)\left(\dfrac{h_c}{h_i}\right)^\lambda + \left(\dfrac{h_c}{h_d}\right)^\lambda}{\dfrac{1}{2\ln\left(\dfrac{h_d}{h_i}\right)}+1}\right]$$

(S11d)

A closed-form expression for the hydraulic conductivity does not exist for this function, and the permitted values for $\kappa$ are not physically acceptable.

Fayer and Simmons (1995) used the approach of Campbell and Shiozawa (1992) to have separate terms for adsorbed and capillary bound water. If the capillary binding is represented by a Brooks-Corey type function, the retention model becomes

$$\theta(h) = \begin{cases} 0, & h \le h_d \\ \theta_a\left(1 - \dfrac{\ln|h|}{\ln|h_d|}\right) + \left[\theta_s - \theta_a\left(1 - \dfrac{\ln|h|}{\ln|h_d|}\right)\right]\left(\dfrac{h_{ae}}{h}\right)^\lambda, & h_d < h < h_{ae} \\ \theta_s, & h \ge h_{ae} \end{cases} \quad \text{(S12a)}$$

This expression is denoted FSB below. The derivative is

$$\frac{\mathrm{d}\theta}{\mathrm{d}h} = \begin{cases} 0, & h \le h_d \\ \dfrac{1}{|h|}\left(\dfrac{h_{ae}}{h}\right)^\lambda\left[\lambda(\theta_s - \theta_a) + \theta_a\left(\dfrac{\ln|h|}{\ln|h_d|} - \dfrac{1}{\ln|h_d|}\right)\right], & h_d < h < h_{ae} \\ 0, & h \ge h_{ae} \end{cases} \quad \text{(S12b)}$$

The corresponding conductivity model is

$$K(h) = \begin{cases} 0, & h \le h_d \\ K_s S_e^\tau \left\{ \dfrac{\left[\dfrac{|h_{ae}|^\lambda}{\ln|h_d|(\lambda+\kappa)}\left[\theta_a\left(\dfrac{\lambda+\kappa-1}{\lambda+\kappa} - \ln|h|\right) - \lambda(\theta_s-\theta_a)\ln|h_d|\right]|h|^{-\lambda-\kappa}\right]_{h_d}^h}{\left[\dfrac{|h_{ae}|^\lambda}{\ln|h_d|(\lambda+\kappa)}\left[\theta_a\left(\dfrac{\lambda+\kappa-1}{\lambda+\kappa} - \ln|h|\right) - \lambda(\theta_s-\theta_a)\ln|h_d|\right]|h|^{-\lambda-\kappa}\right]_{h_d}^{h_{ae}}} \right\}^\gamma \\ = K_s\left\{\dfrac{\theta_a}{\theta_s}\left(1 - \dfrac{\ln|h|}{\ln|h_d|}\right) + \left[1 - \dfrac{\theta_a}{\theta_s}\left(1 - \dfrac{\ln|h|}{\ln|h_d|}\right)\right]\left(\dfrac{h_{ae}}{h}\right)^\lambda\right\}^\tau \\ \qquad \left\{\dfrac{\left[\theta_a(G - \ln|h|) - I\right]h^{-\lambda-\kappa} - J}{\left[\theta_a(G - \ln|h_{ae}|) - I\right]h_{ae}^{-\lambda-\kappa} - J}\right\}^\gamma, \quad h_d < h \le h_{ae} \\ K_s, \qquad h \ge h_{ae} \end{cases} \quad \text{(S12c)}$$

where

$$G = \frac{\lambda + \kappa - 1}{\lambda + \kappa} \quad \text{(S12d)}$$

$$I = \lambda(\theta_s - \theta_a)\ln|h_d|$$

(S12e)

$$J = \left[\theta_a\left(G - \ln|h_d|\right) - I\right]|h_d|^{-\lambda-\kappa}$$

(S12f)

Note that the above model is valid if $h_{ae}$ does not exceed -1 cm. This condition will usually be met, unless the soil texture is very coarse.

If capillary binding is described by a van Genuchten function, the resulting equation is

$$\theta(h) = \begin{cases} 0, & h \le h_d \\ \theta_a\left[1 - \dfrac{\ln|h|}{\ln|h_d|}\right] + \left\{\theta_s - \theta_a\left[1 - \dfrac{\ln|h|}{\ln|h_d|}\right]\right\}\left[1 + (-\alpha h)^n\right]^{\frac{1}{n}-1}, & h_d < h < 0 \end{cases}$$

(S13a)

with derivative

$$\frac{d\theta}{dh} = \begin{cases} 0, & h \le h_d \\ \dfrac{\theta_a}{h\ln|h_d|}\left\{\left[1 + (-\alpha h)^n\right]^{\frac{1}{n}-1} - 1\right\} \\ \quad + \alpha(1-n)(-\alpha h)^{n-1}\left[1 + (-\alpha h)^n\right]^{\frac{1}{n}-2}\left\{\theta_a\left[1 - \dfrac{\ln|h|}{\ln|h_d|}\right] - \theta_s\right\}, & h_d < h < 0 \end{cases}$$

(S13b)

The derivative has several terms that pose severe restrictions on the value of $\kappa$ (the first term even requires that $\kappa$ < -1), and other terms that limit the permitted values of $n$. The conductivity function is therefore omitted here.

In the original equations of both versions as presented by Fayer and Simmons (1995), the adsorbed water content reached zero at $h_d$, while there is still some capillary bound water at and below that matric potential, which is inconsistent. Furthermore, the terms with ratios of logarithms become negative for matric potentials below $h_d$. We therefore modified the original equations by setting the water content to zero below $h_d$.

Zhang (2011) presented a logarithmic extension of van Genuchten's (1980) model (Eq. (S4a)) in the dry end that is very similar to Eq. (S13a). The associated hydraulic conductivity model was the sum of Mualem's (1976) model and an expression for film flow conductivity. This approach only alleviated the issue of the residual water content but had the same problems near saturation as Eq. (S4a), and will therefore not be analyzed further.

Kosugi (1996) and Kosugi (1999) presented a soil water retention curve for soils with a lognormal pore size distribution. Khlosi et al. (2008) extended the approach of Campbell and Schiozawa (1992) and Fayer and Simmons (1995) to Kosugi's (1996, 1999) model. We again set the water content to zero for matric potentials below $h_d$:

$$\theta(h) = \begin{cases} 0, & h \leq h_d \\ \theta_a\left(1 - \dfrac{\ln|h|}{\ln|h_d|}\right) + \left[\theta_s - \theta_a\left(1 - \dfrac{\ln|h|}{\ln|h_d|}\right)\right]\dfrac{1}{2}\,\mathrm{erfc}\left[\dfrac{\ln\left(\dfrac{h}{h_m}\right)}{\sigma\sqrt{2}}\right], & h_d < h < 0 \end{cases}$$

(S14a)

with the derivative (see Olver et al., 2010, p. 163 and p. 443)

$$\frac{\mathrm{d}\theta}{\mathrm{d}h} = \begin{cases} 0, & h \leq h_d \\ \dfrac{\theta_a}{h\ln|h_d|}\left\{\dfrac{1}{2}\,\mathrm{erfc}\left[\dfrac{\ln\left(\dfrac{h}{h_m}\right)}{\sigma\sqrt{2}}\right] - 1\right\} + \dfrac{\theta_a\left(1 - \dfrac{\ln|h|}{\ln|h_d|}\right) - \theta_s}{h\sigma\sqrt{2\pi}}\exp\left\{-\left[\dfrac{\ln\left(\dfrac{h}{h_m}\right)}{\sigma\sqrt{2}}\right]^2\right\}, & h_d < h < 0 \end{cases}$$

(S14b)

Parameter $h_m$ [L] represents the matric potential corresponding to the median pore size, and $\sigma$ characterizes the width of the pore size distribution. The behavior of the derivative near saturation is not readily clear. Expressions for the corresponding hydraulic conductivity function can only be found for integer values of $\kappa$. For $\kappa = 1$, the expression

for the hydraulic conductivity is

$$K(h) = \begin{cases} 0, & h \le h_d \\[2ex] K_s S_e^\tau \left\{ \dfrac{\left[ \dfrac{\theta_a}{2\theta_s |h_m| \ln|h_d|} \left\{ \begin{array}{l} e^{L^2} \operatorname{erf}(P(h)+L) + \dfrac{h_m}{h}[\operatorname{erfc}(P(h))-2] - \\[1ex] M_1 e^{L^2} \operatorname{erf}(P(h)+L) - \dfrac{2Lh_m}{h\sqrt{\pi}} e^{-P^2(h)} \end{array} \right\} \right]_{h_d}^{h} }{ \left[ \dfrac{\theta_a}{2\theta_s |h_m| \ln|h_d|} \left\{ \begin{array}{l} e^{L^2} \operatorname{erf}(P(h)+L) + \dfrac{h_m}{h}[\operatorname{erfc}(P(h))-2] - \\[1ex] M_1 e^{L^2} \operatorname{erf}(P(h)+L) - \dfrac{2Lh_m}{h\sqrt{\pi}} e^{-P^2(h)} \end{array} \right\} \right]_{h_d}^{0} } \right\}^\gamma \\[4ex] = K_s S_e^\tau \left\{ \dfrac{\begin{array}{l} e^{L^2} [\operatorname{erf}(P(h)+L) - \operatorname{erf}(P(h_d)+L)] + \\[1ex] h_m \left[ \dfrac{\operatorname{erfc}(P(h))-2}{h} - \dfrac{\operatorname{erfc}(P(h_d))-2}{h_d} \right] - \\[1ex] M_1 e^{L^2} [\operatorname{erf}(P(h)+L) - \operatorname{erf}(P(h_d)+L)] - \\[1ex] \dfrac{2Lh_m}{\sqrt{\pi}} \left( \dfrac{e^{-P^2(h)}}{h} - \dfrac{e^{-P^2(h_d)}}{h_d} \right) \end{array}}{\begin{array}{l} -e^{L^2}[1+\operatorname{erf}(P(h_d)+L)] + \\[1ex] h_m \left[ \dfrac{\operatorname{erfc}(P(0))-2}{0} - \dfrac{\operatorname{erfc}(P(h_d))-2}{h_d} \right] + \\[1ex] M_1 e^{L^2}[1+\operatorname{erf}(P(h_d)+L)] - \dfrac{2Lh_m}{\sqrt{\pi}} \left( \dfrac{e^{-P^2(0)}}{0} - \dfrac{e^{-P^2(h_d)}}{h_d} \right) \end{array}} \right\}^\gamma, & h_d < h \le 0 \end{cases}$$

(S14c)

where $S_e$ is obtained by dividing Eq. (S14a) by $\theta_s$. The following functions and derived variables have been used for clarity:

$$L = \frac{\sigma\sqrt{2}}{2}$$

(S14d)

$$P(h) = \frac{\ln\left(\dfrac{h}{h_m}\right)}{\sigma\sqrt{2}}$$

(S14e)

$$M_1 = \left(1 - \frac{\theta_s}{\theta_a}\right)\ln|h_d| - \ln|h_m| + \sigma^2$$

(S14f)

For $\kappa = 2$, the expression for the hydraulic conductivity reads:

$$K(h) = \begin{cases} 0, & h \leq h_d \\[4pt]
K_s S_e^\tau \left\{ \dfrac{\left[ \dfrac{\theta_a}{2\theta_s h_m^2 \ln|h_d|} \left\{ \begin{array}{l} \dfrac{e^{4L^2}\operatorname{erf}(P(h)+2L)}{2} + \dfrac{1}{2}\left(\dfrac{h_m}{h}\right)^2 [\operatorname{erfc}(P(h))-2] - \\[6pt] M_2 e^{4L^2}\operatorname{erf}(P(h)+2L) - \dfrac{2Lh_m^2}{h^2\sqrt{\pi}} e^{-P^2(h)} \end{array} \right\} \right]_{h_d}^{h}}{\left[ \dfrac{\theta_a}{2\theta_s h_m^2 \ln|h_d|} \left\{ \begin{array}{l} \dfrac{e^{4L^2}\operatorname{erf}(P(h)+2L)}{2} + \dfrac{1}{2}\left(\dfrac{h_m}{h}\right)^2 [\operatorname{erfc}(P(h))-2] - \\[6pt] M_2 e^{4L^2}\operatorname{erf}(P(h)+2L) - \dfrac{2Lh_m^2}{h^2\sqrt{\pi}} e^{-P^2(h)} \end{array} \right\} \right]_{h_d}^{0}} \right\}^\gamma & \\[40pt]
= K_s S_e^\tau \left\{ \dfrac{\begin{array}{l} \dfrac{e^{4L^2}}{2}[\operatorname{erf}(P(h)+2L) - \operatorname{erf}(P(h_d)+2L)] + \\[8pt] \dfrac{h_m^2}{2}\left[\dfrac{\operatorname{erfc}(P(h))-2}{h^2} - \dfrac{\operatorname{erfc}(P(h_d))-2}{h_d^2}\right] - \\[8pt] M_2 e^{4L^2}[\operatorname{erf}(P(h)+2L) - \operatorname{erf}(P(h_d)+2L)] - \\[8pt] \dfrac{2Lh_m^2}{\sqrt{\pi}}\left(\dfrac{e^{-P^2(h)}}{h^2} - \dfrac{e^{-P^2(h_d)}}{h_d^2}\right) \end{array}}{\begin{array}{l} -\dfrac{e^{4L^2}}{2}[1 + \operatorname{erf}(P(h_d)+2L)] + \\[8pt] \dfrac{h_m^2}{2}\left[\dfrac{\operatorname{erfc}(P(0))-2}{0^2} - \dfrac{\operatorname{erfc}(P(h_d))-2}{h_d^2}\right] + \\[8pt] M_2 e^{4L^2}[1 + \operatorname{erf}(P(h_d)+2L)] - \dfrac{2Lh_m^2}{\sqrt{\pi}}\left(\dfrac{e^{-P^2(0)}}{0^2} - \dfrac{e^{-P^2(h_d)}}{h_d^2}\right) \end{array}} \right\}^\gamma, & h_d < h \leq 0 \end{cases}$$

(S14g)

with

$$M_2 = \left(1 - \frac{\theta_s}{\theta_a}\right)\ln|h_d| - \ln|h_m| + 2\sigma^2$$

(S14h)

There are several terms with zero in the denominator in Eqs. (S14c) and (S14h). In these terms, the numerator is zero as well. The terms $\exp(P^{-2}(h))\cdot h^{-1}$ and $\exp(P^{-2}(h))\cdot h^{-2}$ appearing in Eqs. (S14c) and (S14h) both become infinite for all physically acceptable values of $h_m$ and $\sigma$. As a consequence, the unsaturated hydraulic conductivity for both values of $\kappa$ suffers from the non-realistic increase near saturation diagnosed by Ippisch et al. (2006) for van Genuchten's (1980) soil water retention model, and the use of Eqs. (S14c-h) is not recommended.

Groenevelt and Grant (2004) proposed:

$$
\theta(h) = \begin{cases}
0, & h \leq -10^{6.9}\ \text{cm} \\[2mm]
g_1\left\{\exp\left(\dfrac{-g_0}{6.9^\eta}\right) - \exp\left[\dfrac{-g_0}{\left(\log_{10}|h|\right)^\eta}\right]\right\}, & -10^{6.9} \leq h \leq -1\ \text{cm} \\[2mm]
g_1\exp\left(\dfrac{-g_0}{6.9^\eta}\right), & h \geq -1\ \text{cm}
\end{cases}
\tag{S15a}
$$

where $g_0$, $g_1$, and $\eta$ are fitting parameters. The constant water content for matric potentials larger than -1cm is imposed. Groenevelt and Grant (2004) proposed a more flexible curve-shifting approach, but that procedure is cumbersome to perform in a global search parameter fitting operation. The derivative is

$$
\frac{\mathrm{d}\theta}{\mathrm{d}h} = \begin{cases}
0, & h \leq -10^{6.9}\ \text{cm} \\[2mm]
\dfrac{g_0 g_1 \eta [\ln(10)]^\eta}{|h|\left(\ln|h|\right)^{\eta+1}}\exp\left\{\dfrac{-g_0[\ln(10)]^\eta}{\left(\ln|h|\right)^\eta}\right\}, & -10^{6.9} \leq h \leq -1\ \text{cm} \\[2mm]
0, & h \geq -1\ \text{cm}
\end{cases}
\tag{S15b}
$$

This expression does not permit a closed-form expression for the hydraulic conductivity function.

Peters (2013) introduced four soil water retention models. He used a logarithmic model for adsorbed water that differed from that of Campbell and Shiozawa (1992) and the capillary model of either van Genuchten (1980) or Kosugi (1999). He developed versions for which the water content could be non-zero at the oven-dry matric potential $h_d$, which is incorrect but permits closed-from expressions of the hydraulic conductivity function. He also presented versions for which the water content is forced to be zero at $h_d$.

For the versions with nonzero water contents at $h_d$, the capillary bound and adsorbed water contents are added (Peters, 2013, Eq. (2))

$$
S_e(h) = wS^{cap}(h) + (1-w)S^{ad}(h)
\tag{S16}
$$

where the superscripts *cap* and *ad* reflect capillary bound and adsorbed water, respectively, and $w$ is a weighting factor ranging between 0 and 1. The van Genuchten-version with non-zero water content at $h_d$ is

$$\theta(h) = \begin{cases} \theta_s w\left[1 + (-\alpha h)^n\right]^{\frac{1}{n}-1} + \theta_s(1-w)\dfrac{1 - \dfrac{\ln\left(1+\dfrac{h}{h_a}\right)}{\ln\left(1+\dfrac{h_d}{h_a}\right)}}{1 - \dfrac{\ln(2)}{\ln\left(1+\dfrac{h_d}{h_a}\right)}}, & h_d \le h \le h_a \\[4em] \theta_s w\left[1 + (-\alpha h)^n\right]^{\frac{1}{n}-1} + \theta_s(1-w), & 0 \ge h \ge h_a \end{cases}$$

(S17a)

with derivative

$$\frac{\mathrm{d}\theta}{\mathrm{d}h} = \begin{cases} -\theta_s w \alpha(1-n)(-\alpha h)^{n-1}\left[1+(-\alpha h)^n\right]^{\frac{1}{n}-2} + \dfrac{\theta_s(1-w)}{h+h_a}\dfrac{1}{\ln\left(1+\dfrac{h_d}{h_a}\right) - \ln(2)}, & h \le h_a \\[3em] -\theta_s w \alpha(1-n)(-\alpha h)^{n-1}\left[1+(-\alpha h)^n\right]^{\frac{1}{n}-2}, & 0 \ge h \ge h_a \end{cases}$$

(S17b)

The parameter $h_a$ [L] represents the matric potential at which the soil reaches the maximum adsorbed water content.

The Kosugi-version with non-zero water content at air-dryness is

$$\theta(h) = \begin{cases} \dfrac{\theta_s w}{2}\,\mathrm{erfc}\left[\dfrac{\ln\left(\dfrac{h}{h_m}\right)}{\sigma\sqrt{2}}\right] + \theta_s(1-w)\dfrac{1 - \dfrac{\ln\left(1+\dfrac{h}{h_a}\right)}{\ln\left(1+\dfrac{h_d}{h_a}\right)}}{1 - \dfrac{\ln(2)}{\ln\left(1+\dfrac{h_d}{h_a}\right)}}, & h_d \le h \le h_a \\[5em] \dfrac{\theta_s w}{2}\,\mathrm{erfc}\left[\dfrac{\ln\left(\dfrac{h}{h_m}\right)}{\sigma\sqrt{2}}\right] + \theta_s(1-w), & 0 \ge h \ge h_a \end{cases}$$

(S18a)

with derivative

$$
\frac{d\theta}{dh} = \begin{cases}
-\dfrac{\theta_s w}{h\sigma\sqrt{2\pi}}\exp\left\{-\left[\dfrac{\ln\left(\dfrac{h}{h_m}\right)}{\sigma\sqrt{2}}\right]^2\right\} + \dfrac{\theta_s(1-w)}{h+h_a}\dfrac{1}{\ln\left(1+\dfrac{h_d}{h_a}\right)-\ln(2)}, & h \le h_a \\[4ex]
-\dfrac{\theta_s w}{h\sigma\sqrt{2\pi}}\exp\left\{-\left[\dfrac{\ln\left(\dfrac{h}{h_m}\right)}{\sigma\sqrt{2}}\right]^2\right\}, & 0 \ge h \ge h_a
\end{cases}
$$

(S18b)


The van Genuchten-version with zero water content when the soil is air dry is

$$
\theta(h) = \begin{cases}
0, & h \le h_d \\[2ex]
\theta_s\left(w\left\{\left[1+(-\alpha h)^n\right]^{\frac{1}{n}-1}-1\right\}+1\right)\dfrac{1-\dfrac{\ln\left(1+\dfrac{h}{h_a}\right)}{\ln\left(1+\dfrac{h_d}{h_a}\right)}}{1-\dfrac{\ln(2)}{\ln\left(1+\dfrac{h_d}{h_a}\right)}}, & h_d \le h \le h_a \\[4ex]
\theta_s w\left[1+(-\alpha h)^n\right]^{\frac{1}{n}-1}+\theta_s(1-w), & 0 \ge h \ge h_a
\end{cases}
$$

(S19a)

with derivative

$$\frac{\mathrm{d}\theta}{\mathrm{d}h} = \begin{cases} 0, & h \le h_d \\[2em] -\theta_s w \alpha (1-n)(-\alpha h)^{n-1}\left[1+(-\alpha h)^n\right]^{\frac{1}{n}-2} \dfrac{1-\dfrac{\ln\left(1+\dfrac{h}{h_a}\right)}{\ln\left(1+\dfrac{h_d}{h_a}\right)}}{1-\dfrac{\ln(2)}{\ln\left(1+\dfrac{h_d}{h_a}\right)}} \\[3em] +\dfrac{\theta_s\left(w\left\{\left[1+(-\alpha h)^n\right]^{\frac{1}{n}-1}-1\right\}+1\right)}{\left(h+h_a\right)\left[\ln\left(1+\dfrac{h_d}{h_a}\right)-\ln(2)\right]}, & h \le h_a \\[2em] -\theta_s w \alpha (1-n)(-\alpha h)^{n-1}\left[1+(-\alpha h)^n\right]^{\frac{1}{n}-2}, & 0 \ge h \ge h_a \end{cases} \tag{S19b}$$

The Kosugi-version with zero water content at $h_d$ is


$$\theta(h) = \begin{cases} 0, & h \le h_d \\[2em] \theta_s\left(w\left\{\frac{1}{2}\mathrm{erfc}\left[\dfrac{\ln\left(\dfrac{h}{h_m}\right)}{\sigma\sqrt{2}}\right]-1\right\}+1\right)\dfrac{1-\dfrac{\ln\left(1+\dfrac{h}{h_a}\right)}{\ln\left(1+\dfrac{h_d}{h_a}\right)}}{1-\dfrac{\ln(2)}{\ln\left(1+\dfrac{h_d}{h_a}\right)}}, & h_d \le h \le h_a \\[3em] \dfrac{\theta_s w}{2}\mathrm{erfc}\left[\dfrac{\ln\left(\dfrac{h}{h_m}\right)}{\sigma\sqrt{2}}\right]+\theta_s(1-w), & 0 \ge h \ge h_a \end{cases} \tag{S20a}$$

with derivative

$$\frac{\mathrm{d}\theta}{\mathrm{d}h} = \begin{cases} 0, & h \le h_d \\[2em] -\dfrac{\theta_s w}{h\sigma\sqrt{2\pi}}\exp\left\{-\left[\dfrac{\ln\left(\dfrac{h}{h_m}\right)}{\sigma\sqrt{2}}\right]^2\right\}\dfrac{1-\dfrac{\ln\left(1+\dfrac{h}{h_a}\right)}{\ln\left(1+\dfrac{h_d}{h_a}\right)}}{1-\dfrac{\ln(2)}{\ln\left(1+\dfrac{h_d}{h_a}\right)}} \\[2em] +\left(\left[w\left\{\dfrac{1}{2}\mathrm{erfc}\left[\dfrac{\ln\left(\dfrac{h}{h_m}\right)}{\sigma\sqrt{2}}\right]-1\right\}+1\right]\right)\dfrac{\theta_s}{(h+h_a)\left[\ln\left(1+\dfrac{h_d}{h_a}\right)-\ln(2)\right]}, & h \le h_a \\[2em] -\dfrac{\theta_s w}{h\sigma\sqrt{2\pi}}\exp\left\{-\left[\dfrac{\ln\left(\dfrac{h}{h_m}\right)}{\sigma\sqrt{2}}\right]^2\right\}\Bigg\}, & 0 \ge h \ge h_a \end{cases}$$

(S20b)

Both water retention functions based on van Genuchten's (1980) model (Eqs. (S17a) and (S19a)) lead to the requirement that $\kappa$ be smaller than $n$-1 (see Eq. (9)) and therefore do only have a physically acceptable conductivity curve associated with them for a very limited range of $\kappa$. The Kosugi-based versions (Eqs. (S18a) and (S20a)) suffer from the same lack of clarity about the behavior of the derivative as Khlosi et al.'s (2008) modified Kosugi function and require integer values of $\kappa$. Because of these limitations and the unwieldy nature of the equations (compare Eqs. (S14c-h)), their practical value seems limited.

Iden and Durner (2014) proposed modifications of Peters' (2013) models that permitted an analytical expression for the conductivity function even if the water content was forced to be zero at $h_d$. To apply the criterion of Eq. (4) to this modification, we multiply the derivative of their retention curve (their Eq. (3)) for adsorbed water by $h^{-\kappa}$:

$$\theta_s|h|^{-\kappa}\frac{\mathrm{d}S^{ad}}{\mathrm{d}h} = \frac{\theta_s|h|^{-\kappa-1}}{\ln(10)(\log|h_a|-\log|h_d|)}\left[1-\frac{\exp\left(\dfrac{\log|h_a|-\log|h|}{b}\right)}{1+\exp\left(\dfrac{\log|h_a|-\log|h|}{b}\right)}\right]$$

(S21)

where $b$ is a shape parameter. High values of $b$ lead to a sharp transition between the two linear segments in the semi-logarithmic form of the adsorbed water retention curve with different slopes. Iden and Durner recommend values of $b$ between 0.1 and 0.3.

In the limit as $h$ approaches zero, Eq. (S21) simplifies to

$$\lim_{h \to 0}\left(\theta_s |h|^{-\kappa} \frac{\mathrm{d}S^{ad}}{\mathrm{d}h}\right) = \frac{\theta_s |h|^{-\kappa-1}}{\ln(10)\left(\log|h_a| - \log|h_d|\right)}\left[1 - \frac{\exp\left(\dfrac{-\log|h|}{b}\right)}{1+\exp\left(\dfrac{-\log|h|}{b}\right)}\right]$$

(S22)

The approximation in the last term leads to the requirement that $\kappa < -1$ for the limit to go to zero for any value of $b$, but small values of $b$ allow larger ranges of $\kappa$. For $b = 0.3$, trial calculations showed that the value in the limit appears to be zero for $\kappa < 0.2$, which still rules out the established conductivity models. For $b = 0.1$, the limit is zero

even for large positive values of $\kappa$. It might be recommendable to fix $b$ at 0.1 instead of treating it as a fitting parameter.

The scaling of the capillary soil water retention curves proposed by Iden and Durner (2014) does not alleviate the problems with the van Genuchten curve near saturation while the Kosugi-function remains unwieldy. Conductivity functions for Peters' (2013) retention models will therefore not be derived.

Rudiyanto et al. (2015) developed a hysteretic version of Iden and Durner's (2014) model with the associated conductivity function. While of considerable interest, this model suffers from the same limitation as the original, and it will therefore not be further explored here.

In summary, many of the retention curves examined result in conductivity curves with physically unacceptable behavior near saturation, even though several of these expressions were derived with the explicit

purpose of providing closed-form expressions for the hydraulic conductivity. Only the Brooks-Corey function (1964) (BCO, Eq. (S1a)), the junction model of Rossi and Nimmo (1994) without the parabolic correction (RNA, Eq. (S9a)), and the model of Fayer and Simmons (1995) based on the Brooks-Corey (1964) retention function (FSB, Eq. (S12a)) lead to an acceptable conductivity model with full flexibility (three free parameters: $\kappa$, $\gamma$, $\tau$). The modified van Genuchten (1980) retention curve with a distinct air-entry value by Ippisch et al. (2006) (VGA, Eq.

(S7a)) leads to a conductivity model with two fitting parameters if $m = 1- 1/n$ because $\kappa = 1$.


**S2. Fitted parameter values for the 21 soils selected from the UNSODA database**

Table S1: The fitting parameters and their values for five parameterizations for clayey soils. The three-character parameterization label is explained in the main text. The soils are presented in the order of presentation of Fig. S1.

| | | | Soil (UNSODA identifier and classification according to Twarakavi et al. (2010)) | | | | | |
|---|---|---|---|---|---|---|---|---|
| | | | 1135 C2 | 1182 C2 | 1122 C4 | 1123 C4 | 1180 C4 | 1181 C4 |
| Paramete-rization | Parame-ter | Unit | | | | | | |
| BCO | $\theta_r$ | - | 4.79E-6 | 1.10E-4 | 3.76E-4 | 3.01E-4 | 1.63E-4 | 2.33E-5 |
| | $\theta_s$ | - | 0.420 | 0.549 | 0.362 | 0.358 | 0.497 | 0.456 |
| | $h_{ae}$ | cm | -106 | -0.980 | -10.0 | -10.0 | -10.9 | -5.17 |
| | $\lambda$ | - | 7.81E-2 | 4.40E-1 | 3.37E-2 | 2.70E-2 | 5.63E-2 | 5.39E-2 |
| FSB | $\theta_s$ | - | 0.420 | 0.548 | 0.360 | 0.356 | 0.495 | 0.456 |
| | $\theta_a$ | - | 0.400 | 0.306 | 0.350 | 0.340 | 0.491 | 0.348 |
| | $h_{ae}$ | cm | -106 | -0.229 | -5.74 | -10.0 | -8.58 | -13.2 |
| | $\lambda$ | - | 0.172 | 5.63E-2 | 6.59E-2 | 5.70E-2 | 100 | 8.08E-2 |
| RNA | $\theta_s$ | - | 0.420 | 0.549 | 0.370 | 0.370 | 0.497 | 0.456 |
| | $h_{ae}$ | cm | -106 | -3.62 | -9.99 | -10.0 | -0.149 | -7.63 |
| | $h_j$ | cm | -109 | -12.3 | -10.7 | -10.7 | -23.8 | -22.0 |
| | $h_d$ | cm | -1.66E8 | -1.00E9 | -1.00E9 | -1.00E9 | -1.00E9 | -1.00E9 |
| VGA | $\theta_r$ | - | 2.48E-2 | 5.09E-5 | 0.105 | 0.182 | 2.20E-2 | 2.12E-6 |
| | $\theta_s$ | - | 0.418 | 0.548 | 0.359 | 0.354 | 0.497 | 0.456 |
| | $\alpha$ | cm$^{-1}$ | 1.59E-3 | 1.33 | 1.27E-2 | 3.25E-3 | 15.1 | 1.70 |
| | $n$ | - | 1.18 | 1.05 | 1.08 | 1.16 | 1.06 | 1.05 |
| | $h_{ae}$ | cm | -45.6 | -0.523 | -2.97 | -9.50 | -6.45E-2 | -4.83 |
| VGN | $\theta_r$ | - | 0.270 | 9.53E-5 | 6.58E-4 | 0.213 | 1.18E-5 | 3.34E-6 |
| | $\theta_s$ | - | 0.412 | 0.548 | 0.359 | 0.354 | 0.497 | 0.456 |
| | $\alpha$ | cm$^{-1}$ | 1.02E-3 | 0.738 | 1.38E-2 | 2.87E-3 | 9.18 | 0.142 |
| | $n$ | - | 2.57 | 1.05 | 1.05 | 1.22 | 1.06 | 1.06 |

Table S2: The fitting parameters and their values for five parameterizations for silty soils. The three-character parameterization label is explained in the main text. The soils are presented in the order of presentation of Fig. S1.

| Parameterization | Parameter | Unit | Soil (UNSODA identifier and classification according to Twarakavi et al. (2010)) | | | | | |
| --- | --- | --- | --- | --- | --- | --- | --- | --- |
| | | | 3260 B2 | 3261 B2 | 3263 B2 | 3250 B4 | 3251 B4 | 4450 B4 |
| BCO | $\theta_r$ | - | 5.45E-6 | 8.42E-6 | 2.72E-7 | 8.12E-6 | 2.39E-5 | 5.36E-6 |
| | $\theta_s$ | - | 0.470 | 0.499 | 0.460 | 0.540 | 0.500 | 0.380 |
| | $h_{ae}$ | cm | -28.6 | -13.5 | -28.8 | -30.5 | -18.2 | -4.82 |
| | $\lambda$ | - | 0.281 | 0.256 | 0.255 | 0.182 | 9.57E-2 | 9.50E-2 |
| FSB | $\theta_s$ | - | 0.470 | 0.499 | 0.460 | 0.540 | 0.500 | 0.380 |
| | $\theta_a$ | - | 1.42E-5 | 6.90E-5 | 1.01E-5 | 0.173 | 0.431 | 0.320 |
| | $h_{ae}$ | cm | -28.6 | -13.5 | -28.8 | -30.0 | -10.9 | -0.888 |
| | $\lambda$ | - | 0.281 | 0.256 | 0.255 | 0.242 | 0.197 | 0.196 |
| RNA | $\theta_s$ | - | 0.470 | 0.499 | 0.460 | 0.540 | 0.500 | 0.380 |
| | $h_{ae}$ | cm | -28.6 | -13.5 | -28.8 | -30.5 | -18.2 | -4.81 |
| | $h_j$ | cm | -8.05E4 | -6.31E4 | -7.76E4 | -6.02E4 | -2.20E4 | -1.69E4 |
| | $h_d$ | cm | -2.82E6 | -3.14E6 | -3.89E6 | -1.45E7 | -7.66E8 | -6.23E8 |
| VGA | $\theta_r$ | - | 5.27E-2 | 4.89E-2 | 1.02E-3 | 1.58E-2 | 1.20E-4 | 4.77E-4 |
| | $\theta_s$ | - | 0.472 | 0.491 | 0.458 | 0.540 | 0.500 | 0.379 |
| | $\alpha$ | cm$^{-1}$ | 1.62E-2 | 1.84E-2 | 2.59E-2 | 1.311E-2 | 3.57E-2 | 0.164 |
| | $n$ | - | 1.47 | 1.52 | 1.30 | 1.26 | 1.11 | 1.10 |
| | $h_{ae}$ | cm | -1.66E-3 | -2.08E-3 | -19.2 | -5.36 | -7.11 | -5.93E-3 |
| VGN | $\theta_r$ | - | 5.27E-2 | 4.88E-2 | 4.52E-2 | 3.11E-2 | 8.93E-6 | 8.91E-5 |
| | $\theta_s$ | - | 0.472 | 0.491 | 0.461 | 0.540 | 0.501 | 0.379 |
| | $\alpha$ | cm$^{-1}$ | 1.62E-2 | 1.84E-2 | 1.53E-2 | 1.21E-2 | 2.61E-2 | 0.164 |
| | $n$ | - | 1.47 | 1.51 | 1.41 | 1.28 | 1.11 | 1.10 |


Table S3: The fitting parameters and their values for five parameterizations for sandy soils (A3 and A4 soils according to Twarakavi et al., 2010) . The three-character parameterization label is explained in the main text. The soils are presented in the order of presentation of Fig. S1.

| Parameter-ization | Parameter | Unit | Soil (UNSODA identifier and classification according to Twarakavi et al. (2010)) | | | | | |
| | | | 1120 A3 | 1143 A3 | 2110 A3 | 2132 A3 | 1121 A4 | 1133 A4 |
|---|---|---|---|---|---|---|---|---|
| BCO | $\theta_r$ | - | 1.77E-5 | 4.61E-6 | 2.31E-2 | 5.43E-5 | 2.72E-5 | 1.02E-4 |
| | $\theta_s$ | - | 0.311 | 0.279 | 0.360 | 0.303 | 0.350 | 0.330 |
| | $h_{ae}$ | cm | -10.0 | -7.00 | -18.5 | -8.00 | -10.0 | -206 |
| | $\lambda$ | - | 0.204 | 0.168 | 0.305 | 0.107 | 0.117 | 0.102 |
| FSB | $\theta_s$ | - | 0.311 | 0.279 | 0.360 | 0.308 | 0.346 | 0.330 |
| | $\theta_a$ | - | 5.27E-5 | 1.95E-4 | 7.30E-2 | 0.298 | 0.324 | 0.310 |
| | $h_{ae}$ | cm | -10.0 | -7.00 | -18.4 | -3.24 | -10.0 | -206 |
| | $\lambda$ | - | 0.204 | 0.169 | 0.342 | 0.422 | 0.377 | 0.216 |
| RNA | $\theta_s$ | - | 0.311 | 0.279 | 0.360 | 0.303 | 0.350 | 0.330 |
| | $h_{ae}$ | cm | -10.0 | -7.00 | -18.0 | -8.00 | -10.0 | -220 |
| | $h_j$ | cm | -8.09E4 | -7.59E4 | -9.83E4 | -3.90E4 | -7.26E4 | -6.22E4 |
| | $h_d$ | cm | -1.10E7 | -2.86E7 | -3.78E6 | -4.37E8 | -3.53E8 | -7.96E8 |
| VGA | $\theta_r$ | - | 7.21E-2 | 9.77E-2 | 0.126 | 1.26E-4 | 5.20E-5 | 0.201 |
| | $\theta_s$ | - | 0.305 | 0.278 | 0.360 | 0.306 | 0.339 | 0.324 |
| | $\alpha$ | cm$^{-1}$ | 1.72E-2 | 4.54E-2 | 2.63E-2 | 6.10E-2 | 7.22E-3 | 7.34E-4 |
| | $n$ | - | 1.69 | 1.52 | 1.84 | 1.14 | 1.27 | 2.99 |
| | $h_{ae}$ | cm | -3.81E-2 | -6.43E-3 | -1.35E-2 | -3.32E-3 | -5.00E-2 | -25.8 |
| VGN | $\theta_r$ | - | 7.21E-2 | 9.16E-2 | 0.126 | 2.02E-2 | 4.19E-5 | 0.201 |
| | $\theta_s$ | - | 0.304 | 0.278 | 0.360 | 0.305 | 0.339 | 0.324 |
| | $\alpha$ | cm$^{-1}$ | 1.72E-2 | 4.71E-2 | 2.63E-2 | 5.46E-2 | 7.15E-3 | 7.34E-4 |
| | $n$ | - | 1.69 | 1.48 | 1.84 | 1.16 | 1.26 | 3.02 |


Table S4: The fitting parameters and their values for five parameterizations for sandy soils (A1 and A2 soils according to Twarakavi et al., 2010) . The three-character parameterization label is explained in the main text. The soils are presented in the order of presentation of Fig. S1.

| | | | Soil (UNSODA identifier and classification according to Twarakavi et al. (2010)) | | |
|---|---|---|---|---|---|
| | | | 2126 A1 | 1142 A2 | 2104 A2 |
| Parameterization | Parameter | Unit | | | |
| BCO | $\theta_r$ | - | 1.63E-2 | 9.36E-5 | 2.27E-2 |
| | $\theta_s$ | - | 0.377 | 0.250 | 0.398 |
| | $h_{ae}$ | cm | -6.78 | -7.00 | -6.79 |
| | $\lambda$ | - | 0.846 | 0.211 | 0.434 |
| FSB | $\theta_s$ | - | 0.377 | 0.250 | 0.398 |
| | $\theta_a$ | - | 2.59E-2 | 2.96E-4 | 5.46E-2 |
| | $h_{ae}$ | cm | -6.76 | -7.00 | -6.73 |
| | $\lambda$ | - | 0.862 | 0.210 | 0.468 |
| RNA | $\theta_s$ | - | 0.378 | 0.250 | 0.398 |
| | $h_{ae}$ | cm | -6.37 | -7.00 | -6.17 |
| | $h_j$ | cm | -9.08E4 | -8.17E4 | -7.52E4 |
| | $h_d$ | cm | -3.68E5 | -9.45E6 | -1.13E6 |
| VGA | $\theta_r$ | - | 3.39E-2 | 9.64E-2 | 3.42E-2 |
| | $\theta_s$ | - | 0.376 | 0.242 | 0.398 |
| | $\alpha$ | cm$^{-1}$ | 6.84E-2 | 1.98E-2 | 6.97E-2 |
| | $n$ | - | 2.73 | 3.05 | 1.64 |
| | $h_{ae}$ | cm | -3.49E-2 | -0.246 | -1.62E-2 |
| VGN | $\theta_r$ | - | 3.39E-2 | 9.42E-2 | 3.41E-2 |
| | $\theta_s$ | - | 0.376 | 0.242 | 0.398 |
| | $\alpha$ | cm$^{-1}$ | 6.84E-2 | 1.98E-2 | 6.97E-2 |
| | $n$ | - | 2.73 | 2.93 | 1.64 |


**S3. Root means square errors of the parameter fits for the 21 soils selected from the UNSODA database**

Table S5.  Root mean square errors of the parameter fits for the clayey soils.

| Parameterization | Soil (UNSODA identifier and classification according to Twarakavi et al. (2010)) | | | | | |
|---|---|---|---|---|---|---|
| | 1135 C2 | 1182 C2 | 1122 C4 | 1123 C4 | 1180 C4 | 1181 C4 |
| BCO | 0.0913 | 0.0494 | 0.0349 | 0.0489 | 0.0187 | 0.0428 |
| FSB | 0.0721 | 0.0441 | 0.0212 | 0.0320 | 0.1196 | 0.0360 |
| RNA | 0.0812 | 0.0913 | 0.1235 | 0.1501 | 0.0347 | 0.0570 |
| VGA | 0.0487 | 0.0485 | 0.0204 | 0.0243 | 0.0192 | 0.0429 |
| VGN | 0.0208 | 0.0488 | 0.0197 | 0.0244 | 0.0194 | 0.0433 |


Table S6. Root mean square errors of the parameter fits for the silty soils.

| Parameterization | Soil (UNSODA identifier and classification according to Twarakavi et al. (2010)) | | | | | |
| --- | --- | --- | --- | --- | --- | --- |
| | 3260 B2 | 3261 B2 | 3263 B2 | 3250 B4 | 3251 B4 | 4450 B4 |
| BCO | 0.0793 | 0.1316 | 0.0973 | 0.0822 | 0.0551 | 0.0499 |
| FSB | 0.0794 | 0.1316 | 0.0973 | 0.0815 | 0.0395 | 0.0445 |
| RNA | 0.0793 | 0.1316 | 0.0973 | 0.0822 | 0.0551 | 0.0499 |
| VGA | 0.0455 | 0.0607 | 0.0818 | 0.0413 | 0.0466 | 0.0485 |
| VGN | 0.0455 | 0.0607 | 0.0638 | 0.0413 | 0.0474 | 0.0485 |

Table S7. Root mean square errors of the parameter fits for the sandy soils (A3 and A4 soils according to Twarakavi et al., 2010)

| Parameterization | Soil (UNSODA identifier and classification according to Twarakavi et al. (2010)) | | | | | |
| --- | --- | --- | --- | --- | --- | --- |
| | 1120 A3 | 1143A3 | 2110 A3 | 2132 A3 | 1121 A4 | 1133 A4 |
| BCO | 0.0926 | 0.0501 | 0.0507 | 0.0356 | 0.1288 | 0.0803 |
| FSB | 0.0926 | 0.0500 | 0.0507 | 0.0292 | 0.1054 | 0.0700 |
| RNA | 0.0926 | 0.0500 | 0.0507 | 0.0356 | 0.1288 | 0.0775 |
| VGA | 0.0446 | 0.0334 | 0.0377 | 0.0203 | 0.0720 | 0.0175 |
| VGN | 0.0446 | 0.0334 | 0.0377 | 0.0207 | 0.0720 | 0.0175 |

Table S8. Root mean square errors of the parameter fits for the sandy soils (A1 and A2 soils according to Twarakavi et al., 2010)


| Parameterization | Soil (UNSODA identifier and classification according to Twarakavi et al. (2010)) | | |
| --- | --- | --- | --- |
| | 2126 A1 | 1142 A2 | 2104 A2 |
| BCO | 0.0620 | 0.0990 | 0.0480 |
| FSB | 0.0626 | 0.0990 | 0.0517 |
| RNA | 0.0659 | 0.0989 | 0.0553 |
| VGA | 0.0330 | 0.0250 | 0.0278 |
| VGN | 0.0330 | 0.0252 | 0.0278 |

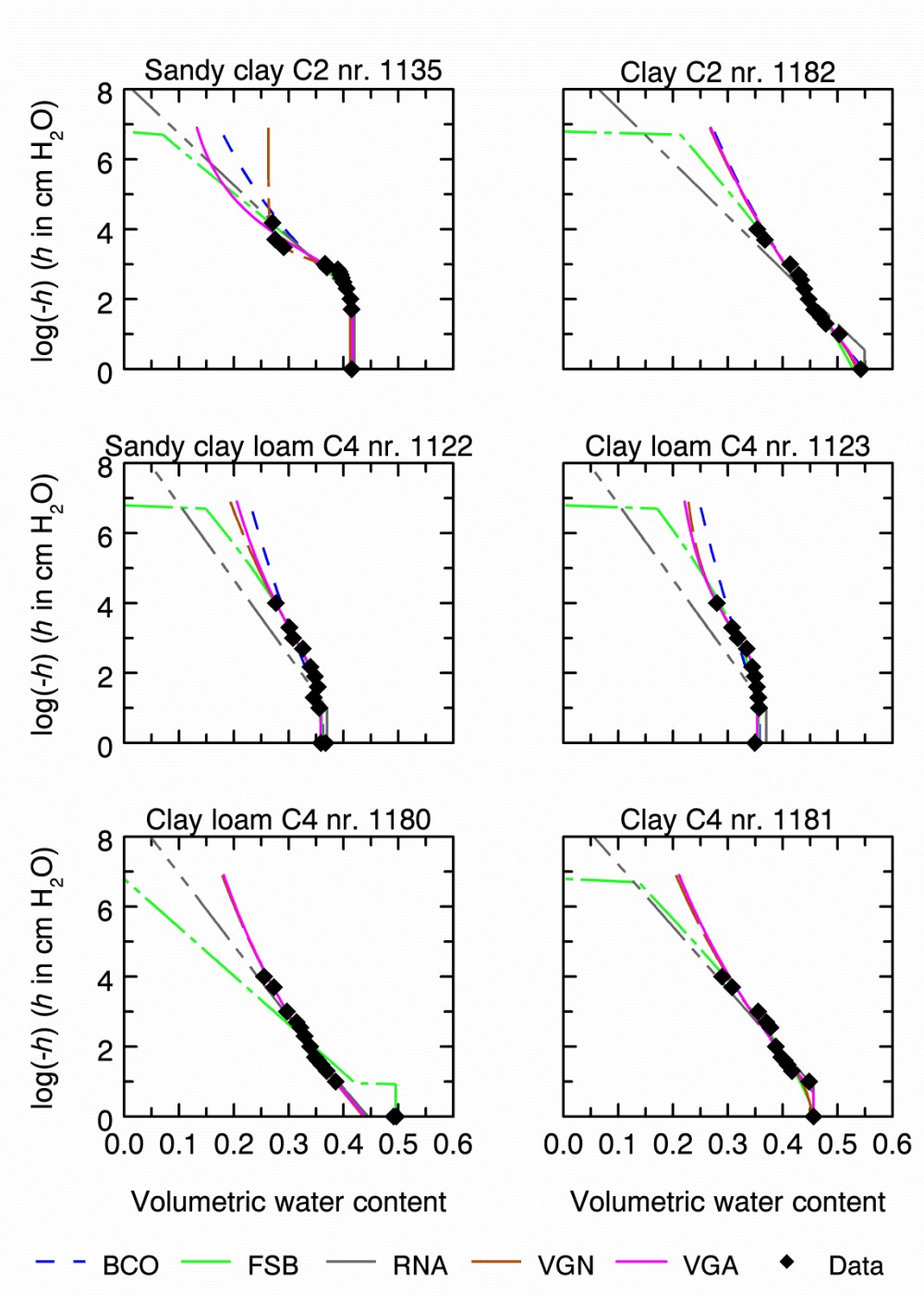

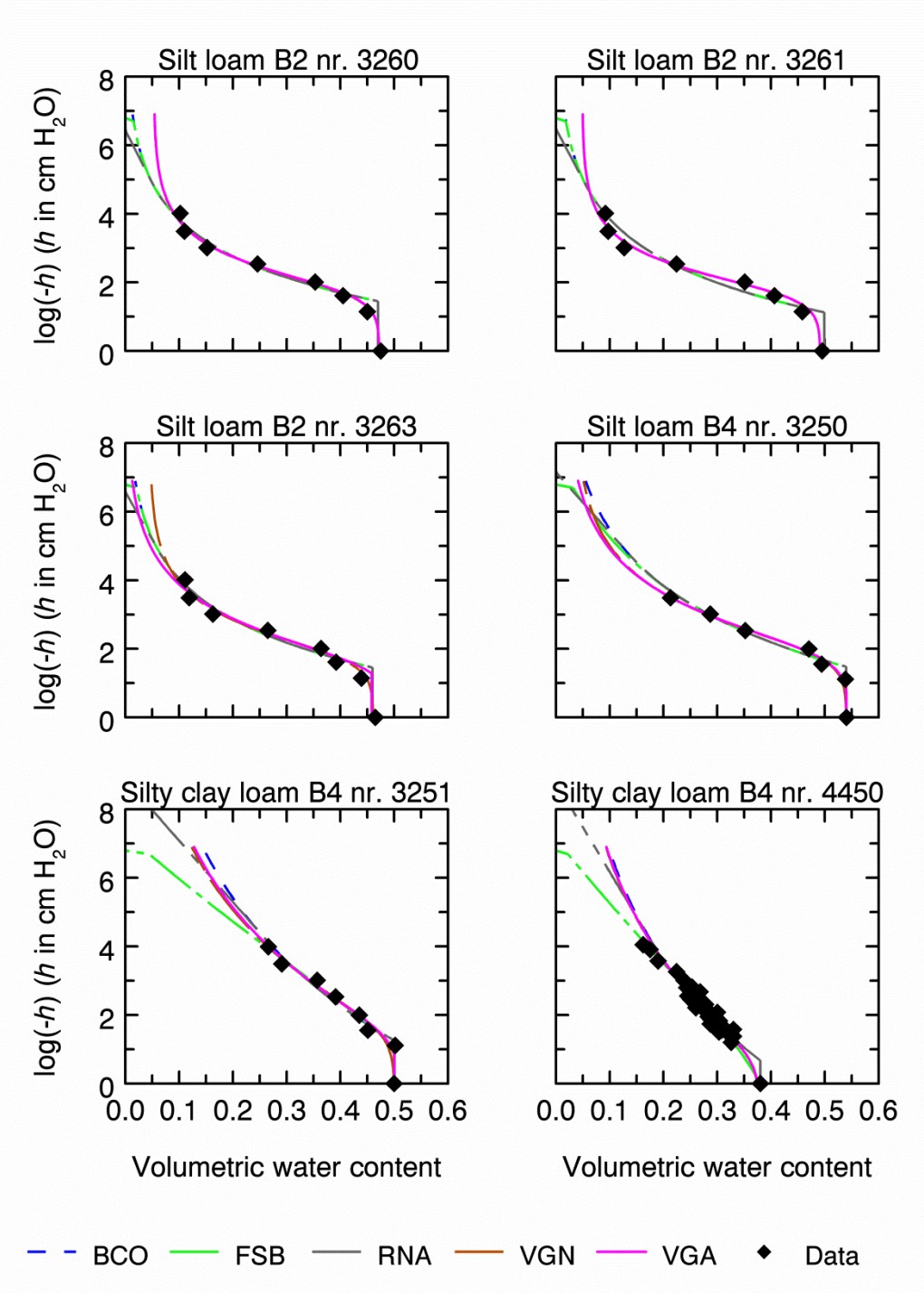

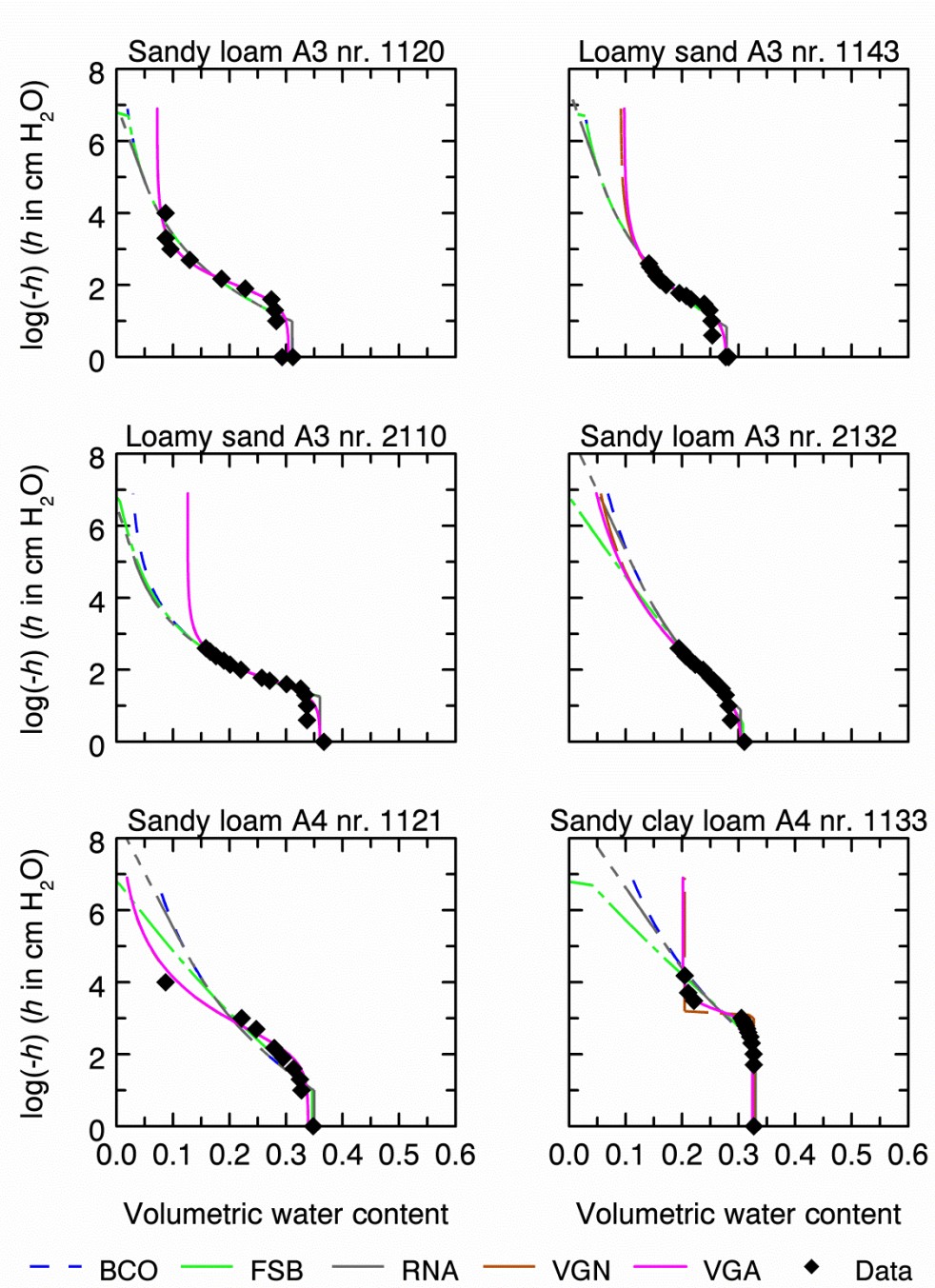

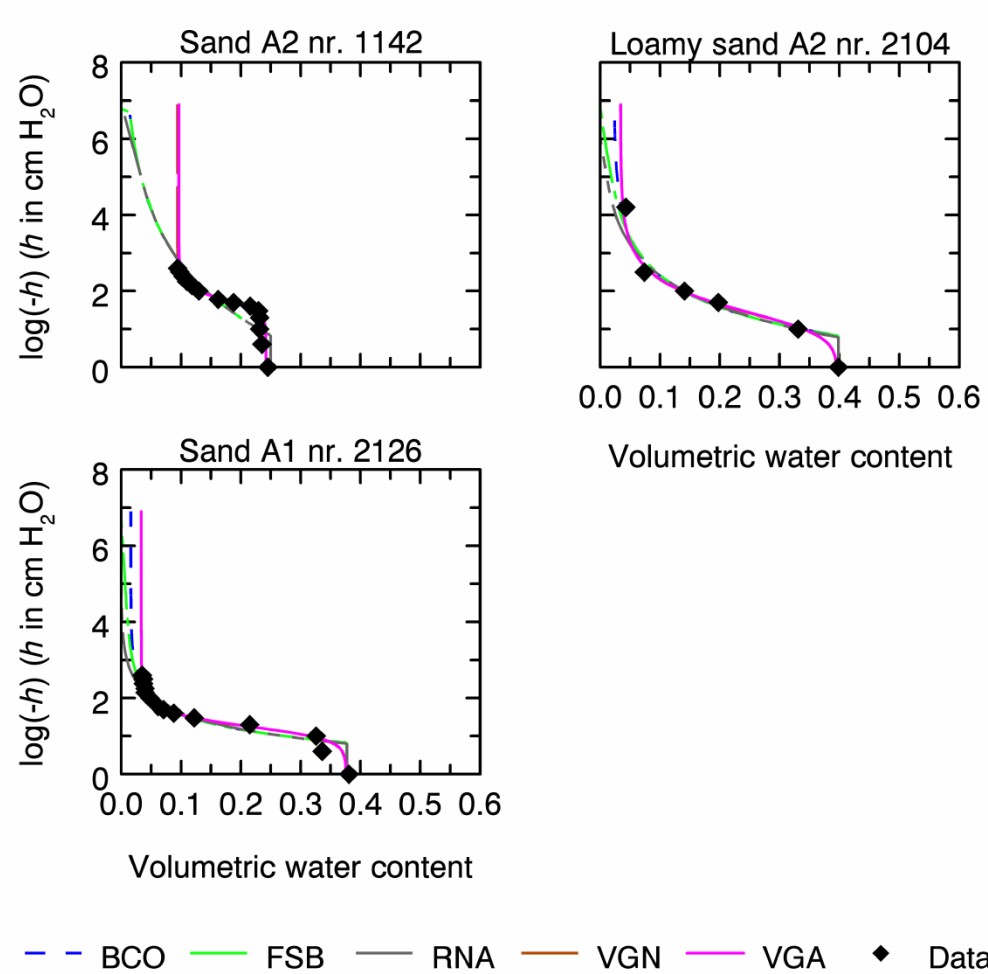

Figure S1. Fits of five parameterizations to data from 21 soils selected from the UNSODA database. The soils are characterized by their USDA texture classification and Twarakavi et al.'s (2010) classification, and identified by the four-digit number in the UNSODA database. The parameterizations are those of Brooks and Corey (1964) (BCO), Fayer and Simmons (1995) with the capillary bound water content forced to zero when the adsorbed water content reaches zero (FSB), Rossi and Nimmo (1994), but with a non-zero air-entry value (RNA), van Genuchten (1980) (VGN), and Ippisch et al. (2006) (VGA). Note that the vertical variation of the water content in samples at hydrostatic equilibrium was accounted for during the fitting process. The data in the wet range may therefore give a smoother representation than the underlying retention curve (Liu and Dane, 1995). N.B. Data points obtained at zero matric potential are plotted for pF = 0 (corresponding to $h$ = -1 cm).