# Peer review of "Parametric soil water retention models: a critical evaluation of expressions for the full moisture range."

_Hydrology and Earth System Sciences, 2016_

## Referee Comment (RC1) · Anonymous Referee #1 · 25 May 2016

This paper compares models of water retention and hydraulic conductivity for theoretical consistency, their ability to match measurements in four soils across a wide range of potentials from near saturation to very dry conditions and, from a functional point of view, in terms of simulated water balances in a dry climate.

The review of different hydraulic models approaches is comprehensive (at least for models based on the capillary bundle concept) as well as revealing, while the functional model comparison under dry climate conditions is also potentially very interesting. I only have a few concerns, which I think could easily be addressed by the authors:

1.) No information is given on the four soils. I checked in Schelle et al. (2013) and discovered that they investigated samples with these four textures taken from three

different sites in Germany and also for disturbed and undisturbed soil. The authors must give specific information on the location of the sampling sites, land use at the sites, whether the samples were disturbed (packed) or undisturbed, sample diameters and number of replicates.

2.) Looking at the figures, I am a little surprised by the apparent lack of structural pores that fill/drain in the tension range close to saturation, say < 10 cm (especially in the finer-textured soils). It does seem to me that throughout the paper the authors only consider the effects of textural pores and do not consider or acknowledge the existence of structural pores. Is this because you only looked at disturbed (packed) samples? Please discuss and clarify this point.

3.) The authors comprehensively present the equations of the models, but they write nothing about their conceptual basis. A few introductory sentences are needed to explain the concepts and assumptions underlying these capillary bundle models, including the fact, for example, that they assume a mono-modal size distribution. Alternative approaches could also be mentioned (e.g. bimodal models, fractal models etc.).

4.) A more extensive database than four soils would ideally be preferable to enable a reliable discrimination between alternative water retention models although I understand that few datasets include the very dry end of the range. The authors could discuss this.

5.) The simulation set-up does not appear to be optimal. My concern relates to the initial condition (hydrostatic equilibrium) in relation to both the bottom boundary condition (unit hydraulic gradient) and the length of the simulation, which was quite short (999 days). Judging from what the authors write (e.g. at lines 733-734), it appears that for this dry climate, this combination results in a simulated water balance that includes a non-negligible term for the change in profile water storage, which is not satisfactory. Water balances in the field should have a negligible change of storage in the long-term and scenario simulations with models should be set-up to mimic this as far as possible. As the authors note, the change of storage is different for the different models, which

makes it difficult to compare them with respect to the important terms in the water balance (i.e. recharge, evaporation).

The best way to deal with this is to run a 'spin-up' ('warm-up') period first (separately for each model), then use the final state variables (water contents, potentials) at the end of this period as the initial condition for the actual simulation period (using the same driving data for both periods). The water balances for the second simulation period should then be checked to make sure that the change of storage is negligible. If it is not, the warm-up and simulation periods should be extended until it is. Only then can the water balances simulated by the different models be properly compared. If the authors do this, I suspect the differences between the model formulations will be smaller, though probably not negligible.

It would also be a good idea to summarize the simulated water balances for the different models in a simple table (precipitation, evaporation, recharge, change of storage).

---

## Referee Comment (RC2) · Anonymous Referee #2 · 28 Jul 2016

The manuscript tries to address several issues, e.g., the deficiency of soil water retention (SWR) models near saturation, SWR models near the dry end, development of a general criterion for plausible hydraulic conductivity (K) curves, comparison of different SWR and K models, different methods for parameter optimization, numerical simulation to evaluate model selection on drainage and evapotranspiration, and model calibration/inverse model. Even the abstract contains multiple paragraphs, each of which addresses a different issue. As a result, neither of the issues is convincingly addressed.

In my opinion, the development of a general criterion (Eq. 4) for plausible K curves is interesting and can be the main issue of the manuscript. If so, the manuscript needs to

provide convincingly theory and experiment results and the conditions a model can or cannot be used. However, to validate the correctness of the criterion, experiment errors need to be considered as well. For example, the SWRs were measured with several methods and the results differ more or less for a given soil. If the difference among different SWR models is less than the measurement error, the SWR model should be fine. The manuscript needs to provide the implications to the readers how they can use the models correctly or appropriately. Section 2.2 is very long and can go to an appendix.

I don't think the numerical simulations using different SWR and K models can be used to validate or invalidate the models. First, modeling evaporation and drainage is challenging and different simulators can produce very different results because they may use different algorithms to solve the problem. Second, some models perform better for certain flow process (e.g., infiltration, redistribution) or soil types while other models perform better for different processes (e.g., evaporation, drainage) or soil types. Third, the assignment of initial and boundary conditions can lead to very different results. For example, for a soil that is never saturated for a simulation, the inaccuracy at the near saturation condition probably does not matter much.

The dry-end issue may be left out because it was mentioned but not addressed. The parameter optimization should just be the methods to obtain parameters. It's better if the uncertainty in the optimized parameters be given.

For the reasons above, the manuscript is not publishable in the current form.

---

## Referee Comment (RC3) · Anonymous Referee #3 · 1 Aug 2016

This paper deals with the important issue of the choice of the appropriate water retention (WR) and hydraulic conductivity (HC) model combinations to be used for simulations of hydrological processes. The objectives of the paper were:

1. Reviewing a number of WR and HC models and introducing a general criterion for WR functions parameterization to ensure physical plausibility of HC curves, especially near saturation. This was done by generalizing the approach proposed by Ippisch et al. (2006) (originally by Vogel et al., 2000) and by extending it to WR and HC parameterizations different from those analyzed by Ippisch et al.;

2. Verifying the performance of the different model combinations in matching experimental WR and HC data for a wide range of water contents, between saturation and

very dry conditions, and for four soils (sandy, silty, silty-loam and clayey). In order to account for the different experimental errors related to the different experimental techniques for measuring WR data, the authors introduced an objective function accounting for errors varying with the range of water contents considered;

3. Finally, the performance of the model combinations was analyzed in terms of functional properties, by looking at the numerical predictions of drainage, evaporation and infiltration processes obtained by the different WR and HC models in different soils.

The main finding was that different parameterizations may lead to drainage fluxes that may vary of more than one order magnitude, while they may have only limited effects on evaporation fluxes. Also, for a given WR model, the Bourdine (B) and Mualem (M) models for the HC provided similar hydrological behavior, while the Alexander and Skaggs (AS) model gave physically unreasonable predictions of the processes examined.

Based on my reading of the manuscript, I think it is in general significant even if the approach is not novel. The manuscript is fairly structured. The introduction of the paper illustrates quite clearly the rationale and the objectives of the work. However, it does not provide an exhaustive literature review about the approach used, with references missing important papers (since the 1990s) dealing with the same issue. Figures and Tables supports the findings, especially the part on the prediction of the hydrological processes selected for analysing the model performance in terms of functional properties.

The strength of the work lies in the fact that the authors provide a systematic and comprehensive review of the WR and HC models available for hydrological analysis. Crucial in the manuscript is the effort to unify and generalize the analyses of the WR behavior provided by the different models, especially near saturation.

On the other side, I see some limits in the manuscript that can be summarized as follows: 1. The comparison among models is not novel and is based on a too limited WR and HC dataset . There are papers in the past dealing with the same issue of

analyzing the performance of WR and HC functions, based on huge datasets, that the authors do not consider at all. I mainly refer for example to the work by Leij et al. (1997). The authors assembled different types of mathematical formulations and tested them on a large data set crossing practically the whole textural triangle. I would also add Cornelis et al. (2005).

2. The approaches used are all unimodal. As noted by the anonymous Referee #1, the dataset misses most of the information on structural pores. It is not a case that most of the WR curves in figure 2 provide a similar flat behavior in the pF range 0-1. And yet, central in the manuscript is the behavior of the WR functions near saturation. It is well known that HC models based on the statistical capillary-bundle approach, which are based on the Hagen-Poiseuille law and which integrate the reciprocal of the pressure head to obtain the hydraulic conductivity (as in the case of the Mualem conductivity expression), are particularly sensitive to the slope of the water retention near saturation (Durner, 1994; Coppola, 2000). The effects of a wrong description of the WR close to saturation may have impressive effects on the hydraulic conductivity estimation, with an impact on the soil hydrological processes predictions which may well be larger than the effects observed by the authors in their unimodal analysis (see for example Coppola et al., 2012). Bimodality may also exist quite far from saturation. By looking at the figure 2d, the data trend may well suggest a bimodal behavior in the pF range 2-3 (more or less). It is thus not a case that for the silty loam all the WR functions give a poor description of the data in the drier region.

3. There is no effort for recommending which model combination is the most suitable to be used in a given water content range;

4. One of the objectives of the paper is " . . . a robust fitting method applicable to various parameterizations and capable of handling data with different data errors" (see end of page 2). The authors introduce the reciprocal of the variance of the errors for weighting differently the single data in the various water content ranges. This is an extension of what one generally does when introducing different quantities in the

objective function. And yet, they do not give any information on how they estimated these variances. Selection of data error variances seems to be crucial for determining the performance of the different WR models in describing the experimental data. In other words, the fitting results shown in the figure 2 may be partly an effect of the model parameterization and partly an effect of the error selection for single data;

Some other remarks

By looking at the HC curves in the figure 3, it seems that the AS curves of the three soils are almost the same. The same may be said for the Mualem and Burdine curves for the silt and the silt-loam. It may only be a result of the axis extent used for the HC curves. Please, consider to show the curves for a smaller HC range of values. In general, the AS curve remains stably higher than the B and the M HC curves. As the authors are providing "...a critical evaluation of parametric expressions", the authors should even shortly explain the reason for this behavior compared to the Burdine and the Mualem models.

In any case, in the section 4.2.1., what the authors consider as "physically implausible" behavior (when discussing about sustained, constant flux leaving the silt soil profile during prolonged dry periods in the AS case) strictly depends on the high values the AS HC curve keeps even for very dry conditions. I would not like it to be also the effect of numerical problems. For example, by looking at the graphs in the panel 4c, the silt-RNA_Mualem/Burdine cumulative drainage curves cross the silt-RNA_AS curve. The latter remains unexpectedly lower, given that the WR curve is the same and the AS curve is systematically higher than the Burdine and the Mualem curves). Maybe, the authors should give more details about the evolution of the pressure head at the bottom boundary conditions during the numerical simulations. They do this only for sand. The reader could do an effort for extending the discussion for the sand to the silt soil. However, the authors may agree that this may be quite laborious.

In the figure 5, it seems that the evaporation fluxes are inversely related to the drainage

fluxes. Higher drainage induces lower pressure head in the soil profile resulting in lower upward fluxes. Again, one should have a look at the pressure gradients at the soil surface. Nonetheless, in the dry region the AS curve may be even five or more orders of magnitude larger than the Mualem and Burdine HC curves. Thus, in the panel 4d (just as an example), I would not expect higher VGA_Mualem than VGA_AS evaporative fluxes, unless the hydraulic gradient at the soil surface in AS case be five or more order of magnitude lower than in the M/B cases.

Overall, I have no major problems with the manuscript and recommend publication after the authors have discussed these remarks.

References

Ippisch, O., Vogel, H.-J. and Bastian, P.: Validity limits for the van Genuchten-Mualem model and implications for parameter estimation and numerical simulation, Adv. Water Resources, 29,1780-1789, 1050 doi:10.1016/j.advwatres.2005.12.011, 2006

Vogel, T., van Genuchten, M.Th. and Cislerova, M.: Effect of the shape of the soil hydraulic functions near saturation on variably-saturated flow predictions, Adv. Water Resour, 24, 133-144, 2000.

Leij et al., 1997. Closed-form expressions for water retention and conductivity data. Ground Water, vol. 35, n.5, 848-858

Cornelis et al., 2005. Comparison of Unimodal Analytical Expressions for the Soil-Water Retention Curve. Soil Sci. Soc. Am. J. 69:1902–1911

Coppola, A., 2000. Unimodal and bimodal descriptions of hydraulic properties for aggregated soils. Soil Science Society American Journal 64, 125–1262.

Durner, W., 1994. Hydraulic conductivity estimation for soils with heterogeneous pore structure. Water Resources Research 30, 211–223.

Coppola, A., A. Basile, A. Comegna, and N. Lamaddalena (2009c), Monte Carlo analysis of field water flow comparing uni- and bimodal effective hydraulic parameters for structured soil, J. Contam. Hydrol., 104,

---

## Author Comment (AC1) · 4 Aug 2016

Reply to referee 1 Gerrit H. de Rooij, Raneem Madie, Henrike Mielenz, and Juliane Mai

Reviewer 1 provided some thoughtful comments and useful suggestions in a review that is overall positive. We adopt the numbering of the review in our response.

1) Reviewer 1 would like to have more information on the soils. This is available and more complete information can be provided in a revised version of the paper, should the editor decide that a revision is warranted. In this stage it is sufficient to clarify that the data sets we used were obtained with undisturbed samples.

2) Reviewer 1 expresses concerns about the lack of representation of structural pores in the retention functions that we analyzed. If the soil is aggregated, one could end up with a system in which the pores within the aggregates and the pores between the aggregates combined produce the retention curve of the soil. W. Durner (1994, Water Resour. Res. 30: 211-223) proposed bimodal retention curves for this, for which he chose superpositions of two VGN curves (abbreviation according to the manuscript). This approach can be generalized to other retention models, and conductivity curves should be able to be derived from those, but we considered that a step too far at this stage of the work, since this approach has its own sets of issues (see our response to Reviewer 3, who also brings up soils with multimodal retention curves). We therefore intend to discuss the issue of multimodality in the Introduction to better indicate the context of our work, but at this stage we would like to limit ourselves to unimodal soils too avoid cluttering the paper.

When structural pores are understood to be macropores, it transpires from our literature review that not representing these in retention curves seems to be the rule, and there are some valid reasons for this. The flow through macropores is generally considered separated from matrix flow. Macropore flow may well be turbulent, and tends to respond very rapidly to rainfall events heavy enough to fill these macropores. The exchange of water between the macropore network and the matrix is orders of magnitude slower, as is the matrix flow itself. Models that incorporate macropores therefore are of the multidomain-type: one domain is reserved for the macropores, and at least one other domain comprises the matrix. Different equations describe the flow in each domain, and there is at least one coupling equation that governs the exchange of water between the domains. For such models, a soil water retention curve that only describes the matrix is ideal.

A second reason to exclude the macropores from the model equations is the limited size of the samples that are used to determine the data points to which the equations are fitted (we will report these as well if we are permitted to revise the paper). These

samples cannot be too large since it is desirable that the water content at hydrostatic equilibrium does not vary too much over the vertical extent of the column (even though the fitting code we developed takes this into account). Also, if one wishes to capture soil heterogeneity, one should aim for a sample size that is not much larger than the representative elementary volume. In most soil physics labs, the samples used are about 100 cm3. This usually is too small to be able to realistically sample both the matrix and the macropore network. A macropore may well be present, but it is likely to be cut off by the sample cylinder and by necessity disconnected from the rest of the network. The presence or absence of a macropore may also be a stochastic process if the macropore spacing is significantly larger than the sample diameter.

With this in mind we decided not to try to find retention curve expressions that include macropores to be included in the evaluation. We agree with the reviewer that the role of macropores is significant though, so we intend to clarify the text by incorporating elements of this reply in the main text.

3) The reviewer asks a theoretical basis for the nature of the soil water retention and conductivity models. Durner (1994) (reference provided by reviewer 3) provides a thorough overview of this in his introduction, and there are some review papers on this. We will expand on this in the revision, probably by taking the key elements of earlier discussions in the literature and referring the readers to the relevant papers for more details.

4) The reviewer would have liked to see a larger database of soils. We agree with the reviewer that such data are not always easy to come by in the dry end, but more importantly we wanted to examine the effect of the parameterizations on various fluxes for various textures. With four widely different textures we were already quite comprehensive, and each additional texture would increase the number of model runs considerably without adding much more clarity. We also point out that these parameterizations are not new and each was presented in the literature with its own tests on various soils, and then reviewed using data from additional soils by Khlosi et al. (2008, reference in
the original paper) and Leij et al. (1997, reference provided by reviewer 3 – will be included in the revised version). Adding even more textures to this already rich spectrum did not seem to offer clear benefits, and would expose us to the risk of repeating Khlosi et al.'s and Leij et al.'s work.

5) The reviewer argues that the simulation period should be extended to ensure that the storage change between the start and the end of the simulations is nearly zero. This approach is popular in moderate climates with groundwater tables within a few meters of the soil surface. In such climates evapotranspiration is small in winter, and winter rains and snowmelt wet up the soil profile. In summer, evapotranspiration kicks in and depletes the water in the soil profile. There typically is no 'memory' in the unsaturated zone of rainfall in the previous years. At some time well after snowmelt and just before evapotranspiration starts to rise, the soil profile will generally be well wetted but not so strongly that there is still a high downward flux siphoning off excess water to the groundwater. If one uses that date to start a one-year simulation one should be able to achieve the near-zero storage change that the reviewer desires.

Desert climates and desert soils have a different dynamic: the water delivered by small or isolated rain showers will not infiltrate deeply and evaporate completely. Only rainfall clusters and/or massive showers delivering large volumes of water will cause deep infiltration. This water will create a wetting front of which trial calculations for a related study (not yet published) showed that it will be visible in the soil profile for many years, contributing to the storage term all the time. Extrapolating this to unsaturated zone that can be several hundred meters deep this leads to the conclusion that the unsaturated zone in a desert can have a memory of past major rainfall events that spans many decades if not longer.

This makes the zero-storage change somewhat arbitrary: in the presence of such a deep wetting front that will eventually provide groundwater recharge, the occurrence of small, irrelevant rainfall events shortly before the start or the end of the simulation period will affect the sign and the magnitude of the storage change over the simulation

period, but this will not provide any useful information about the magnitude of ground-water recharge. We therefore did not make this an objective of our simulations and believe it does not constitute a valid benchmarking criterion for comparing models.

Also, the magnitude of the storage change is not so important for this particular study. We were interested in seeing how unsaturated zones of different textures described by different parameterizations responded to strong atmospheric forcings: long periods of high potential evapotranspiration interspersed with heavy rainfall. We designed a scenario using meteorological data from a climatic region that had these extreme characteristics.

That being said, the initial condition of hydrostatic equilibrium had an effect on the simulated results. As we mentioned in the paper, the effect of the initial condition became very small well before the first rainfall in the simulation period. Particularly the effect on the bottom flux damped out almost instantly during which there was little or no precipitation. We therefore propose to consider this initial rain-free period the 'warm-up period' and analyze the simulation results for the remaining period of time, starting with a rather dry soil. Given the fact that this is a test of several parameterizations and we are not evaluating any scenarios for the area from which the data were obtained we believe this to be a satisfactory solution.

We note that this will, in fact, increase the difference in storage between the start and the end of the simulation. Given the rainfall data record in Fig. 1 (no rainfall for the nearly 300 days at the start, but considerable rainfall during the last 200 days), the soil will be relatively wet at the end of the simulation period. With the inclusion of the 'warm-up period', the simulations will start after a prolonged dry period. We consider this an artefact that is inherent in this extreme environment.

---

## Author Comment (AC2) · 4 Aug 2016

Reply to referee 2

Gerrit H. de Rooij, Raneem Madi, Henrike Mielenz, and Juliane Mai

Reviewer 2 offers a very brief review without real recommendations for improvement.

Reviewer 2 states in the first paragraph of the review: 'The manuscript tries to address several issues, e.g., the deficiency of soil water retention (SWR) models near saturation, SWR models near the dry end, development of a general criterion for plausible hydraulic conductivity (K) curves, comparison of different SWR and K models, different methods for parameter optimization, numerical simulation to evaluate model selection

on drainage and evapotranspiration, and model calibration/inverse model. Even the abstract contains multiple paragraphs, each of which addresses a different issue. As a result, neither of the issues is convincingly addressed.' We disagree with this appraisal, and give three points to build our case.

1) It appears to us that 'the deficiency of soil water retention (SWR) models near saturation, SWR models near the dry end', and the 'development of a general criterion for plausible hydraulic conductivity (K) curves' are really issues that are connected in that they all relate to properties of the parameterizations of soil water retention curves. Therefore, they can all be considered elements of the 'critical evaluation of expressions for the full moisture range' we set out to provide. We fail to see how and why this is confusing, particularly since we explain our approach at the start of section 2.2. We (as well as the other reviewers) believe that section 2.2 does a good job revealing the strengths and weaknesses of the reviewed expressions. Reviewer 2 seems not to have grasped this and cursorily states that this section should be an Appendix.

2) The reviewer states that we were addressing 'different methods for parameter optimization' or 'model calibration/inverse model' (sic), but this is not the case. Neither of these are part of the three objectives we identified in the Introduction, and we did not devote paragraphs to them. The word 'calibration' appears once in the paper, when we use it to introduce the SCE algorithm. The term 'inverse modeling' also appears once, in a sentence in section 4.3 (General ramifications) where we express an interest to see if an inverse modeling of a dynamic experiment would lead to a different choice of parameterization that fitting a curve to hydrostatic data points. Later in the review, it is stated that we should give the method by which we estimated the parameters of the retention functions, which is exactly what we did in section 3.2 of the paper.

3) The statement 'Even the abstract contains multiple paragraphs, each of which addresses a different issue.' is puzzling. HESS guidelines do not require the abstract to be single paragraph, so we do not understand why the reviewer decided to object to that. The structure of the abstract is straightforward: the first paragraph provides

the rationale for the work, explains how we evaluated existing retention functions, and summarizes the main outcome by indicating that only a few parameterizations make physical sense. This part of the abstract therefore summarizes the Introduction and Theory sections of the paper. The second paragraph builds on the first by explaining how we fitted the parameters of the few useful parameterizations and explains that we subjected them to a numerical scenario study (Materials and methods). The third paragraph summarizes the main outcome of the numerical test and gives a recommendation for selecting a suitable parameterization (Results and discussion, General ramifications).

In the following we address some specific points brought up by the reviewer.

As we mentioned above, reviewer 2 recommends transferring section 2.2 to an appendix. In our understanding, appendices are useful for material that is needed in the paper but outside the central flow of thought of the paper. The reviewer does not present a rationale for this change of the paper's structure, other than that the section is long. In defense of the section's length, we would like to point out that if we are going to review soil water retention functions, we need to be reasonably comprehensive. We already left out those parameterizations that are very similar to the ones we included, that are more cumbersome in practical use without offering clear benefits to justify the extra effort by modelers, or that are evidently performing less well than those we included even after a cursory review. We used the critical review in section 2.2 to weed out the functions that we found to be physically unsound. This selection process is a novel element and a main contribution of the paper, as the other two reviewers acknowledge. These findings may have a significant impact on water flow and solute transport modeling practices in unsaturated zone research. Relegating this material to an appendix with the sole argument that the section is long reinforces the impression that the reviewer did not fully grasp the substance of the review or its consequences. We concur with the other reviewers on the relevance of section 2.2.

We presented a generalized criterion to ensure that the soil hydraulic conductivity does

not increase at a physically unrealistically high rate close to saturation. Reviewer 2 states that the correctness of the criterion requires the consideration of experimental errors. This is simply untrue: the derivatives of the mathematical expressions used to describe the shape of the retention curve can be evaluated at saturation. A non-zero slope at saturation implies that there is at least one pore that is still saturated with water at zero matric potential. From the Laplace-Young equation it follows that at least one of the principal radii of curvature of this pore is then infinite, which is a physical impossibility in a realistic soil and for a soil sample of finite dimensions. With the hydraulic conductivity function used, a pore of unbounded size gives rise to an infinite conductivity of this pore. The product of the fraction of the soil cross-section occupied by this hypothetical pore and its conductivity must remain bounded in order to prevent the soil hydraulic conductivity to become infinite at saturation, which is the physical basis of our criterion. This requirement has nothing whatsoever to do with experimental error, it is a physical-mathematical imperative. This was already explained in some detail by Ippisch et al. (2006; reference listed in the paper).

The reviewer requires us to point out how the models that meet our criterion should be used. We believe that users of numerical solvers for Richards' equation have an under-standing of the way in which soil hydraulic properties are used and how to implement them in their solver of choice. The exact operating details depend on the solver and can be obtained from the respective manual or model support website.

The reviewer states that numerical simulations cannot be used to validate models. That may be the case, but it is of limited relevance for our paper since we do not claim to do so. Instead, such simulations can be used to great effect to 'examine the difference in soil water fluxes . . . calculated on the basis of various parameterizations' as we stated in the last paragraph of the Introduction. As we explained we did so because we considered a simple goodness-of-fit test of the soil hydraulic property curves alone (as done by Leij et al., 1997, and Khlosi et al. (2008)) an insufficient comparison: we needed to see how strongly solutions to flow problems differ for different parameterizations. The

results show that this was an effective approach.

We believe that the statement by the reviewer that different solvers can give different results because they use different algorithms is a bit too bold. In practice there are two numerical techniques underlying such algorithms: finite difference and finite element techniques. For both, the mass balance problems that plagued them early on have been resolved in the 1990s, and currently differences between solutions are largely caused by differences in spatial and temporal discretization, not by differences between the different algorithms. The appropriate choice of the spatial grid and the sizing of the time steps is the task of the modeller and depend on the numerical intricacies of the problem that needs to be solved. The reviewer recognizes this, but this does not really bear on the substance of our paper.

---

## Author Comment (AC3) · 4 Aug 2016

Reply to referee 3

Gerrit H. de Rooij, Raneem Madi, Henrike Mielenz, and Juliane Mai

Reviewer 3 gives a good summary of our work and its significance, but states that our literature review has some omissions. The reviewer gracefully provides these missing papers, and we gladly will incorporate these in the text once we studied them. Reviewer 3 more strongly than reviewer 1 criticizes the fact that we used only four soils to test the data on. This may in part be related to the fact that s/he places an emphasis on the behaviour near saturation, while we are equally interested in the very dry range,

where, as reviewer 1 correctly noted, there are fewer data sets that cover that range.

Furthermore, the reviewer mentions two papers in addition to those we already referred to that tested more parameterizations over a wider range of soils, and many of the parameterizations we cover were included in those tests. As we point out in the text, these tests looked at various goodness-of-fit criteria of the retention and the conductivity curve when compared directly to data, but did not evaluate their importance in terms of calculated fluxes when implemented in a Richards' solver. This is an element of our work that contributes something that has not been done to this extent before. We only found something similar in Coppola et al. (2009) (reference provided by this reviewer), for a single field, wet conditions (mainly downward flow) and a short time period (10 days). The focus of that work was different, as it also accounted for field-scale heterogeneity of the soil hydraulic parameters, which we do not consider at this point. Still, our model test covers four soils instead of one, offers a more comprehensive set of atmospheric forcings for a much longer period of time (nearly 3 years!), and tests a much wider array of combinations of retention and conductivity curves. We believe this aspect of our work to be a bit underrated by the reviewer.

2. (There is no comment numbered 1). Like reviewer 1, reviewer 3 point out that we only consider unimodal soils. In response to the comments of these reviewers we will add a section to the Introduction discussing these models in more detail. The references provided by reviewer 3 will be of considerable help in this. An initial reading of these papers revealed that parameter correlation is a point of concern for such models, and detailed and accurate measurements near saturation are of utmost importance. Since our work is done within the framework of groundwater recharge in semi-arid areas that are notoriously data-scarce we believe we should leave the intricacies of multimodal retention curves alone at this stage of the work, and limit ourselves to unimodal retention curves for the time being. This choice notwithstanding, we will endeavour to rewrite the paper in such a way that this choice is explicitly stated, and that the problems associated with aggregated or otherwise structured soils are better represented

in the text in order to let the readers know this is worthy of consideration in the future.

Reviewer 3 suggests that the retention data for the silt loam (Fig. 2d) may point to bi-modality. However, different measurement methods typically have different systematic errors leading to shifts in the retention data at the matric potential where two measurement methods meet. Since in this case the shape is not convincingly bimodal, we are wary to jump to conclusions. The other soils show even less evidence of multimodal behaviour.

3. We refrain from recommending different model combinations for different water content ranges because of our interest in semi-arid regions. There, the top few centimeters of the soil in particular experience the full range of water contents from saturation during infrequent but hydrologically important catastrophic rainfall to nearly oven-dry after months of rainless heat.

4. The error variances for the difference measurement methods were estimated based on such things as sample size in conjunction with the accuracy of the balance. This is explained in lines 554 – 557 in the paper. The reviewer may have overlooked this.

The review lists some additional unnumbered remarks. The first of these suggest to check if we perhaps should rescale the axes of the figures because some curves look very similar. We will review these figures in detail and see what can be done.

Reviewer 3 would like us to discuss why the Alexander-Skaggs conductivity model behaves so differently from Mualem's and Burdine's. We observed that as well, and will see if we can come up with an explanation.

Reviewer 3 is not sure if the flux leaving some of the simulated columns is a numerical artefact. The continued, constant bottom flux during prolonged dry periods is consistent with a mass balance discrepancy that we saw in the simulations, which is why we attributed it to numerical difficulties. Consistent slow convergence also pointed in that direction. A constant flux leaving a drying soil seems physically unlikely: it would require the gradient at the bottom of the profile to increase proportionally to the decrease of the hydraulic conductivity there, and higher up in the profile something similar must occur to deliver the water to the bottom of the profile. If the soil remains equally wet over time, such a constant flux would be consistent with a unit gradient profile with a flow rate that reflects the long-term average net infiltration, but the flux rate is too high for that. But we will revisit these graphs with the reviewer's comments in mind and see if a fresh look brings new insights. The same applies to the comments related to Figure 5.

We thank the reviewer for the suggestions regarding the missing literature, and for offering an interesting analysis of some of our results and other useful suggestions and remarks.

---

## Author Response (AR1)

**Rebuttal letter accompanying the revised version of manuscript HESS-2016-168**

Gerrit H. de Rooij, Raneem Madi, Henrike Mielenz, and Juliane Mai

Below we address the reviewer's comments (in blue italics) point by point. In many cases, our response to these comments led to changes in the paper. These are highlighted in the submitted revision. All figures except Fig. 1-3 were also adapted.

Reviewer 1.
*1.) No information is given on the four soils. I checked in Schelle et al. (2013) and discovered that they investigated samples with these four textures taken from three different sites in Germany and also for disturbed and undisturbed soil. The authors must give specific information on the location of the sampling sites, land use at the sites, whether the samples were disturbed (packed) or undisturbed, sample diameters and number of replicates.*

Response: The requested information was added in section 3.1. All samples for the lower suctions were undisturbed, those for higher suction (where water content is determined by texture rather than structure) are disturbed. For this study, the land use at the sampling sites and their precise locations are not so important (we are not carrying out a site-specific study). The sample dimensions are more important.

*2.) Looking at the figures, I am a little surprised by the apparent lack of structural pores that fill/drain in the tension range close to saturation, say < 10 cm (especially in the finer-textured soils). It does seem to me that throughout the paper the authors only consider the effects of textural pores and do not consider or acknowledge the existence of structural pores. Is this because you only looked at disturbed (packed) samples? Please discuss and clarify this point.*

Response: We focus on unimodal soils only. We made this explicit in the Introduction and added a paragraph discussing structural pores and multimodal models, as requested by the reviewer. At this stage of the research the non-uniqueness problems that the large number of parameters of most multimodal models would interfere with the main objectives of the paper – we would chew off more than we could swallow in one paper.

*3.) The authors comprehensively present the equations of the models, but they write nothing about their conceptual basis. A few introductory sentences are needed to explain the concepts and assumptions underlying these capillary bundle models, including the fact, for example, that they assume a mono-modal size distribution. Alternative approaches could also be mentioned (e.g. bimodal models, fractal models etc.).*

Response: A theoretical background of the equations was added to the Introduction. There seems to be some confusion regarding terminology in this comment. The retention models never are capillary bundle models, only the conductivity models are. This is explained in the revised text. The revision also addresses multimodal models, as requested by the reviewer, but we had more trouble with the term 'fractal models'. The only fractal approach we found is that of Tyler and Wheatcraft (1990), which does not generate a water retention curve but rather underpins the Brooks-Corey model. We were left wondering how a fractal model would generate a water retention curve. In fractal models, the porosity itself is fractal and becomes dependent upon

sample size. The expression for the retention curve would therefore have to be conditioned on the sample size, but we have not come across such a parameterization.

*4.) A more extensive database than four soils would ideally be preferable to enable a reliable discrimination between alternative water retention models although I understand that few datasets include the very dry end of the range. The authors could discuss this.*

Response: Reviewer 3 made a similar comment, but on the other hand kindly provided references to studies that already did just that. When we let go of the simulation tests and increase the number of soils for which we measure the performance of different parameterizations by comparing goodness-of-fit criteria we would be repeating earlier papers, notably Leij et al. (1997) and Khlosui et al. (2008). We argue instead for using simulations of unsaturated flow instead of merely goodness-of-fit to a static curve to evaluate the performance of any parameterization. This procedure is much more laborious than curve-fitting, and with the four soils we used we could already diagnose significant differences. Adding 4 (or 40) soils would not have made a massive difference but would have come at a considerable computational cost. We note that we also considered the conductivity curves associated with the retention curve, adding two or three cases for each parameterization that needed to be run for all soils under consideration. By selecting a limited number of soils that covered a wide range of textures we aimed to keep the work load and the computational burden manageable while at the same time being able to draw solid conclusions about the performance of the various parameterizations. In the revision we more explicitly contrasted our approach to those of earlier papers.

To our knowledge this is the first paper that compares existing parameterizations by including the effect on the soil hydraulic conductivity in the analysis, and we believe this is a significant contribution to the body of literature on the subject. We made that more explicit in the revised text.

*5.) The simulation set-up does not appear to be optimal. My concern relates to the initial condition (hydrostatic equilibrium) in relation to both the bottom boundary condition (unit hydraulic gradient) and the length of the simulation, which was quite short (999 days). Judging from what the authors write (e.g. at lines 733-734), it appears that for this dry climate, this combination results in a simulated water balance that includes a non-negligible term for the change in profile water storage, which is not satisfactory. Water balances in the field should have a negligible change of storage in the long-term and scenario simulations with models should be set-up to mimic this as far as possible. As the authors note, the change of storage is different for the different models, which makes it difficult to compare them with respect to the important terms in the water balance (i.e. recharge, evaporation).*

*The best way to deal with this is to run a 'spin-up' ('warm-up') period first (separately for each model), then use the final state variables (water contents, potentials) at the end of this period as the initial condition for the actual simulation period (using the same driving data for both periods). The water balances for the second simulation period should then be checked to make sure that the change of storage is negligible. If it is not, the warm-up and simulation periods should be extended until it is. Only then can the water balances simulated by the different models be properly compared. If the authors do this, I suspect the differences between the model formulations will be smaller, though probably not negligible.*

*It would also be a good idea to summarize the simulated water balances for the different models in a simple table (precipitation, evaporation, recharge, change of storage).*

Response: We adopted the suggestion of including a warm-up period and modified the text and the figures accordingly. This should lessen the valid concerns of the reviewer regarding the effect of the initial conditions.

We chose not to increase the simulation period to achieve a closed water balance. In semi-arid climates this might take decades or longer, as infiltration from heavy rainfall clusters lingers for decades in the deep unsaturated zone. The meteorological data for such a long-term study are lacking, and we would have had to set up the soil columns differently. We intend to eventually carry out such simulations, but as a follow-up to this work. We will have to rely on simulated rainfall for those studies since the 3-year record we use here is one of the longest we could find for this type of climate.

Incidentally, for our short columns, closure of the mass balance depends less on the duration of the monitoring period than on the strategic choice of the start and end time: if both the start and the end time are chosen after a prolonged dry period, all infiltrated water will have evaporated if all showers were small, and some of it will have contributed to deep drainage if some of the showers were heavy enough. Especially when heavy showers were lacking, not too much can be learned about the effect of the parameterizations on the partitioning between evaporation and deep drainage, even if the mass balance is nearly perfect. We therefore prefer our limited meteorological record, which luckily contained a wide range of rain showers, individually and in clusters.

We clarified the text in that we indicate that we want to study the effect on the fluxes of liquid water and vapor under widely different circumstances: large gradients and sharp contrasts in water content during infiltration after long dry spells, shallow infiltration and subsequent evaporation of small rain showers, and prolonged periods of combined liquid and vapor flow after deeper infiltration of heavier showers. The simulation period includes all these and is therefore well suited to compare the various parameterizations, even when the water balance does not close over the simulated period.

Reviewer 2.
*The manuscript tries to address several issues, e.g., the deficiency of soil water retention (SWR) models near saturation, SWR models near the dry end, development of a general criterion for plausible hydraulic conductivity (K) curves, comparison of different SWR and K models, different methods for parameter optimization, numerical simulation to evaluate model selection on drainage and evapotranspiration, and model calibration/inverse model. Even the abstract contains multiple paragraphs, each of which addresses a different issue. As a result, neither of the issues is convincingly addressed.*

*In my opinion, the development of a general criterion (Eq. 4) for plausible K curves is interesting and can be the main issue of the manuscript. If so, the manuscript needs to provide convincingly theory and experiment results and the conditions a model can or cannot be used. However, to validate the correctness of the criterion, experiment errors need to be considered as well. For example, the SWRs were measured with several methods and the results differ more or less for a given soil. If the difference among different SWR models is less than the measurement error, the*

*SWR model should be fine. The manuscript needs to provide the implications to the readers how they can use the models correctly or appropriately. Section 2.2 is very long and can go to an appendix.*

Response: The reviewer states that we did not address various issues comprehensively, yet later states that section 2.2 should become an Appendix on the grounds that it is long. This leaves us to believe that the reviewer failed to grasp crucial elements of the paper. The most obvious is the fact that nearly all of the parameterizations currently used in numerical Richards solvers are plainly physically wrong, including the one that has become the de facto standard worldwide. This is what section 2.2 sets out to prove, and does so convincingly according to the other reviewers.

This lack of understanding is also apparent from the choice of issues presented in the first paragraph. We pointed to some of the more glaring misconceptions in our on-line reply and will not address these here again. We make an exception for the erroneous thought expressed by the reviewer in the second paragraph that measurement errors somehow affect the validity of the mathematical analysis of the behavior of parametric conductivity models that generate infinite gradients near saturation that translate physically into the existence of a soil pore of infinite radius. This reasoning is so patently flawed that we cannot use it at all to improve the clarity of our text.

*I don't think the numerical simulations using different SWR and K models can be used to validate or invalidate the models. First, modeling evaporation and drainage is challenging and different simulators can produce very different results because they may use different algorithms to solve the problem. Second, some models perform better for certain flow process (e.g., infiltration, redistribution) or soil types while other models perform better for different processes (e.g., evaporation, drainage) or soil types. Third, the assignment of initial and boundary conditions can lead to very different results. For example, for a soil that is never saturated for a simulation, the inaccuracy at the near saturation condition probably does not matter much.*

Response: This paragraph has no relevance for the paper. Not only can model simulations very well be used to compare the performance of different parameterizations of the soil hydraulic properties, they are, in fact, the only viable way to do so considering the fact that goodness-of-fit type evaluations do not give an indication of the effect on water fluxes in soils that different parameterizations have. The sole purpose of the parameterizations of the soil hydraulic property curves is to allow numerical models to quantify fluxes in the unsaturated zone. We therefore fail to see how the reviewer can conclude that such calculations should not be used to test the performance of such parameterizations. The reviewer goes on by mentioning several different aspects of unsaturated flows that could be considered when evaluating the performance of the parameterizations but somehow missed that all of these were represented in the test scenario that we developed.

*The dry-end issue may be left out because it was mentioned but not addressed. The parameter optimization should just be the methods to obtain parameters. It's better if the uncertainty in the optimized parameters be given.*

*For the reasons above, the manuscript is not publishable in the current form.*

Response: Here too, the reviewer is off the mark: the fact that many of the parameterizations tested had their dry branches tailored to better represent observations in the dry range apparently escaped the reviewer. Also unnoticed went the fact that our scenario used boundary conditions representing a desert climate with very long dry spells under high evaporative demand. The explicit comparison between the fluxes generated by parameterizations with and without a specific dry branch also was also ignored. Thus, in this one comment the reviewer disregarded key elements of the theory section, the methodology, and of the results and discussion section.

Reviewer 3
Reviewer 3 starts with summarizing the main objectives and findings of the paper. In our view, this summary is correct, so we will not comment on it further. Below, we therefore only repeat the comments that follow this summary.

*Based on my reading of the manuscript, I think it is in general significant even if the approach is not novel. The manuscript is fairly structured. The introduction of the paper illustrates quite clearly the rationale and the objectives of the work. However, it does not provide an exhaustive literature review about the approach used, with references missing important papers (since the 1990s) dealing with the same issue. Figures and Tables supports the findings, especially the part on the prediction of the hydrological processes selected for analysing the model performance in terms of functional properties.*

Response: We gratefully acknowledge the literature provided by the reviewer and have included the suggested papers and others in the Introduction. We also better explained in the text that including the conductivity curves in the analysis and adding model simulations as a performance assessment tool are novel elements in our paper.

*The strength of the work lies in the fact that the authors provide a systematic and comprehensive review of the WR and HC models available for hydrological analysis. Crucial in the manuscript is the effort to unify and generalize the analyses of the WR behavior provided by the different models, especially near saturation.*

*On the other side, I see some limits in the manuscript that can be summarized as follows: 1. The comparison among models is not novel and is based on a too limited WR and HC dataset . There are papers in the past dealing with the same issue of analyzing the performance of WR and HC functions, based on huge datasets, that the authors do not consider at all. I mainly refer for example to the work by Leij et al. (1997). The authors assembled different types of mathematical formulations and tested them on a large data set crossing practically the whole textural triangle. I would also add Cornelis et al. (2005).*

Response: We included the papers mentioned by the reviewer in the Introduction. Given the availability of these and of Khlosi et al. (2008), we consider the comparison of soil water retention models based on goodness-of-fit to static water retention data points adequately covered in the literature.

We therefore included in the comparison the hydraulic conductivity curves associated with the retention curves according to three models, and evaluated the resulting combinations through numerical simulations. To our knowledge this is the first comparative study to do so, and we believe this contributes meaningfully to the existing body of literature. This approach is much

more laborious than simply fitting curves to data and compare fits. Doing this for 40 instead of 4 soils is hardly feasible, and would not change our conclusions very much: we showed that the choice of parameterization makes a significant difference for four widely different soils, and we expect this to hold for a larger selection of soils as well.

We discussed the literature offered by the reviewer in the text and better explained how our approach differs from and adds to these earlier studies.

*2. The approaches used are all unimodal. As noted by the anonymous Referee #1, the dataset misses most of the information on structural pores. It is not a case that most of the WR curves in figure 2 provide a similar flat behavior in the pF range 0-1. And yet, central in the manuscript is the behavior of the WR functions near saturation. It is well known that HC models based on the statistical capillary-bundle approach, which are based on the Hagen-Poiseuille law and which integrate the reciprocal of the pressure head to obtain the hydraulic conductivity (as in the case of the Mualem conductivity expression), are particularly sensitive to the slope of the water retention near saturation (Durner, 1994; Coppola, 2000). The effects of a wrong description of the WR close to saturation may have impressive effects on the hydraulic conductivity estimation, with an impact on the soil hydrological processes predictions which may well be larger than the effects observed by the authors in their unimodal analysis (see for example Coppola et al., 2012). Bimodality may also exist quite far from saturation. By looking at the figure 2d, the data trend may well suggest a bimodal behavior in the pF range 2-3 (more or less). It is thus not a case that for the silty loam all the WR functions give a poor description of the data in the drier region.*

Response: In the revised text we discuss structural pores and multimodal models. We also explicitly acknowledge that we limit ourselves to unimodal models. We agree with the reviewer's statement regarding the sensitivity of the capillary bundle models near saturation, which was a major motivator for the mathematical analysis we carried out. This analysis is valid for multimodal models as well: each of the underlying unimodal equations should meet the criterion we formulated, so the generalization to this category of models is straight-forward.

We are not sure why the reviewer believes a combination of multiple retention curves would give a dramatically different hydraulic conductivity near saturation. Our analysis showed that models need a non-zero air-entry value to ensure physically realistic behavior of the conductivity. The reviewer's statement implies that she/he believes that the multimodal conductivity (comprised of the conductivities associated with the composing unimodal retention models) differs much more from the unimodal value at the air-entry value than the difference of either of them from the values from either the unimodal or the multimodal conductivity at zero matric potential derived from retention models without water-entry values. But those (incorrect) values will go to infinity (corresponding to the conductivity of a pore with infinite radius), so we think this statement cannot be mathematically correct.

N.B. The conductivity models are formulated as relative conductivities that are scaled by the value of the conductivity at saturation. This forces the relative conductivity at saturation to be equal to zero, even though its gradient there is infinite. The necessary consequence of this therefore is that both the relative and the absolute conductivity drop to zero as soon as the matric potential drops below zero. The physical analysis of the retention curve near saturation shows that there is a fraction of the pore space that only desaturates at zero matric potential, which corresponds to an infinite pore radius, which in turn leads to an infinite conductivity.

The problem with many of the multimodal models is the risk of non-uniqueness brought about by the large number of parameters, and we did not want that aspect to cloud our findings, which is why we chose not to extend the study to multimodal models at this time (see also our response to reviewer 1). The soils we selected had very little or no evidence of multimodality (we are less convinced than the reviewer that Fig. 2d provides evidence of this).

*3. There is no effort for recommending which model combination is the most suitable to be used in a given water content range.*

Response: We believe we should not give such a recommendation. Even in dry climates, infiltration into dry soils will result in a very high degree of saturation at the wetting front, so even then, the full range of water contents and matric potentials will be encountered. Conversely, even moderate climates have precipitation deficits in summer with crops reaching wilting point from time to time and top soils drying out strongly. Only under very humid climates, scenarios with year-long moderate matric potentials can be imagined, but not many simulation studies are carried out for such areas.

We also expect that the most frequently encountered matric potential range depends strongly on specific circumstances: soil type, weather, vegetation/crop, water management and irrigation regime, etc. A researcher with the competence to run a Richards solver is probably knowledgeable enough to make a sound judgement call on what parameterization to choose from the leads we provided in the paper and her/his own expertise and preliminary model runs.

*4. One of the objectives of the paper is " . . . a robust fitting method applicable to various parameterizations and capable of handling data with different data errors" (see end of page 2). The authors introduce the reciprocal of the variance of the errors for weighting differently the single data in the various water content ranges. This is an extension of what one generally does when introducing different quantities in the objective function. And yet, they do not give any information on how they estimated these variances. Selection of data error variances seems to be crucial for determining the performance of the different WR models in describing the experimental data. In other words, the fitting results shown in the figure 2 may be partly an effect of the model parameterization and partly an effect of the error selection for single data.*

Response: We actually give this information in section 3.1. We believe the reviewer overlooked this.

*Some other remarks*
*By looking at the HC curves in the figure 3, it seems that the AS curves of the three soils are almost the same. The same may be said for the Mualem and Burdine curves for the silt and the silt-loam. It may only be a result of the axis extent used for the HC curves. Please, consider to show the curves for a smaller HC range of values. In general, the AS curve remains stably higher than the B and the M HC curves. As the authors are providing ". . .a critical evaluation of parametric expressions", the authors should even shortly explain the reason for this behavior compared to the Burdine and the Mualem models.*

Response: The reviewer states that the AS curves in Fig. 3 are nearly the same for all thee soils. The vertical log scale still results in several orders of magnitude difference in the conductivity, for instance at pF 6. We agree that the curves according to Burdine and Mualem differ little between silt and silt loam, understimating the conductivity for silt and mostly overestimating it for silt loam.

Changing the axes does not bring much benefit in our view - we prefer to show the curves in their entirety and with the possibility to compare between the panels. The reviewer capitalized on by doing exactly that. Given those preferences, the scales are quite adequate. Also, the fact that HESS is an on-line journal allows the reader to zoom in on figures at will.

The reviewer would like us to explain the difference between the conductivity curves accordint to Burdine/Mualem on one hand and Alexander and Skaggs on the other, but we do not see what exactly the reviewer expects us to do. The fits of the retention curves are what they are and the corresponding parameter values translate directly into the three conductivity functions displayed in the figure. The three conductivity models differ only in the (fixed) values of their three additional parameters, resulting in small differences between Burdine and Mualem (also alluded to by van Genuchten (1980)), and a much larger difference between these two models and that of Alexander and Skaggs. The difference can necessarily only be caused by the difference in the values of the three parameters, but there no need to point that out for lack on an alternative explanation. The graphs do a nice job showcasing these differences, and are preferable over a verbal description of those differences.

*In any case, in the section 4.2.1., what the authors consider as "physically implausible" behavior (when discussing about sustained, constant flux leaving the silt soil profile during prolonged dry periods in the AS case) strictly depends on the high values the AS HC curve keeps even for very dry conditions. I would not like it to be also the effect of numerical problems. For example, by looking at the graphs in the panel 4c, the silt-RNA_Mualem/Burdine cumulative drainage curves cross the silt-RNA_AS curve. The latter remains unexpectedly lower, given that the WR curve is the same and the AS curve is systematically higher than the Burdine and the Mualem curves). Maybe, the authors should give more details about the evolution of the pressure head at the bottom boundary conditions during the numerical simulations. They do this only for sand. The reader could do an effort for extending the discussion for the sand to the silt soil. However, the authors may agree that this may be quite laborious.*

Response: We respectfully disagree with the reviewer and stand by our explanation. A constant bottom flux under conditions of zero influx requires the soil to dry out. For the flux to remain constant, the matric potential at the lower boundary must increase to compensate exactly for the non-linear drop in the conductivity brought about by the soil drying. By extension, something similar must happen throughout the drying profile to deliver the water to the lower boundary so it can flow out there.

The reviewer did not consider Fig. 14 in this comment. There we show clear evidence of infiltration during dry periods (with an upper boundary condition of a fixed, very low matric potential), which is direct evidence of numerical irregularities. We doublechecked the input files and found no errors in them. As we point out in the paper, there are various signals from the model output files can calculation times that indicate numerical difficulties.

*In the figure 5, it seems that the evaporation fluxes are inversely related to the drainage fluxes. Higher drainage induces lower pressure head in the soil profile resulting in lower upward fluxes. Again, one should have a look at the pressure gradients at the soil surface. Nonetheless, in the dry region the AS curve may be even five or more orders of magnitude larger than the Mualem and Burdine HC curves. Thus, in the panel 4d (just as an example), I would not expect higher VGA_Mualem than VGA_AS evaporative fluxes, unless the hydraulic gradient at the soil surface in AS case be five or more order of magnitude lower than in the M/B cases.*

*Overall, I have no major problems with the manuscript and recommend publication after the authors have discussed these remarks.*

Response: An inverse relationship between drainage and evaporation fluxes is to be expected: water that leaves the profile through the upper boundary cannot drain through the lower boundary and vice versa.

We believe the reviewer overstates the role of the drainage flux in driving the upward evaporative fluxes. The latter are driven by strong gradients very close to the soil surface. The gradient driving the drainage flux is generated by water 'escaping' to larger depths where the atmospheric boundary condition is not felt that strongly. The large gradients in the top soil dampen the influence of the atmospheric boundary condition at larger depths, and conversely strongly limit the influence of the lower boundary condition on the conditions in the top soil. The difference between AS and Mualem that the reviewer alludes to is largely an effect of the response to the initial condition. In the revision we corrected for that by treating the first 250 d as a burn-in period (as suggested by this reviewer), essentially eliminating the issue.

---

## Referee Report (RR1)

Review of the revised manuscript HESS-2016-168

**Parametric soil water retention models:**
**A critical evaluation of expressions for the full moisture range**

Dear Editor:

After having read the revised manuscript, together with the referees' comments on the original paper and the related responses, I report below my appraisal. The authors made a number of corrections that the previous referees asked for, but I feel that some key points still require being discussed and addressed adequately.

I have listed one general comment and several specific remarks below, the most significant of which are starred (*).

**General comment**

This study fits within the trend of articles that propose refinements or changes to some parametric relations of the soil water retention function (WRF), enabling this property to be better described in the entire range of matric suction head (namely from 0 cm to $10^7$ cm of $H_2O$). In particular, the Authors have designed this work to address the following two main issues (as stated in the abstract): (*i*) to develop a general criterion [i.e. their Eq.(4)] that needs to be met by soil water retention parameterizations to ensure a physically plausible hydraulic conductivity function (HCF) to be obtained, and (*ii*) to select suitable soil hydraulic parametric relations enabling basic hydrologic processes to be well simulated.

The paper reads well, but I would suggest it should be re-organized in a different manner, whereas the tables and figures appear to be satisfactorily. I suggest the dots (observations) in Figs. 2and 3 should be made clearer and well visible.

However, I think the Authors fail to meet their targets adequately as the readership of HESS would expect. The paper, as it stands now, has more the aspect of an internal report than a scientific article. Overall, the manuscript requires major revisions, or should be rejected altogether. At least, after having been re-organized appropriately and better focused, it might be considered for publication as a technical note.

**Specific remarks**

- (*) P.5-26, sections 2.1 and 2.2. Section 2.1 is rather confusing. The title refers to "hydraulic conductivity models", but actually line 149 and some other parts talk about the water retention function. As stated already by the previous Ref.#2, I also strongly suggest that most of section 2.2 should be put in an appendix, while leaving that this part only discusses about the "critical evaluation" of the criterion expressed by Eq.(4). In this way the paper will definitely read better and, more importantly, the reader will capture more directly the essence of the aim of this section.

- (*) P.26, section 3.1. It is not completely clear to me why the Authors used only four soils for their evaluations. This question was also raised by Ref.#1 at his point 4). Two comments from my side on that issue. First of all, it is well known that the problem at hand is highly nonlinear and that the hydraulic response of natural soils varies greatly. In view of this need, putting forward the limitation of computational costs does not seem very convincing. Working with more soils provides a sort of sensitivity analysis of the problem at hand. Schelle et al. (2013) used 8 soils and there are papers in the literature that use as many soils as possible [e.g. Zhang (2011), Chen et al. (2014), Rudiyanto et al. (2015)]. I may agree that, for the sake of effectiveness, in the final part of the paper the scenario analyses are presented only for a limited number of soils.

- P.26, section 3.1. This comment is linked somehow to the previous one and to the Authors' reply to point 4) of Ref.#1 as well as to Ref.#3. To my knowledge, there are papers that have evaluated the effects on estimating or predicting unsaturated hydraulic conductivities. These studies were carried out on the wave of having acknowledged that fitting poorly the WRF, especially close to full saturation (at low suction heads), leads often to a poor prediction of the HCF [e.g. Priesack and Durner (2006), Lebeau and Konrad (2010), Romano et al. (2011)]. Moreover, I would bring to the Authors' attention the recent study by Romano and Nasta (2016) who not only compared different parametric models (both unimodal and bimodal), but also performed functional evaluations when computing the soil-water balance for short-term and long-term periods. Please note that the papers dealing with bimodal and multimodal parametric relationships have also shown indirectly that there is a very good improvement in the description of the water retention toward the dry and very dry range.

- P.28, L.620-626. I understand the rationale behind this choice, although I disagree slightly with this effort. My reasoning for that criticism is that it is easy to show (by simulation, for example) that during the drying of a soil core (relatively small in height and placed on a suction plate or subjected to an evaporation experiment), the soil-water contents change little between its upper and lower ends. The matric suctions change more between these two boundaries and this is why the evaporation experiment works well when using not less than 2 or 3 (or even 4, depending of the height of the core) tensiometers. As far as I know, only the paper by Romano and Santini (1999) discussed somehow that point while indirectly justifying the monitoring of time variations in the average soil-water content by weighing the soil core (at each step in the case of the suction plate, or continuously during the evaporation experiment).

- (*) P.28, L.642-647. One point that deserves attention, and is definitely of interest for a number of more specialized readers, is how the Authors obtained the terms $\sigma_h$ and $\sigma_\theta$ from their experiments. Are these standard deviations constant or depend on the measured $h$ or $\theta$, respectively? It has been found, for example, that the values of $\sigma_h$ increase as $h$ increases. Please provide specific values or relations. This point is also interesting to ascertain the differences in deviations among the different methods employed to cover the full range of the WRFs. On this matter, the Authors may be interested in reading and citing the work by Bittelli and Flury (2009).

- P.28, L.645-646. The reason why the standard deviation, $\sigma$, has to be scaled is not very clear (at least to me). It seems an artifact to adjust the optimization procedure, but this is definitely not in the spirit of the maximum-likelihood method. The errors between observed and computed values are already scaled by the respective standard deviations. The additional scaling of $\sigma$ to improve the fitting performance is not justified by the theory. Did the Authors plot the response surfaces and see what happen around the minima? What kind of improvements are the Authors looking for?

- (*) P.30, section 3.3, L.699-700. I have perhaps missed something, but the selected initial and boundary conditions seem a bit weird and not really consistent among them. Specifically, the imposed initial condition of a soil profile being at hydrostatic equilibrium clashes a bit with the choice of "free drainage" as lower boundary condition (i.e. the unit gradient of the total potential head during the entire process). Commonly, the hydrostatic equilibrium is a final condition that a soil profile reaches when infiltration stops and the profile drains under the effect of gravity, with a constant water table at the lower end of the flow domain. Accounting for a "spin-up time" is a good thing, but I think that the paper improves if the choice of the initial and boundary conditions allows simulating hydrologic processes much closer to real situations.

- P.30, section 3.3. Simulating the long-term water balance in a soil profile initially at hydrostatic equilibrium and under dry conditions (especially at the uppermost soil horizons) is numerically challenging. The Authors should make an effort to convince the reader that the unavoidable numerical errors are lower than the discrepancies associated to the use of the different soil hydraulic parametric relations.

**References cited**

Bittelli, M., and M. Flury, 2009. Errors in water retention curves determined with pressure plates. Soil Sci. Soc. Am. J. 73:1453-1460.

Chen, C., K. Hu, W. Li, G. Wang, and G. Liu, 2014. Estimating the wet-end section of soil water retention curve by using the dry-end section. Soil Sci. Soc. Am. J. 78:1878-1883.

Lebeau, M., and J.-M. Konrad, 2010. A new capillary and thin film flow model for predicting the hydraulic conductivity of unsaturated porous media, Water Resour. Res., 46, W12554, doi:10.1029/2010WR009092.

Priesack, E., and W. Durner, 2006. Closed-form expression for the multi-modal unsaturated conductivity function. Vadose Zone J. 5:121-124.

Romano, N., and P. Nasta, 2016. How effective is bimodal soil hydraulic characterization? Functional evaluations for predictions of soil water balance. Eur. J. Soil Sci. 67:523-535.

Romano, N., and A. Santini, 1999. Determining soil hydraulic functions from evaporation experiments by a parameter estimation approach: Experimental verifications and numerical studies. Water Resour. Res. 35:3343-3359.

Romano, N., P. Nasta, G. Severino, and J.W. Hopmans, 2011. Using bimodal log-normal functions to describe soil hydraulic properties. Soil Sci. Soc. Am. J. 75:468-480.

Rudiyanto, M. Sakai, M.T. van Genuchten, A.A. Alazba, B.I. Setiawan, and B. Minasny, 2015. A complete soil hydraulic model accounting for capillary and adsorptive water retention, capillary and film conductivity, and hysteresis, Water Resour. Res. 51:8757–8772.

Zhang, Z.F. 2011. Soil water retention and relative permeability for conditions from oven-dry to full saturation. Vadose Zone J. 10:1299-1308.

---

## Author Response (AR2)

Dear Editor,

We revised manuscript HESS-2016-168 once more. In doing so we also re-read the review reports from the first round to see if there was overlapping opinions among the five reviewers. In our reply we will focus on the second round reviews only, as we already responded to first three reviewers.
* * *
One reviewer submitted his report entirely through the web text box, without supplements. We will address these comments first. We will adhere to the numbering provided by the reviewer.

1. This reviewer clearly does not like our paper, but also seems to have missed the point. We are unsure why there is an implicit suggestion that we should have tried various parameter estimation techniques when the one we used is well established. The other two points (no new data, and clarity of the best method) have been taken care of by including 21 more soils in the analysis and including our findings fror these soils in the discussion.

2 and 3. We reorganized section 2 by placing it in the supplement and only retaining those parameterizations that were selected for further testing.

4. Corrected as stated and rephrased as suggested.

5. Done.

6. The order in which the four test soils were introduced was partly caused by the fact that the clay samples were from another region in than the rest. We wrote different version of the results section and found that the line of thought was easiest to follow of we started with the intermediate texture and contrasted the coarser and finer textures with the intermediate. The tables are in the same order since they will probably be consulted each time a texture is discussed. The figures are there to compare different textures and were therefore arranged according to texture. There is something to be said for both viewpoints. The labeling is such that ambiguity is avoided, so we kept thigs as they are.

7. Vapor flow played a minor role. We added a statement to that effect at the start of the discussion.

8. Free drainage has always be defined as drainage under unit gradient at the matric potential of the lowest node. A seepage phase requires the matric potential to reach zero at the lowest node before water can leave the soil. Free drainage gives good results when the groundwater tables is so much lower than the lower boundary that it does not affect the matric potential at the lower boundary. We explained what it implies and how it affected early drainage in the Materials and Methods section.

9. One of the authors used pF in an earlier paper in HESS and found that non soil physicists were not always familiar with the term. Soil physicists are used to the pF scale. We resolved this by writing out in full what pF defines in order to conform to established soil physics practice without leaving the rest of the readership in the dark.

10. Done.

11. Before submitting we tried plots with the same axis but found that their readability deteriorated too much for the curves with low values, and therefore we selected the current graphs.

12. This is clear from section 3.2.2.

13. These parameter values for each conductivity model are given in the text. They are not fitted and should therefore not be placed in a table next to a table with fitted parameters to avoid confusion.

14. We rephrased this to clarify that this applies to the relative differences between the fluxes.
* * *
The other reviewer presented a report as a supplement. In his general comment he suggested to make the data points in two figures better visible. In one Fig. 4 (previously Fig. 3) there were so many data points that the overlapping could not be avoided while keeping the data points large enough to remain visible. We redid Fig. 3 (previously Fig. 2) and improved the readability.

The specific remarks by the reviewer are bulleted. We go by those in the order in which they are presented.

1. We rewrote section 2 and created a supplement. This should improve clarity.

2. We added 21 more soils to the analysis, covering a wide range of textures.

3. We considerably expanded the literature review, adding several references kindly provided this reviewer and a reviewer from the first round, and adding some more that we found along the way. We also added a discussion about multimodality in the Introduction and discuss the potential of the parameterizations that we tested for being expanded to multimodal forms.

4. The reviewer discusses the water potential profile during evaporation, while we addressed this under hydrostatic equilibrium. Lui and Dane (1995) found a profound effect. In particular they demonstrated that it could make brooks-Corey type soil with a sharp air-entry value and a power-law retention curve look like a van Genuchten-type soil with a smooth transition without clearly defined air-entry value and a sigmoid shape.

5. We added a discussion about the values of the variance of the data points to the Materials and Methods section. We fully agree with the reviewer that these values depend on the measurement method, and therefore vary with the matric potential.

6. Regarding the scaling of the error standard deviations (ESD), we found this to be necessary because the algorithm was struggling navigating the objective function when ESD values were small compared to the noise in the data. The four test soils had data points from several replicates in one data cloud, and the ESDs did not reflect that noise. More generally, the SCE algorithm works best when the ESDs are no more than an order of magnitude smaller than the values of the variable. We implemented the scaling to ensure that this is the case.

We disagree with the reviewer that this compromises the maximum likelihood attribute of the estimates. The ESD values are used to determine the weighting factor of each data point. The scaling preserved the relative magnitude of each ESD value with respect to all other ESDs, thereby ensuring that the weighting factors are not affected.

7. The reviewer is correct: the initial and boundary conditions are not consistent. The unit gradient imposed at the lower boundary leads to rapid drainage at the start of the simulation because the bottom of the profile is relatively wet. The deep soil dries quickly and consistency is achieved. The spin-up period takes care of this. In the second part of the comment the reviewer claims that conditions can be made more realistic, but we disagree. The free drainage lower boundary condition creates unit gradient flow at the bottom, which is routinely in soils above a groundwater level that is so deep that it does not affect the upper meters of the soil. Implementing this boundary condition eliminates the need to simulate the entire soil profile until the groundwater without much loss of accuracy in the top soil. Particularly in dry climates, hydrostatic equilibrium with the groundwater table is never achieved, the soil is always drier than that.

8. Hydrus reports mass balance errors and the number of iterations required. The user can observe computation times. Only for the Alexander-Skaggs conductivity model did these indicators signal that something was wrong, and we reported that dutifully in the paper.

We thank the reviewer for the list of references. We read these papers and included those that we considered to fit well into the paper.

---

## Author Response (AR3)

Response to the third round of reviews by Gerrit de Rooij and Raneem Madi.

**Reponse to the review by Dr. Coppola**

Dr. Coppola's review indicates that he is largely satisfied with the revisions but would like to have seen multimodal functions included. We therefore carried out an analysis of the three multimodal models we discussed in the Introduction. Only one of them met the criterion for plausible behaviour of the hydraulic conductivity near saturation. None of the multimodal models are well adapted for dry conditions. A modification of the unimodal version of the one multimodal model that was well-behaved near saturation was already analyzed in the supplemental material with discouraging results.

We included this information in the revised version, thereby including multimodal models in our analysis, albeit in concise form. The conclusion must be that at this time, multimodal models have not yet reached a stage of development where they can be used with confidence over the full range of moisture contents.

We discussed the problems with the multimodal models in a way that reflects our optimism that they can potentially be solved. If this paper appears in published form it may stimulate researchers with an interest in multimodal curves to address these issues. If so, we look forward to reading their papers.

By including the multimodal models we also addressed the concern that we discussed multimodal models in the Introduction without addressing them in the analysis.

**Response to the review by referee nr. 4**

This review consists of four paragraphs. We discuss these one by one.

*First paragraph*

We are not sure what is meant by the comment that suggests that the paper is more organized like a report than a paper. The Introduction (mentioned as an example by the referee) reviews the available literature (expanded as recommended by earlier reviews), provides a rationale for our work, and declares its objectives. This is pretty standard for a research paper.

The paragraph in L 407-421 (also brought up as an example) explains why and where the available literature provides compelling arguments against publishing parameter correlations (as suggested in an earlier review) and offers an alternative for detecting parameter interdependence. This too is valid material in a research paper and relevant for the work we present here. It also directly addresses a point of concern raised in the review process.

*Second paragraph*

The referee notes that we spent more space on the wet end (air-entry value, near saturation) than on the dry end of the retention curves. Our review of available retention curve parameterizations showed that most of them potentially can give completely unrealistic hydraulic conductivities near saturation caused by the infinite hydraulic conductivity of non-existent huge pores. In order to demonstrate this rather unsettling result we needed to develop a test criterion and then apply this criterion to all reviewed functions. This naturally takes up space in the paper.

In the dry range the review of the existing parameterizations showed that one has the choice between an asymptotic and a logarithmic dry end. The literature demonstrates that the logarithmic form is superior,

and more often than not was introduced as a modification to an earlier model with an asymptotic dry end with the specific goal to improve its performance for dry conditions. This is clearly documented in the reviewed papers and well established, with no need for additional research or analysis by us. Hence the shorter page count dedicated to this aspect.

These very different demands placed on us to carry out our analysis and reach our conclusions both explains and justifies the difference in amount of space devoted to the dry and the wet end of the retention curve.

In the second part of the paragraph the referee focuses on differences between the suction table method and the evaporation method for measuring data points in the wet end of the retention curve. We are not sure why this is brought up for this theoretical, physical-mathematical paper. That being said, we believe the referee underestimates how short the columns are in modern versions of the evaporation method.

The referee suspects that, because the evaporation process is initiated with the bottom of the column at zero matric potential, a soil column to which this method is applied starts at desaturation because the air entry value may already have been exceeded at some depth range within the column. In modern, commercially available set-ups the columns are five centimeters high, the same height as that of most sampling rings used on suction tables. Suction tables commonly do not exceed zero matric potential at the bottom, so the methodologies suffer from the same issue. We therefore do not share the referee's point of view that this problem arises with the evaporation method but not with the suction table method. With sample rings of 5 cm height, the problem is limited to coarse soils.

The water retention data points at the start of the measurement process (when the soil rings in both methods are at hydrostatic equilibrium) therefore have a comparable accuracy for both methods. The conductivity data in the wet range (which the evaporation method can measure but the suction table method cannot) can be noisy when the soil is so conductive that the evaporation causes only a small matric potential gradient. The two tensiometers in the short columns naturally are only a few centimeters apart and each has its own measurement error, leading to a large potential error in the matric potential gradient for conductive soils. Hence, the problem of the evaporation method in the wet range lies in the conductivity measurement, not in the water retention measurement.

For completeness we point out that our fitting code takes into account the vertical variation of the water content over the height of the column for those data points for which the user specified that the method used relied on hydrostatic equilibrium.

For what it is worth, most of the wet-end retention data for the soils in the UNSODA database were obtained through suction table methods.

*Third paragraph*

The referee expresses some confusion about the text discussing the initial condition (hydrostatic equilibrium) and the lower boundary condition (free drainage).

A careful reading of this comment reveals that the referee understands matric potential profiles under hydrostatic equilibrium and unit gradient conditions. We therefore can safely assume s/he is aware of the change in the lower profile that the change from a hydrostatic initial condition to a free drainage lower boundary condition causes. This invariably involves rapid drainage during the initial stages as the matric potential profile in the lowest part of the column changes from linearly increasing with depth to being constant with depth. Depending on the initial matric head at the bottom of the column, the shape of the retention curve around that matric head, and the corresponding hydraulic conductivity, the duration and amount of the initial drainage varies.

The referee seems to recognize this as well, as evidenced by her/his remarks. All in all it appears that the viewpoints of the referee and us are quite close. We rephrased the text to make it clear that the duration of the transition and the amount of drainage produced by it can be small. We believe this is the only point where we seem to have lost the referee in our train of thought.

We point out that the burn-in period was long enough to ensure that the effect of this initial perturbation of the system did not affect the results.

*Fourth paragraph*

In a paper investigating the effect of soil hydraulic parameterizations it makes sense to select a test problem that is sensitive to these parameterizations. The final paragraph of the referee report therefore supports our choice of the lower boundary condition.

We agree with the referee that the temporal dynamics of flow through the lower boundary are affected by the choice of parameterization. But we are less firmly convinced than the referee that the unit-gradient lower boundary condition should be expected to have a large effect on the magnitude of the cumulative bottom flux. The amount of water available for downward flow is strongly affected by evaporation and storage changes, and therefore heavily depends on processes in the top soil. Our simulation scenario was designed to ensure the occurrence of arid episodes to allow us to examine the effect of the dry end of the retention curve on the partitioning of infiltrated rainfall between evaporation, storage change, and downward flow, and this worked out well: we were able to interpret these aspects in detail. The referee perhaps did not fully appreciate the near-surface dynamics that were an integral element of the simulation part of the study. We slightly modified the text in Materials and Methods to make this more clear.